# The complex Liouville string:
# the worldsheet

**Scott Collier**[1], **Lorenz Eberhardt**[2], **Beatrix Mühlmann**[3,4], **Victor A. Rodriguez**[5,6]

[1]*Center for Theoretical Physics, Massachusetts Institute of Technology, Cambridge, MA 02139, USA*

[2]*Institute for Theoretical Physics, University of Amsterdam, PO Box 94485, 1090 GL Amsterdam, The Netherlands*

[3]*Department of Physics, McGill University Montréal, H3A 2T8, QC Canada*

[4]*School of Natural Sciences, Institute for Advanced Study, Princeton, NJ 08540, USA*

[5]*Joseph Henry Laboratories, Princeton University, Princeton, NJ 08544, USA*

[6]*Department of Physics, University of California, Santa Barbara, CA 93106, USA*

*E-mail:* sac@mit.edu, l.eberhardt@uva.nl, beatrix@ias.edu, varodriguez@ucsb.edu

ABSTRACT: We introduce a new two-dimensional string theory defined by coupling two copies of Liouville CFT with complex central charge $c = 13 \pm i\lambda$ on the worldsheet. This string theory defines a novel, consistent and controllable model of two-dimensional quantum gravity. We use the exact solution of the worldsheet theory to derive stringent constraints on the analytic structure of the string amplitudes as a function of the vertex operator momenta. Together with other worldsheet constraints, this allows us to completely pin down the string amplitudes without explicitly computing the moduli space integrals. We focus on the case of the sphere four-point amplitude and torus one-point amplitude as worked examples. This is the first in a series of papers on the complex Liouville string: three subsequent papers will elucidate the holographic duality with a two-matrix integral, discuss worldsheet boundaries and non-perturbative effects, and connect the theory to de Sitter quantum gravity.

# 1  Introduction

The landscape of vacua of string theory is a wild, largely uncontrolled terrain (see e.g. [1–4]). The situation with two target space dimensions is under comparatively much stronger theoretical control; this simplified setting provides fruitful ground for exploration of difficult problems like the construction of controllable cosmological backgrounds in string theory (a recent discussion appears in [5]) and provides an arena in which one can make progress on the question of the non-perturbative definition of string theory by deriving new holographic dualities. More ambitiously, one might hope to classify the landscape of consistent two-dimensional string theories from more basic physical principles.

In recent years our understanding of the gravitational path integral has benefited from the study of AdS$_2$ Jackiw-Teitelboim (JT) gravity [6–9], which has been understood to be dual to a double-scaled random matrix integral [10, 11]. This novel holographic duality has recently been embedded in two-dimensional string theory, with JT gravity emerging as a semiclassical limit of the *worldsheet theory* of the $(2, p)$ minimal string [12] and of the Virasoro minimal string [13]. This implants the holographic duality between JT gravity and random matrix theory in an older paradigm of dualities between two-dimensional string theories and matrix integrals, with the new conceptual ingredient being the interpretation of the random matrix as the Hamiltonian of a dual quantum system. It is tempting to explore the question of whether a similar stringy realization of dS$_2$ quantum gravity could exist.

**The landscape of minimal string theories.**  In charting the two-dimensional string landscape, the paradigm that has emerged features a two-dimensional string theory defined by coupling some matter CFT to Liouville CFT together with the $\mathfrak{bc}$-ghosts on the worldsheet

$$
\begin{array}{ccccc}
\text{matter CFT} & & \text{Liouville CFT} & & \mathfrak{bc}\text{-ghosts} \\
c = c_m & \oplus & c = 26 - c_m & \oplus & c = -26
\end{array} ,
\tag{1.1}
$$

which admits an equivalent description in terms of a matrix integral.

An important class of examples that has been well-explored in the literature is furnished by taking the matter CFT to be a Virasoro minimal model [14–20]. In the case of a $(2, p)$ minimal model, there is significant evidence that the $(2, p)$ minimal string is dual to a double-scaled matrix integral. It has been argued that the worldsheet theory reduces to JT gravity in the $p \to \infty$ limit in which $c_m \to -\infty$ [12], although this limit has not been fully understood. On the other hand, the more general $(p, q)$ minimal string is conjecturally dual to a double-scaled *two-matrix* integral [21, 22].

An irrational analog of the $(2, p)$ minimal string, the Virasoro minimal string, was recently introduced in [13]. This model is defined by coupling Liouville CFT with central charge $c \geqslant 25$ to "timelike Liouville CFT" [23–25] with central charge $26 - c \leqslant 1$ on the worldsheet, and was shown to be equivalent to a double-scaled matrix integral (aspects of the matrix integral were studied further in [26–28]). The matrix integral is fully determined at the level of perturbation theory by its leading density of eigenvalues, which is given by the universal Cardy density of states in compact 2d CFT of central charge $c$. The string amplitudes of the theory — referred to as *quantum volumes* $\mathsf{V}_{g,n}^{(b)}(ip_1, \ldots, ip_n)$ — were shown to be finite-degree polynomials in the external momenta $p_j$ that label the on-shell vertex operators. The worldsheet theory admits a semiclassical description in terms of two-dimensional dilaton gravity with a sinh potential for the dilaton, which in the semiclassical $c \to \infty$ limit is equivalent to JT gravity. Indeed, in this limit the quantum volumes reduce to the Weil-Petersson volumes of the moduli space of Riemann surfaces. The loop equations/topological recursion of the matrix integral [29] translate into a recursion relation for the quantum volumes that is a deformation of Mirzakhani's recursion relation for the Weil-Petersson volumes [30].

At this point we ask: what other kinds of physics is the two-dimensional string landscape capable of hosting? And what is the nature of the dual descriptions? As we will show in a series of papers [31–33] starting with this one, the two-dimensional string landscape contains a multitude of further lessons. This paper is an expanded version of the corresponding section of [34].

**A new minimal string.** The main purpose of the present paper is to explore a new two-dimensional string theory that we will refer to as the *complex Liouville string*. It is defined at the level of the worldsheet CFT by coupling two copies of Liouville CFT with complex central charge that may loosely be regarded as complex conjugates of one another:

$$
\begin{array}{ccccc}
\text{Liouville CFT} & & (\text{Liouville CFT})^* & & \mathfrak{bc}\text{-ghosts} \\
c^+ = 13 + i\lambda & \oplus & c^- = 13 - i\lambda & \oplus & c = -26
\end{array}, \tag{1.2}
$$

where $\lambda \in \mathbb{R}_+$. The Virasoro minimal string and the complex Liouville string may be thought of as irrational cousins of the $(2, p)$ and $(p, q)$ minimal string theories, respec-

tively. They are in some sense even simpler theories than their more conventional minimal string counterparts because the string amplitudes are analytic functions of both the central charge and the external momenta. They hence provide tractable testing grounds in which to explore fundamental aspects of string theory, particularly holographic duality and non-perturbative effects.

**Bootstrapping string amplitudes.** The main observables of the theory are string amplitudes $\mathsf{A}_{g,n}^{(b)}(p_1, \ldots, p_n)$, which are defined by integrating worldsheet correlation functions over the moduli space of Riemann surfaces. In contrast to the Virasoro minimal string, we will see that the string amplitudes are not simply polynomials in the external momenta $p_i$. Nevertheless, we can leverage the exact solution of the worldsheet CFT (1.2) [35–37] to deduce the analytic structure of the string amplitudes viewed as analytic functions of the external momenta. In particular, we will see that the string amplitudes exhibit

- An infinite set of poles associated with resonances of the Liouville CFT correlation functions.

- An infinite series of discontinuities that arise when the moduli integral that defines the string amplitudes ceases to converge upon analytic continuation in the external momenta.

This places extremely strong constraints on the string amplitudes. There are yet further constraints from the worldsheet definition of the string amplitude, including a "dilaton equation" that relates the string amplitudes with one of the external momenta tuned to a special value to lower-point amplitudes. This allows us to initiate a bootstrap program that harnesses the analytic structure and other worldsheet considerations to pin down the string amplitudes without directly evaluating the moduli space integral. Remarkably, we will see that the quantum volumes $\mathsf{V}_{g,n}^{(b)}$ of the Virasoro minimal string play a role as building blocks of the string amplitudes $\mathsf{A}_{g,n}^{(b)}$. This technique seems powerful and may be further developed in its own right — it is conceptually similar to the S-matrix bootstrap, but with some input from the worldsheet formulation incorporated.

**Solutions for low $(g, n)$.** In this paper we will investigate in particular the case of the sphere four-point amplitude $\mathsf{A}_{0,4}^{(b)}$ in great detail. In this case there are even more powerful constraints, including an SO(8) triality symmetry that follows from a property of the Liouville four-point function, and "higher equations of motion" that generalize the dilaton equation [38, 39]. We will see that remarkably, the solution to these constraints is *unique*, up to a mild assumption on the asymptotic growth of the amplitude. Moreover, due to a relation between the relevant moduli spaces and properties of Liouville CFT data and conformal blocks [40], the torus one-point amplitude $\mathsf{A}_{1,1}^{(b)}$ may be recovered as a special case of the sphere four-point amplitude.

Collectively, this analysis culminates in the following solutions for the sphere three-point, torus one-point, and sphere four-point amplitudes, respectively:[12]

$$\mathsf{a}_{0,3}^{(b)}(\boldsymbol{p}) = \sum_{m=1}^{\infty} \frac{2b(-1)^m \sin(2\pi mbp_1)\sin(2\pi mbp_2)\sin(2\pi mbp_3)}{\sin(\pi mb^2)} , \tag{1.6a}$$

$$\mathsf{a}_{1,1}^{(b)}(\boldsymbol{p}) = \sum_{m=1}^{\infty} \frac{(-1)^m b\sin(2\pi mbp_1)}{\sin(\pi mb^2)} \left( \mathsf{V}_{1,1}^{(b)}(ip_1) - \frac{1}{16\pi^2 b^2 m^2} \right) , \tag{1.6b}$$

$$\mathsf{a}_{0,4}^{(b)}(\boldsymbol{p}) = \sum_{m=1}^{\infty} \frac{2b^2 \mathsf{V}_{0,4}^{(b)}(ip_1, ip_2, ip_3, ip_4) \prod_{j=1}^{4} \sin(2\pi mbp_j)}{\sin(\pi mb^2)^2}$$

$$- \sum_{m_1,m_2=1}^{\infty} \frac{(-1)^{m_1+m_2} \sin(2\pi m_1 bp_1)\sin(2\pi m_1 bp_2)\sin(2\pi m_2 bp_3)\sin(2\pi m_2 bp_4)}{\pi^2 \sin(\pi m_1 b^2)\sin(\pi m_2 b^2)}$$

$$\times \left( \frac{1}{(m_1+m_2)^2} - \frac{\delta_{m_1 \neq m_2}}{(m_1-m_2)^2} \right) + 2 \text{ perms } . \tag{1.6c}$$

Much of this paper will be devoted to explaining the equality between the string amplitudes $\mathsf{A}_{0,3}^{(b)}$, $\mathsf{A}_{1,1}^{(b)}$, and $\mathsf{A}_{0,4}^{(b)}$ and the proposals listed in (1.6). In the latter two cases we are able to perform the integral over moduli space directly and hence verify the proposals (1.6c) and (1.6b) to a high degree of numerical precision (whereas (1.6a) can be deduced analytically from the structure constants of the worldsheet CFT).

**A dual two-matrix integral.** Like the other examples discussed above, the complex Liouville string also admits a dual matrix integral description. In a companion paper [31], we will argue that the complex Liouville string may be reformulated as a double-scaled *two-matrix* integral. The perturbative expansion of this matrix integral is fully fixed by the geometry of its spectral curve, which exhibits infinitely many nodal singularities and branch points. The topological recursion of the matrix integral [29, 41, 42] leads to a recursion relation for the string amplitudes $\mathsf{A}_{g,n}^{(b)}$ akin

---

[1] Here $b$ labels the central charge of one of the Liouville CFTs by the usual relation

$$c = 1 + 6(b + b^{-1})^2 , \tag{1.3}$$

while the momenta $p_i$ label the on-shell vertex operators in a way that we will make precise in the following section.

[2] $\mathsf{V}_{1,1}^{(b)}$ and $\mathsf{V}_{0,4}^{(b)}$ are the analytic continuation of the Virasoro minimal string quantum volumes, which are given by the following polynomials [13]

$$\mathsf{V}_{1,1}^{(b)}(ip_1) = \frac{1}{24} \left( \frac{c-13}{24} - p_1^2 \right) \tag{1.4}$$

$$\mathsf{V}_{0,4}^{(b)}(ip_1, ip_2, ip_3, ip_4) = \frac{c-13}{24} - p_1^2 - p_2^2 - p_3^2 - p_4^2 . \tag{1.5}$$

to Mirzakhani's recursion relation for the Weil-Petersson volumes [30] (and its deformation for the quantum volumes of the Virasoro minimal string), which fully solves the theory at the level of string perturbation theory.

**Low-dimensional de Sitter quantum gravity.** Like the Virasoro minimal string, the worldsheet theory of the complex Liouville string admits a semiclassical reformulation in terms of two-dimensional dilaton gravity via a field redefinition from the Lagrangian description of Liouville CFT. In particular we will see that we land on a model with a sine potential for the dilaton (rather than the sinh potential that arose in the Virasoro minimal string). Intriguingly, this model admits *both* $AdS_2$ and $dS_2$ vacua as classical solutions. Hence the complex Liouville string provides a fully rigorous and well-defined model of 2d quantum gravity that includes de Sitter solutions. For now we defer further discussion of the path integral description of the model to another companion paper [32].

Moreover, as will be discussed in [33], the string amplitudes $\mathsf{A}_{g,n}^{(b)}$ may be interpreted as late-time cosmological correlators of massive particles in three-dimensional Einstein gravity with positive cosmological constant. This establishes a novel holographic duality between the dual matrix integral and cosmological correlators in $dS_3$.

**Outline of the paper.** The rest of this paper is organized as follows. We begin by introducing the worldsheet CFT and the definition of the string amplitudes in section 2. We discuss some simple properties satisfied by the general string amplitudes and explicitly evaluate the sphere two- and three-point amplitudes. In section 3 we explore in great detail the properties of the sphere four-point amplitude. We demonstrate in particular that the amplitude exhibits an infinite set of poles and discontinuities as a function of the external momenta, which greatly constrain the analytic form of the amplitude. In section 4 we describe a technique that allows us to evaluate the string amplitudes when one of the external momenta is tuned to special degenerate values, which constitute further constraints on the string amplitudes. The technique hinges on a relation known as *higher equations of motion* [38] between such degenerate vertex operators and total derivatives of logarithmic operators built out of the ground ring of the worldsheet CFT, which allows us to localize the moduli integral that defines the string amplitude to the boundary of moduli space. In section 5 we show that our bootstrapped proposals for the sphere four-point and torus one-point amplitudes in (1.6) satisfy all constraints on the string amplitudes that we derived from the worldsheet. Moreover these solutions are unique given a mild assumption on the asymptotic growth of the amplitudes. Finally in section 6 we directly evaluate the moduli integral that defines the sphere four-point amplitude and show that it agrees with the proposal (1.6c) to a high degree of numerical precision. Appendices A, B and C collect some further details and computations.

## 2 Definition of the worldsheet theory

### 2.1 Unitarity and spectrum

As already mentioned in the introduction, we consider a worldsheet theory of two coupled Liouville theories of central charges $c = c^+ = 13 + i\lambda$ and $c^- = 13 - i\lambda$, $\lambda \in \mathbb{R}_+$. We will now be more precise from an axiomatic point of view what is meant by this theory.

**Liouville theory and its analytic continuation.** Liouville theory may be formulated in terms of the following classical action for a scalar field $\varphi$

$$S[\varphi] = \frac{1}{4\pi} \int_{\Sigma_{g,n}} \mathrm{d}^2x \sqrt{\tilde{g}} \left( \tilde{g}^{ij} \partial_i \varphi \partial_j \varphi + Q \widetilde{\mathcal{R}} \varphi + 4\pi\mu \mathrm{e}^{2b\varphi} \right) . \tag{2.1}$$

Here $\mu$ is a dimensionful parameter of the theory that can be absorbed in the string coupling by shifting $\varphi$. The parameters $Q$ and $b$ are related, rendering Liouville theory a two-dimensional conformal field theory with central charge

$$c = 1 + 6Q^2 , \quad Q = b^{-1} + b . \tag{2.2}$$

For real $b \in (0, 1]$ the central charge carves out the positive real axis $c \geqslant 25$. In this work we are instead interested in the case of complex central charge, with a particular emphasis placed on the case

$$c = c^+ \in 13 + i\mathbb{R}_+ , \quad b = b^+ \in \mathrm{e}^{\frac{i\pi}{4}}\mathbb{R} . \tag{2.3}$$

This renders the Liouville action (2.1) complex-valued and thus the definition of the theory from the path integral becomes unclear. However, it can be defined axiomatically by analytically continuing the OPE data away from real $b$ [43] while preserving crossing symmetry of the correlation functions. The only shortcoming of Liouville theory at complex central charge is the non-existence of an inner product, since the reality conditions on the Virasoro algebra $L_n^\dagger = L_{-n}$ are incompatible with a complex central charge.

The worldsheet theory (1.2) combines the complex Liouville theory (2.3) with the complex conjugate theory rendering the combined theory real valued. The reality conditions on the fields are such that the Liouville fields $\varphi$ (2.1) are complex and the second Liouville field is the complex conjugate of the first.

Although we have introduced the worldsheet theory in terms of the Liouville Lagrangian, in what follows we will treat Liouville theory solely as a non-perturbatively well-defined conformal field theory defined by analytic continuation of the OPE data, and postpone a path integral perspective to [32].

**Reality conditions.** Let us explain axiomatically how the combination of the two Liouville theories salvages the existence of a (non-unitary) inner product. Let us denote the Virasoro generators of the two Liouville theories by $L_m^+$ and $L_m^-$. The total worldsheet stress-tensor is hence given by

$$L_m = L_m^+ + L_m^- \ . \tag{2.4}$$

The reality conditions on the Liouville fields imply that we should have the following reality condition on these Virasoro generators:

$$(L_m^+)^\dagger = L_{-m}^- \ . \tag{2.5}$$

This in particular ensures that the total stress-tensor (2.4) is real. Compatibility of the Virasoro algebra with the hermitian adjoint tells us that $(c^+)^* = c^-$ and forces the central charges to have the stated form. We also remark that the form of the central charge equivalently can be written as

$$c^\pm = 1 + 6\big((b^\pm)^{-1} + b^\pm\big)^2 \ , \quad (b^\pm)^2 \in i\mathbb{R} \tag{2.6}$$

with

$$b^- = -ib^+ \ . \tag{2.7}$$

We hence choose $b^+ \in e^{\frac{\pi i}{4}}\mathbb{R}$ and $b^- \in e^{-\frac{\pi i}{4}}\mathbb{R}$. The global Virasoro modes $L_{0,\pm 1}^+$ and $L_{0,\pm 1}^-$ now combine to form a $\mathrm{PSL}(2,\mathbb{C})$ Möbius symmetry (contrary to the $\mathrm{PSL}(2,\mathbb{R})$ Möbius symmetry that we encounter in a generic 2d CFT).

**Vertex operators.** At the level of the spectrum, primary vertex operators are labelled by two conformal weights $h^+$ and $h^-$ associated with primaries of the two Liouville CFTs. Since $(L_0^+)^\dagger = L_0^-$, we must have that $(h^+)^* = h^-$. Together with the on-shell condition $h^+ + h^- = 1$ of string theory, we hence find that

$$\mathrm{Re}(h^+) = \mathrm{Re}(h^-) = \frac{1}{2} \ . \tag{2.8}$$

In other words, physical vertex operators are precisely the Virasoro analogue of the principal continuous representations with respect to the $\mathrm{PSL}(2,\mathbb{C})$ Möbius subgroup. As usual in Liouville theory, it is convenient to parametrize the conformal weights via their Liouville parameters,

$$h^\pm = \frac{c^\pm - 1}{24} - (p^\pm)^2 \ . \tag{2.9}$$

Here we adopt a convention for the Liouville momenta such that the spectrum of ordinary Liouville theory would be supported on the imaginary line, $p \in i\mathbb{R}$.[3] In

---

[3]We use a lower case letter to distinguish it from the convention use in our previous paper [13], where $P = ip$.

our case, the reality conditions that we discussed instead imply that the Liouville momenta take the form

$$(p^\pm)^2 \in i\mathbb{R} \tag{2.10}$$

with

$$p^- = \pm i p^+ \tag{2.11}$$

as the physical state condition, i.e. the Liouville momenta are rotated by 45 degrees in the complex plane. We will usually assume that

$$p^+ \in \mathrm{e}^{-\frac{\pi i}{4}}\mathbb{R} \ , \qquad p^- \in \mathrm{e}^{\frac{\pi i}{4}}\mathbb{R} \ , \tag{2.12}$$

since the opposite choice is obtained by exchanging the two theories, see eq. (2.25) below. We can also state this by saying that

$$b^+ p^+ \in \mathbb{R} \ , \qquad b^- p^- \in \mathbb{R} \ . \tag{2.13}$$

On-shell primary vertex operators are products of primaries in the two Liouville theories,

$$V^+_{p^+=p}(z) V^-_{p^-=ip}(z) \ , \tag{2.14}$$

but recall that $V^+_{p^+}$ and $V^-_{p^-}$ themselves are not real. We similarly will write $b \equiv b^+$ in what follows.

**Global conformal transformations.** We let $L^+_{-1}$ and $L^-_{-1}$ act as usual as $\partial_z$ derivative on the first and second factor of the vertex operator (2.14). Because we Wick rotated the worldsheet signature to Euclidean signature, this does not respect the reality conditions of the algebra as usual in 2d CFT.[4] However from this point on the computation of worldsheet correlators etc proceeds in the standard way. We provide some more details on the reality condition of the worldsheet theory in appendix A, where we in particular prove a no-ghost theorem. Remarkably, physical vertex operators precisely transform in unitary principal series representations of the PSL$(2,\mathbb{C})$ Möbius subgroup of the first or second theory. The inner product is simply the L$^2$-inner product and thus integrated vertex operators may be thought of as computing the norm of the states in the principal series representation. We refer to [44] and appendix A for more details of PSL$(2,\mathbb{C})$ representation theory. It will not be needed in the rest of the paper.

## 2.2 Definition of the string amplitudes

**Structure constants.** Aside from the norm on the Hilbert space, the worldsheet theory is given by a direct product of two Liouville theories. Correlation functions of local operators in the worldsheet CFT are fixed entirely by their structure constants.

---

[4]This actually performs a *double* Wick rotation – both on the worldsheet and in spacetime. We explain this in more detail in appendix A.

The structure constants of Liouville CFT are well-known and are given by the DOZZ formula [35, 36]. The DOZZ formula can be derived without the knowledge of the reality conditions, for example by considering a certain degenerate case of crossing symmetry [45]. This means that the structure constants of each factor should still be given by the DOZZ formula, which takes the form

$$C_b(p_1, p_2, p_3) = \frac{\Gamma_b(2Q)\Gamma_b(\frac{Q}{2} \pm p_1 \pm p_2 \pm p_3)}{\sqrt{2}\Gamma_b(Q)^3 \prod_{j=1}^3 \Gamma_b(Q \pm 2p_j)} \ . \tag{2.15}$$

We chose the same conventions as in [13], since these conventions are conveniently reflection-symmetric. The function that appears in the OPE measure in these conventions reads

$$\rho_b(p) = 4\sqrt{2}\sin(2\pi b p)\sin(2\pi b^{-1} p) \ . \tag{2.16}$$

**Four-point amplitude.** We can hence readily compute the four-point function in Liouville CFT,

$$\langle V_{p_1}^+(0)V_{p_2}^+(z)V_{p_3}^+(1)V_{p_4}^+(\infty)\rangle = i \int_0^{e^{-\frac{\pi i}{4}}\infty} \mathrm{d}p\, \rho_b(p)$$
$$\times C_b(p_1, p_2, p)C_b(p_3, p_4, p)\mathcal{F}_{0,4}^{(b)}(\boldsymbol{p}; p|z)\mathcal{F}_{0,4}^{(b)}(\boldsymbol{p}; p|z^*) \ . \tag{2.17}$$

Here we have used the short-hand notation $\boldsymbol{p} = (p_1, p_2, p_3, p_4)$. After using that $b^- = -ib = (b^+)^*$ and $p^- = ip^+ = (p^+)^*$, we can also write the four-point function of on-shell vertex operators that appears in the string amplitude manifestly as an absolute value squared,

$$\langle V_{p_1}^+(0)V_{p_2}^+(z)V_{p_3}^+(1)V_{p_4}^+(\infty)\rangle \langle V_{ip_1}^-(0)V_{ip_2}^-(z)V_{ip_3}^-(1)V_{ip_4}^-(\infty)\rangle$$
$$= \left| \int_0^{e^{-\frac{\pi i}{4}}\infty} \mathrm{d}p\, \rho_b(p)C_b(p_1, p_2, p)C_b(p_3, p_4, p)\mathcal{F}_{0,4}^{(b)}(\boldsymbol{p}; p|z)\mathcal{F}_{0,4}^{(b)}(\boldsymbol{p}; p|z^*)\right|^2 \ . \tag{2.18}$$

The integration contour $e^{\pm\frac{\pi i}{4}}\mathbb{R}_{\geqslant 0}$ is not the standard integration contour of Liouville theory, where one would integrate over the line $i\mathbb{R}_{\geqslant 0}$. We rotated it because it corresponds to the spectrum of the theory (2.12). By considering the poles of the DOZZ structure constants, one can easily see that the poles are well-separated from the contour of integration and hence there is no need for any additional discrete contributions to the conformal block decomposition.

This definition of the worldsheet theory allows us to define string amplitudes, which we denote by $\mathsf{A}_{g,n}^{(b)}(p_1, \ldots, p_n)$, by integrating the correlators over moduli space. For example, in the case of the four-punctured sphere, we set

$$\mathsf{A}_{0,4}^{(b)}(p_1, p_2, p_3, p_4) \equiv C_{\mathrm{S}^2}^{(b)} \prod_{j=1}^4 \mathcal{N}_b(p_j) \int \mathrm{d}^2 z\, \left|\langle V_{p_1}^+(0)V_{p_2}^+(z)V_{p_3}^+(1)V_{p_4}^+(\infty)\rangle\right|^2 \ . \tag{2.19}$$

Here, $C_{\mathrm{S}^2}^{(b)}$ describes the a priori arbitrary normalization of the string path integral on the sphere, while the so-called leg-factors $\mathcal{N}_b(p)$ correspond to a change of normalization of the vertex operators. It turns out to be convenient to set them to

$$\mathcal{N}_b(p) = -\frac{(b^2 - b^{-2})\rho_b(p)}{8\sqrt{2}\pi\, p \sin(\pi b^2)\sin(\pi b^{-2})} \ , \tag{2.20a}$$

$$C_{\mathrm{S}^2}^{(b)} = 32\pi^4 \left(\frac{\sin(\pi b^2)\sin(\pi b^{-2})}{(b^2 - b^{-2})}\right)^2 \ . \tag{2.20b}$$

This is entirely conventional, but will lead to nicer expressions for the string amplitudes and makes the map to the dual matrix integral discussed in [31] simpler. Notice in particular that the leg factor is *odd* under reflection $p \to -p$ and hence all string amplitudes will be odd functions under reflection of the individual Liouville momenta. Similarly, for the one-point function on the torus, we set

$$\mathsf{A}_{1,1}^{(b)}(p_1) \equiv \frac{1}{2}\,\mathcal{N}_b(p_1)\int_{\mathcal{F}} \mathrm{d}^2\tau\, |2\pi\eta(\tau)^2|^2 \big|\langle V_{p_1}^+(0)\rangle\big|^2 \ . \tag{2.21}$$

The factor of $\frac{1}{2}$ originates from the $\mathbb{Z}_2$ automorphism of the torus. There is no need to include an additional factor $C_{\mathrm{T}^2}^{(b)}$, since the CFT torus partition function is unambiguous. The factor $|2\pi\eta(\tau)^2|^2$ is the ghost contribution in a particular normalization.

**General amplitudes.** We can similarly define any perturbative string amplitude $\mathsf{A}_{g,n}^{(b)}$ by integrating the worldsheet correlators over the moduli space. This requires us to appropriately include the standard string theory $\mathfrak{bc}$-ghosts,

$$\mathsf{A}_{g,n}^{(b)}(\boldsymbol{p}) \equiv C_{\Sigma_g}^{(b)} \int_{\mathcal{M}_{g,n}} \left\langle \prod_{k=1}^{3g-3+n} \mathcal{B}_k \widetilde{\mathcal{B}}_k \prod_{j=1}^{n} \mathcal{V}_{p_j} \right\rangle_{\Sigma_{g,n}} \ , \tag{2.22}$$

where the $\mathcal{V}_p = \mathfrak{c}\tilde{\mathfrak{c}}\mathcal{N}_b(p)V_p^+ V_{ip}^-$ are vertex operators representing on-shell closed string states, $\mathcal{B}_k$ and $\widetilde{\mathcal{B}}_k$ are the holomorphic and anti-holomorphic $\mathfrak{b}$-ghost insertions associated with the moduli of $\Sigma_{g,n}$, and $C_{\Sigma_g}^{(b)}$ are normalization constants of the string path integral on a genus $g$ Riemann surface $\Sigma_g$. The correlator is taken in the full worldsheet theory including the ghosts.

We should remark that those integrals are absolutely convergent and hence our string theory is perturbatively completely well-defined. Indeed, there are potential divergences in the integral (2.19) as, say, $z \to 0$. By using the known behaviour of the Liouville correlators in this limit [46, eq. (3.13)], we see that the integrand behaves as $|z|^{-2}(-\log|z|)^{-3}$ as $z \to 0$, which ensures that the integral is convergent. A similar analysis can be carried out for any $(g,n)$.

Let us comment on the physical reason for this somewhat unusual property in string theory. Normally, the worldsheet spectrum before imposing physical state

conditions contains both states with total conformal weight $h < 1$ and $h \geqslant 1$, which in particular means that the worldsheet OPEs can have singular behaviour as say one puncture approaches another. This is very important since divergences in the moduli space integral produce physical singularities such as poles and branch cuts in string amplitudes. In the present case, all conformal weights are of the form $h \in 1 + i\mathbb{R}$, which means that the integrals are all barely convergent. In the Virasoro minimal string [13], one also gets an absolutely convergent integral, but one is forced to consider an 'internal spectrum' appearing in the OPE expansion that is different from the 'external spectrum' that one uses to define physical vertex operators [47].

**Sum over genera.** Of course we sum over all genera in the full theory. We set

$$\mathsf{A}_n^{(b)}(S_0; \boldsymbol{p}) \equiv \sum_{g=0}^{\infty} \mathrm{e}^{(2-2g-n)S_0} \mathsf{A}_{g,n}^{(b)}(\boldsymbol{p}) \ . \tag{2.23}$$

Motivated by the connection of the worldsheet theory to 2d dilaton gravity, we denote the string coupling constant by $g_{\mathrm{s}} = \mathrm{e}^{-S_0}$. This sum is asymptotic and we will discuss the resurgence properties and non-perturbative effects elsewhere [32].

**Analytic continuation.** We defined the string amplitudes for $b \in \mathrm{e}^{\frac{\pi i}{4}}\mathbb{R}$, $p_j \in \mathrm{e}^{-\frac{\pi i}{4}}\mathbb{R}$. However, one can easily define analytic continuations outside of this physical regime by disregarding the reality conditions. The analytic structure of the string amplitudes will be a very important clue about their closed-form expressions. When analytically continuing, we need to preserve $c^+ + c^- = 26$ and $h^+ + h^- = 1$ for physical vertex operators, since this is needed to get well-defined integrals over moduli space. However, we can take $b^+$ and $p^+$ essentially arbitrary as long as $b^- = -ib^+$ and $p^- = ip^+$ in the definition of the four-point function (2.17) on the worldsheet (or any other worldsheet correlation function). The equation (2.17) remains valid, but the right-hand-side of (2.18) and hence the integrands of (2.19) and (2.21) are no longer an absolute value squared. In a vicinity of the physical spectrum, convergence of the integral (2.19) is unaffected by analytic continuation and thus we obtain an analytic function. As we shall discuss in section 3.1, the global analytic structure is quite intricate and $\mathsf{A}_{0,4}^{(b)}(p_1, p_2, p_3, p_4)$ has in particular various poles coming from resonances in the Liouville correlators and branch cuts coming from divergences in the integral over moduli space.

## 2.3 Simple properties

Let us discuss a few immediate properties of the string amplitudes as defined in section 2.2, see in particular (2.19) for the definition of the sphere four-point amplitude. Although in this section we mostly focus on the sphere four-point amplitude $\mathsf{A}_{0,4}^{(b)}$, these simple properties can be stated for the general string amplitudes $\mathsf{A}_{g,n}^{(b)}$. We discuss less obvious properties in section 3.

**Oddness in $p$.** While the Liouville structure constant $C_b(p_1, p_2, p_3)$ is even under reflection, e.g. $C_b(p_1, p_2, p_3) = C_b(-p_1, p_2, p_3)$, the leg factor $\mathcal{N}_b(p)$ is odd. As a result, the string amplitudes are also odd under reflection. This is perhaps unfamiliar since all CFT quantities depend only on $p^2$, but it will turn out to be a convenient convention. We will usually assume that $bp \in \mathbb{R}_{\geqslant 0}$ for external states.

**Duality and swap symmetry.** The worldsheet theory only depends on the central charge and thus we have the symmetry under $b \to -b$ and $b \to b^{-1}$. Since the path integral normalization is also invariant under these replacements, the only possible non-invariance comes from the leg factors $\mathcal{N}_b(p)$ given in (2.20a). We obtain

$$\mathsf{A}_{g,n}^{(-b)}(\boldsymbol{p}) = \prod_{j=1}^{n} \frac{\mathcal{N}_{-b}(p_j)}{\mathcal{N}_b(p_j)} \mathsf{A}_{g,n}^{(b)}(\boldsymbol{p}) = \mathsf{A}_{g,n}^{(b)}(\boldsymbol{p}) \,, \tag{2.24a}$$

$$\mathsf{A}_{g,n}^{(b^{-1})}(\boldsymbol{p}) = \prod_{j=1}^{n} \frac{\mathcal{N}_{b^{-1}}(p_j)}{\mathcal{N}_b(p_j)} \mathsf{A}_{g,n}^{(b)}(\boldsymbol{p}) = (-1)^n \mathsf{A}_{g,n}^{(b)}(\boldsymbol{p}) \,. \tag{2.24b}$$

In the second equation, we inserted the correct value of the ratio of the leg factors, (2.20a). In particular, the second invariance looks very innocent from a worldsheet point of view, but it is not manifest in the proposed solutions (1.6) and will be highly non-trivial on the matrix integral side that we will discuss in [31]. We call it duality symmetry since it comes from the $b \to b^{-1}$ duality of Liouville theory.

Finally, we can exchange the two Liouville theories on the worldsheet, which amounts to replacing $b \to -ib$ and $\boldsymbol{p} \to i\boldsymbol{p}$. We obtain

$$\mathsf{A}_{g,n}^{(-ib)}(i\boldsymbol{p}) = \prod_{j=1}^{n} \frac{\mathcal{N}_{-ib}(ip_j)}{\mathcal{N}_b(p_j)} \mathsf{A}_{g,n}^{(b)}(\boldsymbol{p}) = (-i)^n \mathsf{A}_{g,n}^{(b)}(\boldsymbol{p}) \,. \tag{2.25}$$

We call this symmetry swap symmetry, since it comes from exchanging the two theories.

**Trivial zeros.** By definition, the leg factors $\mathcal{N}_b(p)$ have a simple zero when $p = \frac{mb}{2}$ or $p = \frac{m}{2b}$ for $m \in \mathbb{Z}$. Thus we have

$$\mathsf{A}_{g,n}^{(b)}(p_1 = \tfrac{mb}{2}, p_2, \ldots, p_n) = \mathsf{A}_{g,n}^{(b)}(p_1 = \tfrac{m}{2b}, p_2, \ldots, p_n) = 0 \,, \qquad m \in \mathbb{Z} \,. \tag{2.26}$$

We call these zeros the trivial zeros. The trivial zeros are actually manifest in our claimed formulas (1.6c) and (1.6b) for the four-point function and the one-point function on the torus, at least those for $p_j = \frac{m}{2b}$. The other series of trivial zeros follows by swap symmetry which however is obscured in that representation.

## 2.4 Three-point function

We will now explicitly evaluate the three-point function of the theory. We show four equivalent formulas, each of which is useful to exhibit different properties of the answer.

The three-point function is one of the simplest observables in the theory, as it requires no integration over moduli space. Hence it is simply given up to leg pole factors and overall normalization by the product of DOZZ structure constants in the partner Liouville CFTs

$$\mathsf{A}^{(b)}_{0,3}(p_1, p_2, p_3) = C^{(b)}_{\mathrm{S}^2} \left( \prod_{j=1}^{3} \mathcal{N}_b(p_j) \right) C_b(p_1, p_2, p_3) C_{-ib}(ip_1, ip_2, ip_3) \,. \tag{2.27}$$

From this expression it is already clear that the three-point function has a somewhat interesting analytic structure. In particular, the DOZZ structure constant $C_b$ exhibits infinitely many simple poles at the following values of its arguments

$$C_b(p_1, p_2, p_3): \text{ poles at } \pm p_1 \pm p_2 \pm p_3 = \left( m + \frac{1}{2} \right) b + \left( n + \frac{1}{2} \right) b^{-1}, \quad m, n \in \mathbb{Z}_{\geqslant 0}\,. \tag{2.28}$$

Combining with the poles of the partner DOZZ structure constant we learn that the three-point function is characterized by poles and zeros at

$$\mathsf{A}^{(b)}_{0,3}(p_1, p_2, p_3) : \text{ poles at } p_1 \pm p_2 \pm p_3 = \left( r + \frac{1}{2} \right) b + \left( s + \frac{1}{2} \right) b^{-1}, \quad r, s \in \mathbb{Z}$$

$$\mathsf{A}^{(b)}_{0,3}(p_1, p_2, p_3) : \text{ zeros at } 2p_j = rb + sb^{-1}\,, \quad r, s \in \mathbb{Z}\,, \tag{2.29}$$

where $2p_j = 0$ is a single zero. In fact, as previously noted by [24], this combination of DOZZ structure constants simplifies drastically. Indeed, making use of the following shift identities satisfied by the Barnes double gamma function $\Gamma_b$ that appears in the definition of the structure constant (2.15)

$$\Gamma_b(z+b) = \frac{\sqrt{2\pi} b^{zb-\frac{1}{2}}}{\Gamma(bz)} \Gamma_b(z), \quad \Gamma_b(z+b^{-1}) = \frac{\sqrt{2\pi} b^{-zb^{-1}+\frac{1}{2}}}{\Gamma(b^{-1}z)} \Gamma_b(z)\,, \tag{2.30}$$

one can show that the product of structure constants together with the leg factors that appears is a doubly-periodic function of the Liouville momenta $p_i$.

**Theta-function representation.** The result is that the three-point function can be written simply as follows

$$\mathsf{A}^{(b)}_{0,3}(p_1, p_2, p_3) = \frac{ib\eta(b^2)^3 \prod_{j=1}^{3} \vartheta_1(2bp_j|b^2)}{2\vartheta_3(bp_1 \pm bp_2 \pm bp_3|b^2)}\,. \tag{2.31}$$

Here $\vartheta_i$ are the Jacobi theta functions, $\eta$ is the Dedekind eta function and the $\pm$ signs in the argument of the theta functions are meant to denote a product over all possible sign combinations. We see that the three-point function is an elliptic function, with

$$\tau := b^2 \in \mathbb{H}^2 \tag{2.32}$$

playing the role of the torus modular parameter.[5] To work out the prefactor in (2.31) requires a bit more work and can be done with the help of the identity [24]

$$\Gamma_b\big(\tfrac{b+b^{-1}}{2} \pm z\big)\Gamma_{-ib}\big(\tfrac{-ib+(-ib)^{-1}}{2} \pm iz\big) = \frac{\mathrm{e}^{-\frac{\pi i z^2}{2}}\vartheta_3(0|b^2)}{\vartheta_3(bz|b^2)} . \tag{2.33}$$

This representation (2.31) exhibits the poles (2.29) via the zeros of the theta functions. This representation also makes the three symmetries (2.24a), (2.24b) and (2.25) essentially manifest. The combination of swapping and duality preserves $b^2 \in \mathbb{H}^2$ and acts as a modular S-transformation. Indeed, making use of the transformation properties of the theta functions under modular S-transform we see that

$$\mathsf{A}_{0,3}^{(\frac{i}{b})}(ip_1, ip_2, ip_3) = -i\,\mathsf{A}_{0,3}^{(b)}(p_1, p_2, p_3)\,, \tag{2.34}$$

which is the combination of (2.24b) and (2.25).

**Infinite sum representation.** Making use of the infinite product representation of the theta functions, one can show that the three-point function (2.31) can be written in terms of the following infinite sum

$$\mathsf{A}_{0,3}^{(b)}(p_1, p_2, p_3) = \sum_{m=1}^{\infty} \frac{2b(-1)^m \sin(2\pi mbp_1)\sin(2\pi mbp_2)\sin(2\pi mbp_3)}{\sin(\pi mb^2)} . \tag{2.35}$$

In writing this expression we have assumed that $|\operatorname{Im}(2b(p_1 \pm p_2 \pm p_3))| < \operatorname{Im}(b^2)$ so that the sum converges. It in particular converges in the main case of interest in this paper for which $b^2 \in i\mathbb{R}$, $bp_i \in \mathbb{R}$. This is the form that we introduced in the introduction (1.6a) and which will most directly generalize to other $(g, n)$.

This infinite sum can be obtained from the following contour integral

$$\mathsf{A}_{0,3}^{(b)}(p_1, p_2, p_3) = -2i \int_\gamma \mathrm{d}p\; \frac{\sin(4\pi pp_1)\sin(4\pi pp_2)\sin(4\pi pp_3)}{\sin(2\pi bp)\sin(2\pi b^{-1}p)} . \tag{2.36}$$

Here the contour $\gamma$ runs around the ray of poles $p = \tfrac{1}{2}mb$, $m \in \mathbb{Z}_{>0}$ in the first quadrant as shown in figure 1. However the contour can be made to run around any of the four rays of poles at $p = \pm\tfrac{1}{2}mb^{\pm 1}$ using oddness of the integrand under $p \to -p$ and the vanishing of the residue at $p = 0$. This makes the three symmetries (2.24a), (2.24b) and (2.25) manifest.

---

[5]We note in passing that, in a particular parameterization, the right-hand side of (2.31) has also recently appeared as the boundary two-point function in double-scaled SYK [48–50]. In [50] it was interpreted in terms of the two-point function in two copies of Liouville CFT with central charges adding up to 26 subject to FZZT conformal boundary conditions.

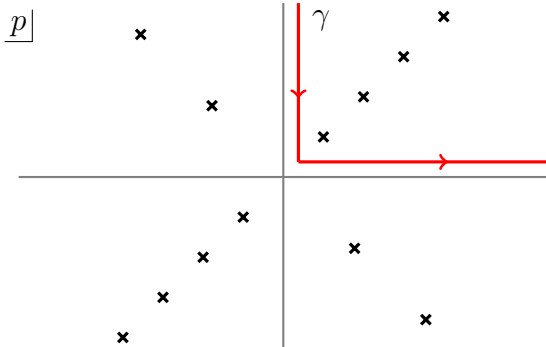

**Figure 1**: The contour $\gamma$ in the integral representation (2.36) of the three-point amplitude $\mathsf{A}_{0,3}^{(b)}(p_1, p_2, p_3)$ runs around the ray of poles located at $p = mb/2$, $m \in \mathbb{Z}_{>0}$. Deforming the contour such that it picks up the residues from this set of poles results in the infinite sum representation (2.35) of the three-point amplitude.

**Second infinite sum representation.** Another useful expression is obtained by expanding the factor of $1/\sin(\pi m b^2)$ in (2.35) as a geometric series and performing the sum over $m$. This leads to

$$\mathsf{A}_{0,3}^{(b)}(p_1, p_2, p_3) = \frac{b}{2} \sum_{k=0}^{\infty} \sum_{\sigma_1, \sigma_2, \sigma_3 = \pm} \frac{\sigma_1 \sigma_2 \sigma_3}{1 + e^{2\pi i b(\sigma_1 p_1 + \sigma_2 p_2 + \sigma_3 p_3 + b(k+\frac{1}{2}))}} \, . \tag{2.37}$$

This expression has the benefit of making the poles (2.29) manifest and converges everywhere in the complex plane.

**Double infinite sum representation.** Finally, we note that the three-point amplitude can be written in terms of the following sum over simple poles

$$\mathsf{A}_{0,3}^{(b)}(p_1, p_2, p_3) = \frac{i}{8\pi} \sum_{m,\, n \in \mathbb{Z}+\frac{1}{2}} \sum_{\sigma_1, \sigma_2, \sigma_3 = \pm} \frac{\sigma_1 \sigma_2 \sigma_3}{\sigma_1 p_1 + \sigma_2 p_2 + \sigma_3 p_3 - mb - nb^{-1}} \, . \tag{2.38}$$

After summing over the three signs, the sum over $m$ and $n$ is absolutely convergent. This expression is manifestly doubly-periodic, odd under reflection of the momenta, and exhibits the poles (2.29). Since it is an elliptic function with the correct poles and residues it can only differ from the exact answer by a constant, which would conflict with oddness of the amplitude under reflection of the momenta. This shows that the sphere three-point amplitude $\mathsf{A}_{0,3}^{(b)}$ is in some sense the simplest possible function consistent with the poles (2.29).

## 2.5 Two-point function

**Dividing by the Möbius group.** Another very important ingredient is the two-point function $\mathsf{A}_{0,2}^{(b)}(p_1, p_2)$. As usual in string theory, this is a bit of a subtle object since there is an infinite Möbius group that compensates for the fact that the world-sheet two-point function has two delta-functions. The prescription how to deal with

this was explained in [51] in the context of $\mathrm{AdS}_3$ string theory and in [52] for flat space amplitudes, but is more general. The correct answer is to cancel one delta-function in worldsheet conformal weight space with the infinite volume of the Möbius group (up to a constant factor, whose value is more subtle to determine). Thus the two-point function is

$$
\begin{aligned}
\mathsf{A}_{0,2}^{(b)}(p_1, p_2) &= \frac{C_{\mathrm{S}^2}^{(b)}}{\mathrm{vol}} \frac{\mathcal{N}_b(p_1)\mathcal{N}_b(p_2)\,\delta(p_1 - p_2)^2}{\rho_{b^+}(p_1^+)\rho_{b^-}(p_1^-)} \\
&\sim \frac{\delta(p_1 - p_2)^2}{p_1^2\,\mathrm{vol}} \\
&\sim \frac{1}{p_1}\delta(p_1 - p_2)\ ,
\end{aligned}
\tag{2.39}
$$

where we suppressed numerical constants and used that the conformal weight is quadratic in the Liouville momenta.

**Integration measure.** This means that the natural measure in *spacetime* for the integration over a complete set of physical string states is $-2p\,\mathrm{d}p$.[6] This is the same measure as in the Virasoro minimal string [13]. The minus sign appears because we are considering lowercase $p$. In other words, if we would consider the theory as a gauge theory analogously to treating JT-gravity in the $\mathrm{SL}(2,\mathbb{R})\ BF$ theory formalism, we would have

$$
\mathsf{A}_{0,4}^{(b)}(p_1, p_2, p_3, p_4) \overset{?}{\sim} \int (-2p\,\mathrm{d}p)\ \mathsf{A}_{0,3}^{(b)}(p_1, p_2, p)\mathsf{A}_{0,3}^{(b)}(p_3, p_4, p)\ ,
\tag{2.40}
$$

since one would simply insert a complete set of states at the neck of the pair of pants decomposition. However, in a theory of quantum gravity, this formula is incorrect since it does not take the gauging of large diffeomorphisms on the worldsheet (mapping class group) into account. Indeed, since $\mathsf{A}_{0,3}^{(b)}(p_1, p_2, p_3)$ is a periodic function, the integral over $p$ diverges. However, this measure still plays an important role just like in JT-gravity, for example for the gluing of asymptotic trumpets. In fact, we will see in [31] that there is a suitable version of this formula that does hold in the full worldsheet theory.

# 3   Properties of $\mathbf{A}_{g,n}^{(b)}$

Our goal will be to determine the string amplitudes $\mathsf{A}_{g,n}^{(b)}$ explicitly. It is not possible with current technology to analytically compute them from their definition (2.19), since there are no analytically tractable expressions for the conformal blocks, etc.

---

[6]The normalization is not clear at this point, but will follow once we have analyzed the four-point function in detail. We chose our conventions so that this coincides with $2P\,\mathrm{d}P$ with $P = ip$, which is the measure that appears in [13].

Instead, our strategy will be to derive various properties that they satisfy and from there show that under mild assumptions these fix the result uniquely. We will also numerically check below that our solution (1.6c) is indeed correct.

## 3.1 Analytic structure

The string amplitudes $\mathsf{A}_{g,n}^{(b)}(p_1, \ldots, p_n)$ as defined by the integral of Liouville CFT correlators over the moduli space of Riemann surfaces converge in the main case of interest in this paper ($b^2 \in i\mathbb{R}$, $bp_i \in \mathbb{R}$). However the condition on the central charge may be relaxed (provided that $b^2 \in \mathbb{H}^2$) and the string amplitudes may naturally be extended to analytic functions of the momenta $p_1, \ldots, p_n$ as explained in section 2.2. We now explain the analytic structure of the analytic continuation of $\mathsf{A}_{g,n}^{(b)}(p_1, \ldots, p_n)$. The function is characterized by an infinite set of poles in addition to infinitely many branch cuts that arise because the integral over moduli space can cease to converge as the momenta are varied away from the physical spectrum. We will explain the example of the sphere four-point amplitude $\mathsf{A}_{0,4}^{(b)}(p_1, \ldots, p_4)$ defined in (2.19) in detail here in order to illustrate the mechanism. The general case proceeds similarly and we will mention the general results at the end. These analyticity properties place powerful constraints on the string amplitudes, and indeed we will see that subject to some mild assumptions one may bootstrap the string amplitudes from them, see section 5.6.

**Discrete contributions.** The correlation functions in the worldsheet Liouville CFTs are computed by the conformal block expansion; the example of the sphere four-point function is given in (2.17). For more general amplitudes $\mathsf{A}_{g,n}^{(b)}$, one performs a pair of pants decomposition of the surface $\Sigma_{g,n}$ and inserts a complete set of states on the internal cuffs. What remains is a product of CFT structure constants multiplying the conformal blocks corresponding to the particular pair of pants decomposition integrated over the complete sets of states running along the internal cuffs. We can take the Liouville momenta $p$ of these internal states to be integrated along the vertical contour $p \in i\mathbb{R}$.[7] In Liouville CFT the structure constants are given by the DOZZ formula $C_b(p_1, p_2, p_3)$ (2.15), which is characterized by the lattices of simple poles specified in (2.28). For sufficiently small momenta $bp_i$ these poles are well-separated from the contour of integration in the conformal block expansion. However, as one varies the external Liouville momenta the poles of the DOZZ formula in the internal Liouville momenta may cross the contours of integration, leading to additional discrete contributions to the conformal block decomposition associated with the residues of the poles.

For concreteness we discuss the sphere four-point function in Liouville CFT that appears in the sphere four-point amplitude (2.19). Consider in particular the family

---

[7]We could rotate the contour to the 45-degree line as described around (2.18), but the discussion that follows is simpler with the vertical contour.

of poles in the internal Liouville momentum $p$ at[8]

$$p = p_* = \pm p_1 \pm p_2 + \left(m + \frac{1}{2}\right)b + \left(n + \frac{1}{2}\right)b^{-1}, \quad m, n \in \mathbb{Z}_{\geqslant 0}. \tag{3.1}$$

There is a similar family of poles for any pair of external momenta $p_i$ and $p_j$. These poles extend infinitely to the right in the complex $p$ plane. However, as we vary $p_1$ and $p_2$ it may be that some of these poles cross the contour of integration. This happens whenever

$$\mathrm{Re}(p_*) < 0. \tag{3.2}$$

Deforming the contour of integration around this pole leads to a contribution of $-2\pi i$ times the residue of the integrand of the conformal block expansion, as shown in figure 2a:[9]

$$\langle V_{p_1}^+(0) V_{p_2}^+(z, \bar{z}) V_{p_3}^+(1) V_{p_4}^+(\infty) \rangle$$
$$\supset -2\pi \sum_{\substack{\mathrm{Re}(p_*)<0}} \mathrm{Res}_{p=p_*} \left( \rho_b(p) C_b(p_1, p_2, p) C_b(p, p_3, p_4) \mathcal{F}_{0,4}^{(b)}(\boldsymbol{p}; p|z) \mathcal{F}_{0,4}^{(b)}(\boldsymbol{p}; p|\bar{z}) \right). \tag{3.3}$$

Combining with the partner Liouville correlator, all together the sphere four-point amplitude receives discrete contributions whenever any of the following holds

$$\mathrm{Re}\left(\pm p_i \pm p_j + \left(m + \frac{1}{2}\right)b + \left(n + \frac{1}{2}\right)b^{-1}\right) < 0, \tag{3.4a}$$

$$\mathrm{Re}\left(\pm p_i \pm p_j - \left(m + \frac{1}{2}\right)b - \left(n + \frac{1}{2}\right)b^{-1}\right) > 0, \tag{3.4b}$$

$$\mathrm{Im}\left(\pm p_i \pm p_j - \left(m + \frac{1}{2}\right)b + \left(n + \frac{1}{2}\right)b^{-1}\right) > 0, \tag{3.4c}$$

$$\mathrm{Im}\left(\pm p_i \pm p_j + \left(m + \frac{1}{2}\right)b - \left(n + \frac{1}{2}\right)b^{-1}\right) < 0, \quad m, n \in \mathbb{Z}_{\geqslant 0}, \tag{3.4d}$$

for any pair of external momenta $p_i, p_j$.

**Poles.** The singularities of the DOZZ structure constants can also generate poles of the Liouville worldsheet correlation functions and hence of the string amplitudes when distinct poles of the structure constants pinch the contour of integration in the conformal block decomposition as shown in figure 2b. In the case of the sphere four-point function, the resulting poles of the Liouville correlation function are located at

$$\pm p_1 \pm p_2 \pm p_3 \pm p_4 = (m+1)b + (n+1)b^{-1}, \quad m, n \in \mathbb{Z}_{\geqslant 0}, \tag{3.5}$$

---

[8]Here the two $\pm$ signs are not correlated; they correspond to four families of poles.

[9]We assume here that the integral over the internal momentum is as in (2.17). There is a second pole at $p = -p_*$ and either $p_*$ or $-p_*$ crosses the contour. Since both give identical contributions, we will continue with the contribution from the pole at $p_*$.

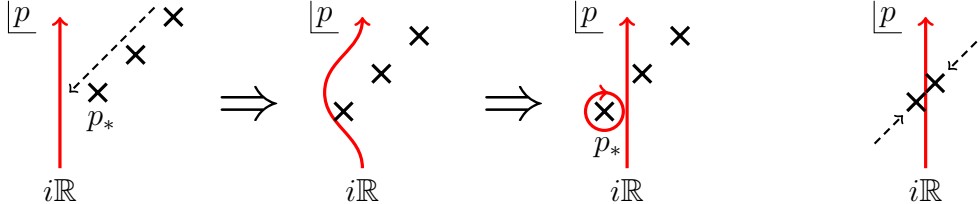

(a) Poles crossing the contour.  (b) Poles pinching the contour.

**Figure 2**: As the external Liouville momenta $p_i$ are analytically continued, it may happen that poles of the conformal block expansion cross the contour of integration over internal Liouville momenta as in figure 2a. When this happens the Liouville correlation function picks up additional discrete contributions associated with the residues of the poles that have crossed the OPE contour as in equation (3.3). It may also happen that singularities of the conformal block expansion pinch the contour of integration as in figure 2b, leading to poles of the full Liouville correlation function.

for all choices of signs on the left-hand side. These poles are well-known to arise in the Coulomb gas formalism and are associated with saturation of the Liouville background charge $Q$, see e.g. [53, 54]. Combining with the poles of the partner Liouville correlator, this leads to poles in the sphere four-point amplitude $\mathsf{A}_{0,4}^{(b)}(p_1, p_2, p_3, p_4)$ when

$$p_1 \pm p_2 \pm p_3 \pm p_4 = rb + sb^{-1}, \quad r, s \in \mathbb{Z}_{\neq 0}, \tag{3.6}$$

for all choices of signs.

This discussion straightforwardly generalizes to more complicated observables. Again the poles are due to singularities of the DOZZ structure constants pinching the contours of integration of intermediate Liouville momenta. The Liouville CFT $n$-point function on a genus-$g$ Riemann surface exhibits poles when viewed as a function of the external momenta for

$$\pm p_1 \pm p_2 \pm \cdots \pm p_n = (2g - 2 + n)\frac{Q}{2} + rb + sb^{-1}, \tag{3.7}$$

with $r, s \in \mathbb{Z}_{\geqslant 0}$ for any choices of signs. Combining with the poles of the partner Liouville correlator, this leads to poles in the genus-$g$ $n$-point string amplitude $\mathsf{A}_{g,n}^{(b)}(p_1, \ldots, p_n)$ for

$$p_1 \pm p_2 \pm \cdots \pm p_n = rb + sb^{-1}, \qquad r, s \in \mathbb{Z} + \frac{n}{2}, \qquad |r|, |s| \geqslant \frac{2g - 2 + n}{2}. \tag{3.8}$$

**Discontinuities.** The Liouville correlators defined with the extra discrete contributions associated with poles of the DOZZ structure constants crossing the OPE contour are perfectly sensible, but they do not necessarily give convergent contributions to the string amplitude when combined with the other Liouville correlator and integrated over moduli space. In the case that they lead to divergent contributions, we must define the amplitude via analytic continuation from a region where

the integral converges. We will see that this generates branch cuts in the string amplitudes.

Consider the discrete contributions to the four-point function enumerated in (3.3) associated with the poles (3.1) crossing the contour of integration in the conformal block expansion. In particular, the poles $p_*$ for which in addition to (3.2)

$$\text{Re}(p_*^2) > 0 \tag{3.9}$$

contribute residues to (3.3) that diverge when integrated over cross-ratio space due to singular behaviour in the degenerating locus $z \to 0$. Let us henceforth consider the contribution of a single discrete pole at $p = p_*$ to the sphere four-point amplitude, starting from the regime that $\text{Re}(p_*^2) < 0$ so that the moduli integral converges. In particular, consider the isolated contribution of this discrete pole to the string amplitude associated with a unit disk $\mathsf{D}^2$ centered around the OPE limit $z = 0$[10]

$$
\begin{aligned}
\mathsf{A}_{0,4}^{(b)}(p_1, p_2, p_3, p_4) \supset &-2\pi C_{\mathsf{S}^2}^{(b)} \left( \prod_{j=1}^{4} \mathcal{N}_b(p_j) \right) \int_{\mathsf{D}^2} \mathrm{d}^2 z \Bigg[ \underset{p=p_*}{\text{Res}}\, \rho_b(p) C_b(p_1, p_2, p) C_b(p_3, p_4, p) \\
&\times \int_{i\mathbb{R}} \frac{\mathrm{d}p^-}{2i} \rho_{-ib}(p^-) C_{-ib}(ip_1, ip_2, p^-) C_{-ib}(ip_3, ip_4, p^-) \\
&\times |z|^{-2-2p_*^2 - 2(p^-)^2} (1 + \mathcal{O}(z, \bar{z})) \Bigg].
\end{aligned}
\tag{3.10}
$$

In order to discuss the analytic continuation to the regime $\text{Re}(p_*^2) > 0$, we need to commute the integral over $z$ with the integral over the internal Liouville momentum $p^-$ in the partner Liouville CFT. This is allowed since by assumption $\text{Re}(p_*^2) < 0$, and so the $z$ integral in the neighborhood of the degeneration limit converges. The integration over the neighborhood of the degenerating locus generates poles in $p^-$ at $\pm i p_*$, since

$$\int_{\mathsf{D}^2} \mathrm{d}^2 z \, |z|^{-2-2p_*^2 - 2(p^-)^2} = -\frac{\pi}{(p^-)^2 + p_*^2}. \tag{3.11}$$

Since we have so far assumed $\text{Re}(p_*) < 0$ and $\text{Re}(p_*^2) < 0$, the resulting poles in $p^-$ do not lie on the contour of integration.

This allows us to discuss the analytic continuation of $p_1$ and $p_2$ to the wedge of divergence defined by $\text{Re}(p_*) < 0$, $\text{Re}(p_*^2) > 0$. We may analytically continue into this wedge either from above or below. Importantly, the side of the $p^-$ contour that the resulting poles at $p^- = \pm i p_*$ lie on is determined by sign of the imaginary part of $p_*$; in other words, by whether we analytically continue $p_*$ into the wedge of divergence from above or below, as illustrated in figure 3. Hence there is a discontinuity in the string amplitude at $p_* = 0$ determined by the residue of the $p^-$ integral at $p^- = \pm i p_*$.

---

[10]The radius of the disk is immaterial for the purposes of this discussion.

This gives the following for the discontinuity of the sphere four-point amplitude

$$
\begin{aligned}
\operatorname*{Disc}_{p_*=0} \mathsf{A}_{0,4}^{(b)}(\boldsymbol{p}) &= -2\pi^3 i C_{\mathrm{S}^2}^{(b)} \left(\prod_{j=1}^{4} \mathcal{N}_b(p_j)\right) \frac{\rho_b(p_*)\rho_{-ib}(ip_*)}{p_*} \operatorname*{Res}_{p=p_*} \left(C_b(p_1,p_2,p)C_b(p_3,p_4,p)\right) \\
&\qquad \times C_{-ib}(ip_1,ip_2,ip_*)C_{-ib}(ip_3,ip_4,ip_*) \\
&= -2\pi^3 i \frac{\rho_b(p_*)\rho_{-ib}(ip_*)}{p_*} \frac{1}{\mathcal{N}_b(p_*)^2 C_{\mathrm{S}^2}^{(b)}} \operatorname*{Res}_{p=p_*} \left(\mathsf{A}_{0,3}^{(b)}(p_1,p_2,p)\mathsf{A}_{0,3}^{(b)}(p_3,p_4,p)\right) \\
&= 8\pi i p_* \operatorname*{Res}_{p=p_*} \left(\mathsf{A}_{0,3}^{(b)}(p_1,p_2,p)\mathsf{A}_{0,3}^{(b)}(p_3,p_4,p)\right) .
\end{aligned}
\tag{3.12}
$$

This analysis carries through for any pair of external momenta $p_i$ and $p_j$ associated with a pole $p_*$ (as in (3.4) that has crossed the OPE contour in either of the worldsheet Liouville CFT correlators and leads to a divergent contribution to the moduli integral in the appropriate OPE degeneration limit at $z = 0$, 1 or $\infty$.

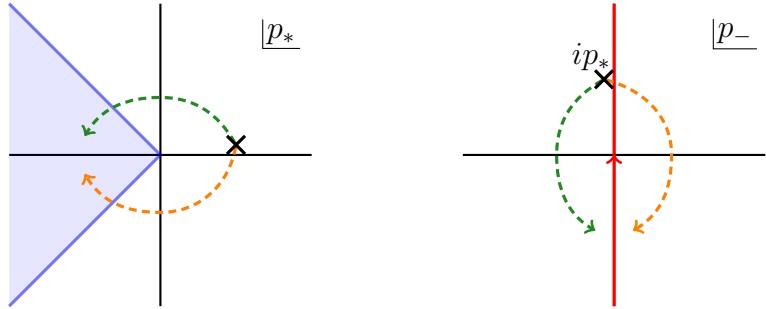

**Figure 3**: On the left, the blue region denotes the wedge of divergence $\operatorname{Re}(p_*) < 0$, $\operatorname{Re}(p_*^2) > 0$ for which the discrete contribution associated with the pole $p_*$ gives a divergent contribution to the sphere four-point amplitude $\mathsf{A}_{0,4}^{(b)}$. The analytic continuation of the string amplitude depends on whether $p_*$ is continued into the wedge of divergence from above or from below. The difference comes from a residue contribution associated with the pole at $p_- = ip_*$ crossing the integration contour of the intermediate Liouville momentum $p_-$ (depicted in the right figure). This leads to the discontinuity (3.12) of the string amplitude.

To summarize, the region of analyticity of the sphere four-point amplitude where the moduli integral converges is given by the complement of the union of the following regions, each of which carves out a series of 90 degree wedges in the $p_i \pm p_j$ plane

$$
\begin{aligned}
&\operatorname{Re}(p_*) < 0 \text{ and } \operatorname{Re}(p_*^2) > 0\,, && p_* = \pm p_i \pm p_j + rb + sb^{-1}\,, && \text{(3.13a)} \\
&\operatorname{Re}(p_*) > 0 \text{ and } \operatorname{Re}(p_*^2) > 0\,, && p_* = \pm p_i \pm p_j - rb - sb^{-1}\,, && \text{(3.13b)} \\
&\operatorname{Im}(p_*) > 0 \text{ and } \operatorname{Re}(p_*^2) < 0\,, && p_* = \pm p_i \pm p_j - rb + sb^{-1}\,, && \text{(3.13c)} \\
&\operatorname{Im}(p_*) < 0 \text{ and } \operatorname{Re}(p_*^2) < 0\,, && p_* = \pm p_i \pm p_j + rb - sb^{-1}\,, && \text{(3.13d)}
\end{aligned}
$$

where $r, s \in \mathbb{Z}_{\geqslant 0} + \frac{1}{2}$, for all choices of the signs. In the case that $b \in e^{\frac{\pi i}{4}}\mathbb{R}$, this is a non-compact region that in particular includes the lines $bp_j \in \mathbb{R}$. The reason for

this is that all of the poles with $r > \frac{1}{2}$ or $s > \frac{1}{2}$ lie within or on the boundary of the wedge associated with $r = s = \frac{1}{2}$. In the more general situation that $b^2 \in \mathbb{H}^2$ away from the positive imaginary axis this defines a compact region in the $p_i \pm p_j$ plane as shown in figure 4.

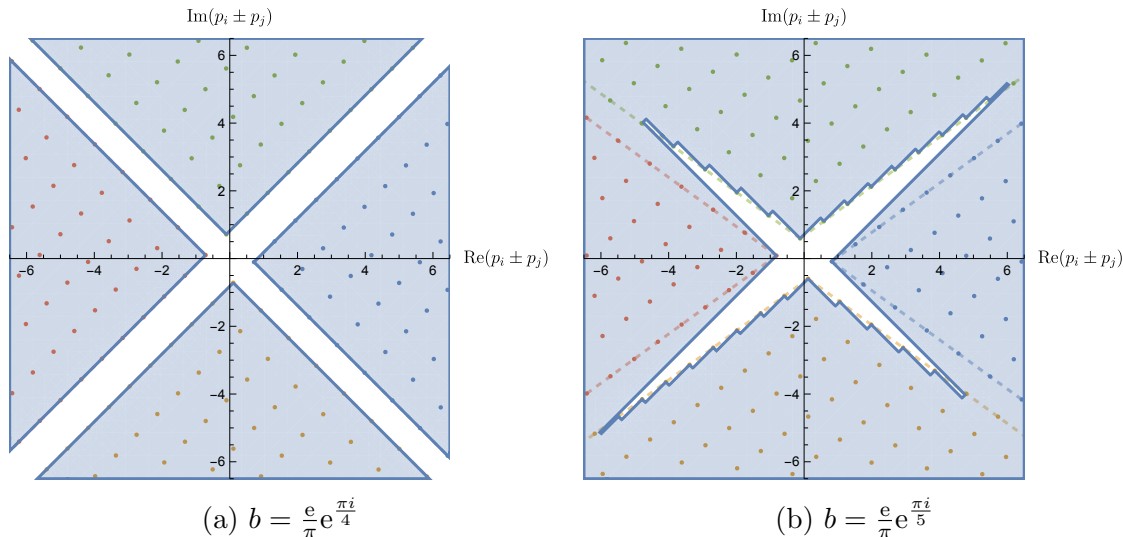

(a) $b = \frac{\mathrm{e}}{\pi}\mathrm{e}^{\frac{\pi i}{4}}$           (b) $b = \frac{\mathrm{e}}{\pi}\mathrm{e}^{\frac{\pi i}{5}}$

**Figure 4**: In each case ($b^2 \in i\mathbb{R}$ on the left, $b^2 \notin i\mathbb{R}$ on the right), the unshaded region is the region in the parameter space of the Liouville momenta where the moduli space integral converges. The four families of dots represent the poles $p_*$ of the integrand of the conformal block decomposition with respect to the internal Liouville momenta, with the left and right families associated to one Liouville correlator and the top and bottom corresponding to the other.

**Cancellation of discontinuities associated with subleading terms in the OPE.** A priori, one might have suspected the existence of infinitely many additional branch cuts associated with subleading terms in the conformal block expansion of the Liouville CFT correlators. On the other hand, these putative discontinuities would be due to the exchange of Virasoro descendants of Liouville primary operators in the OPE; the latter are not in the BRST cohomology of the worldsheet theory and hence are not expected to contribute to qualitative features of physical observables. We will spare the reader the computation, but one can explicitly verify that these terms do not contribute additional discontinuities due to a cancellation originating from structural properties of the conformal block expansion.

**General formula.** We have seen in the example of the sphere four-point amplitude that the discontinuity is given by the residue of the product of sphere three-point string amplitudes associated with the particular degeneration of the surface that gives a divergent contribution to the moduli integral. This is a feature that generalizes straightforwardly to more complicated observables; indeed one can similarly

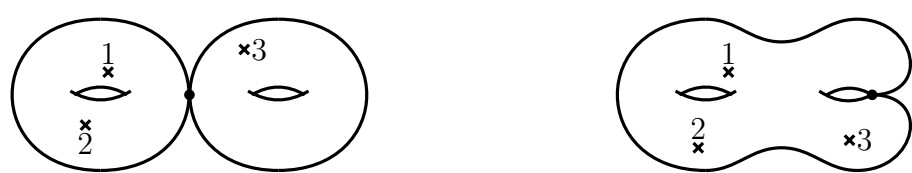

**Figure 5**: Left: The separating divisor $\mathcal{D}_{1,\{1,2\}} = \mathcal{D}_{1,\{3\}}$, right: the non-separating divisor $\mathcal{D}_{\mathrm{irr}}$.

compute the discontinuity of $\mathsf{A}_{g,n}^{(b)}$ in terms of the residue of lower-point functions. The discontinuities originate from the boundary divisors of moduli space. They come in two types: separating divisors which are labelled by $\mathcal{D}_{h,I}$ for $0 \leqslant h \leqslant g$ and $I \subset \{1,2,\ldots,n\}$ and divide the surface into two parts[11] as in figure 5 and the non-separating divisor $\mathcal{D}_{\mathrm{irr}}$ that yields a surface of lower genus with two nodal points. The role of $z$ in the above argument is played by a local coordinate $q$ in moduli space that describes the normal direction to the boundary divisor. We obtain

$$\underset{p_*=0}{\mathrm{Disc}}\,\mathsf{A}_{g,n}^{(b)}(\boldsymbol{p}) = 2\pi i\left[\underset{p=p_*}{\mathrm{Res}}\sum_{\substack{h=0,\ldots,g \\ I\subset\{1,2,\ldots,n\} \\ \text{stable}}} 2p\,\mathsf{A}_{h,|I|+1}^{(b)}(\boldsymbol{p}_I,p)\mathsf{A}_{h,|I^c|+1}^{(b)}(\boldsymbol{p}_{I^c},p)\right.$$

$$\left.+ \underset{p=\frac{1}{2}p_*}{\mathrm{Res}}\,2p\,\mathsf{A}_{g-1,n+2}^{(b)}(\boldsymbol{p},p,p)\right] . \quad (3.14)$$

There is an extra factor of $\frac{1}{2}$ with respect to (3.12) since we are overcounting all divisors by a factor of 2 on the right-hand side (and the divisor $\mathcal{D}_{\mathrm{irr}}$ has an automorphism of order 2 by which we need to divide). The appearing $p_*$ is such that the residue on the right-hand side is non-vanishing and hence runs over all poles of the simpler amplitudes as given in (3.8). For the non-separating divisor, the pole is located at $\frac{1}{2}p_*$ because $\mathsf{A}_{g-1,n+2}^{(b)}(\boldsymbol{p},p,p)$ has two $p$-insertions.

There are in general also sequential discontinuities originating from higher codimension boundaries in moduli space, but we will not work them out since they can be obtained by recursively applying (3.14).

## 3.2   Dilaton equation

We will next derive a general property that is obeyed by any $\mathsf{A}_{g,n}^{(b)}(p_1,\ldots,p_n)$ and hence we will explain the general derivation. We consider $\mathsf{A}_{g,n+1}^{(b)}(p_1,\ldots,p_{n+1})$ and consider the analytic continuation in $p_{n+1}$. We will then take the limit

$$p_{n+1} \to \frac{1}{2}\hat{Q} = \frac{b^{-1}-b}{2} . \quad (3.15)$$

---

[11]There is also a stability condition that requires $|I| \geqslant 2$ when $h = 0$ and $|I^c| \geqslant 2$ when $h = g$.

This corresponds to the vertex operator built out of the marginal operator in one of the Liouville CFTs and the identity in the other. We claim that

$$\lim_{p_{n+1} \to \frac{1}{2}\hat{Q}} \mathsf{A}^{(b)}_{g,n+1}(p_1, \ldots, p_{n+1}) = -\left( \tfrac{1}{2}Q(2g-2+n) - \sum_{j=1}^{n} \sqrt{p_j^2} \right) \mathsf{A}^{(b)}_{g,n}(p_1, \ldots, p_n) . \quad (3.16)$$

We refer to this equation as the dilaton equation. There is a similar equation when we specify $p_{n+1} \to \frac{1}{2}Q$, which is obtained by the swap symmetry (2.25). We recall that the spectrum was such that $bp_j \in \mathbb{R}$. The appearance of the function $\sqrt{p_j^2}$ requires some explanation. For (3.16) to be rigorously true, we need to assume that $\mathrm{Re}(p_j^2) > 0$ since the moduli space integral is only convergent in that case. In this case, the choice of the branch of the square root is unambiguous and is given by the principal branch. However, by analytic continuation, we can extend the region of validity and we write the answer in the form $\sqrt{p_j^2}$ without precisely specifying where the branch cut is located.[12] The case of $g = 0$, $n = 4$ is also a special case of the higher equations of motion that we discuss in section 4.3. We will now explain a different derivation of (3.16) than the derivation using the ground ring operators that will be introduced in section 4.1.

**Reduction to the Liouville dilaton equation.** For $p_{n+1} = \frac{\hat{Q}}{2}$, the operator in the second Liouville theory becomes the identity. This means that the integral over the location of the $(n+1)^{\text{st}}$ vertex operator is only dependent on the first Liouville theory and we can consider independently the integral

$$\int \mathrm{d}^2 z_{n+1} \ \langle V^+_{p_1}(z_1) \cdots V^+_{p_n}(z_n) V^+_{\frac{1}{2}\hat{Q}}(z_{n+1}) \rangle \quad (3.18)$$

over the last vertex operator of the first Liouville theory. This is the marginal operator of Liouville theory.

**Path integral argument.** There is a simple path integral argument to evaluate (3.18). The point is that insertions of $V^+_{\frac{1}{2}\hat{Q}}$ act as $\mu$-derivatives, where $\mu$ is the cosmological constant of Liouville theory appearing in the action (2.1). The KPZ scaling argument [55] shows that the $\mu$-dependence of the correlator takes the universal form $\mu^{-\frac{1}{b}(\sum_i p_i + \frac{Q}{2}(2g-2+n))}$. Thus we have

$$\int \mathrm{d}^2 z_{n+1} \ \langle V^+_{p_1}(z_1) \cdots V^+_{p_n}(z_n) V^+_{\frac{1}{2}\hat{Q}}(z_{n+1}) \rangle$$

---

[12]In literature about the minimal string [39], the function $|x|_{\mathrm{Re}}$ is used instead of $\sqrt{x^2}$. It is defined as

$$|x|_{\mathrm{Re}} = \begin{cases} x , & \mathrm{Re}(x) > 0 , \\ -x , & \mathrm{Re}(x) < 0 . \end{cases} \quad (3.17)$$

For $\mathrm{Re}(x^2) > 0$ it agrees with $\sqrt{x^2}$, but we prefer to write $\sqrt{x^2}$ in the following since the imaginary axis is not a natural choice of branch cut for this function.

$$\propto \frac{\partial}{\partial \mu} \langle V_{p_1}^+(z_1) \cdots V_{p_n}^+(z_n) \rangle$$

$$\propto \left( \sum_i p_i + \frac{Q}{2}(2g - 2 + n) \right) \langle V_{p_1}^+(z_1) \cdots V_{p_n}^+(z_n) \rangle . \tag{3.19}$$

The case distinction for $p_j$ in (3.17) appears because the path integral derivation is only valid for $\text{Re}(p_j) < 0$, which is the Seiberg bound [56]. The other case can be inferred by reflection symmetry under $p_j \to -p_j$. This demonstrates (3.16) up to a constant hidden in the normalization of the path integral.

**Normalization.** Thus it only remains to work out the constant appearing on the RHS of (3.16). We do this by carefully computing the non-analytic terms in (3.16). We already know that the analytic terms will then follow by the path integral argument. The non-analytic terms can be worked out similarly to what we explained in section 3.1. They come from the limit where $z_{n+1}$ collides with one of the other vertex operators. We can look at the OPE, which takes the form

$$V_{\frac{1}{2}\hat{Q}}^+(z) V_p^+(0) \sim i \int_0^{i\infty} \mathrm{d}p' \, \rho_0(p) \, C_b(\tfrac{1}{2}\hat{Q}, p, p') |z|^{2(p^2 - (p')^2 - 1)} V_{p'}^+(0) . \tag{3.20}$$

The factor of $i$ comes from the Jacobian to imaginary $p'$ and an additional minus sign from the convention of $\rho_0(p)$. We can now integrate over a the unit disk in $z$ (or any vicinity of the origin), which gives

$$\int \mathrm{d}^2 z \, V_{\frac{1}{2}\hat{Q}}^+(z) V_p^+(0) \sim \pi i \int_0^{i\infty} \mathrm{d}p' \, \frac{\rho_0(p') \, C_b(\tfrac{1}{2}\hat{Q}, p, p')}{p^2 - (p')^2} V_{p'}^+(0) . \tag{3.21}$$

The non-analytic term comes from the pole at $p' = \pm p$ crossing the contour, which will pick up the residue at $p' = p$. We may hence put everywhere $p' = p$ except in the denominator and compute for the non-analytic piece including the leg factor

$$\mathcal{N}_b(\tfrac{1}{2}\hat{Q}) \int \mathrm{d}^2 z \, V_{\frac{1}{2}\hat{Q}}^+(z) V_p^+(0) \sim \pi i \, \mathcal{N}_b(\tfrac{1}{2}\hat{Q}) \int_0^{i\infty} \mathrm{d}p' \, \frac{\rho_0(p) \, C_b(\tfrac{1}{2}\hat{Q}, p, p)}{p^2 - (p')^2} V_p^+(0)$$

$$= \sqrt{p^2} . \tag{3.22}$$

This confirms the normalization of (3.16).

## 3.3 Triality symmetry

We will now discuss an additional special symmetry of $\mathsf{A}_{0,4}^{(b)}(p_1, p_2, p_3, p_4)$ that does not directly generalize to higher $g$ and $n$.

The Liouville four-point function enjoys an additional symmetry. To make it manifest, one has to consider a different normalization of the vertex operators. Consider

$$\mathsf{A}_{0,4}^{(b)\prime}(p_1, p_2, p_3, p_4) = \prod_{j=1}^4 \frac{\mathrm{e}^{-2\pi i p_j^2}}{\vartheta_1(2b p_j, b^2)} \mathsf{A}_{0,4}^{(b)}(p_1, p_2, p_3, p_4) , \tag{3.23}$$

which would simply come from a normalization of the structure constants of the form

$$C_b'(p_1, p_2, p_3) = \Gamma_b(\tfrac{Q}{2} \pm p_1 \pm p_2 \pm p_3) \,, \tag{3.24}$$

i.e. only the numerator of (2.15) and a trivial leg factor (2.20a). $\mathsf{A}_{0,4}^{(b)\prime}(p_1, p_2, p_3, p_4)$ satisfies

$$\mathsf{A}_{0,4}^{(b)\prime}(p_1, p_2, p_3, p_4) = \mathsf{A}_{0,4}^{(b)\prime}(p_1 - \tfrac{1}{2}p_{1234}, p_2 - \tfrac{1}{2}p_{1234}, p_3 - \tfrac{1}{2}p_{1234}, p_4 - \tfrac{1}{2}p_{1234}) \,, \tag{3.25}$$

where $p_{1234} = \sum_j p_j$. This is precisely the action of the SO(8) triality symmetry on the four weights of an SO(8) representation. This property has a physical origin in the AGT correspondence [57]. The Liouville four-point function in this normalization can be mapped to the partition function of $\mathcal{N} = 2$ SYM with four hypermultiplets on the squashed four-sphere $\mathrm{S}_b^4$. This theory is well-known to have SO(8) flavour symmetry, and S-duality acts by outer automorphisms on the matter fields [58]. S-duality corresponds to the Möbius transformation $z \to 1 - z$ and $z \to \frac{1}{z}$ on the level of the moduli, which can be absorbed into a redefinition of the integration variable $z$. Hence $\mathsf{A}_{0,4}^{(b)\prime}(p_1, p_2, p_3, p_4)$ will be invariant.

One can also give a direct derivation of this fact in 2d CFT, but it is somewhat non-trivial. We refer to [59, 60] for further details.

## 3.4 Relation between $\mathbf{A}_{0,4}^{(b)}$ and $\mathbf{A}_{1,1}^{(b)}$

The moduli space $\mathcal{M}_{0,4}$ can be realized as a 12-fold cover of the moduli space $\mathcal{M}_{1,1}$. This follows by noting that every torus admits a two-fold covering map to the sphere that is branched over four points, which is known as the pillow map in the physics literature [61].

For a special choice of the four momenta, one can also relate the integrands of $\mathsf{A}_{1,1}^{(b)}$ and $\mathsf{A}_{0,4}^{(b)}$. The value of $b$ has to be modified because of the double covering. Using known relations of the conformal blocks and of the structure constants, we derive the relation

$$\mathsf{A}_{1,1}^{(b)}(p_1) = \frac{\vartheta_4(bp_1|b^2)}{12 b \vartheta_3(0|b^2)\vartheta_4(0|b^2)\vartheta_3(bp_1|b^2)} \mathsf{A}_{0,4}^{(\sqrt{2}b)}\left(\tfrac{p_1}{\sqrt{2}}, \tfrac{b}{2\sqrt{2}}, \tfrac{b}{2\sqrt{2}}, \tfrac{b}{2\sqrt{2}}\right) \,. \tag{3.26}$$

The details of this computation can be found in appendix C.1. Thus in principle $\mathsf{A}_{1,1}^{(b)}$ is determined once $\mathsf{A}_{0,4}^{(b)}$ is known.

## 3.5 $\mathbf{A}_{0,1}^{(b)}$, $\mathbf{A}_{0,0}^{(b)}$ and $\mathbf{A}_{1,0}^{(b)}$

Computing low-point functions is subtle because of the residual Möbius group on the worldsheet.

**The two-point function revisited.** The two-point function was already discussed in section 2.5. It is the inverse of the measure $-2p\,\mathrm{d}p$, i.e. to first approximation

$$\mathsf{A}_{0,2}^{(b)}(p_1, p_2) = -\frac{1}{2p_1}\delta(p_1 - p_2) \ . \tag{3.27}$$

This measure will be further demonstrated to be the relevant one in [31, 32]. One needs to be slightly careful about the meaning of the delta-function in the complex plane, since this formula doesn't look particularly analytic. One way to define it is to remember the appearance of the delta function as the discontinuity of $\frac{1}{2\pi i p}$, i.e. $2\pi i\delta(p) = \lim_{\varepsilon \to 0}\left((p - i\varepsilon)^{-1} - (p + i\varepsilon)^{-1}\right)$. Thus it can be realized as a limiting value of an analytic function. With this understanding, the correct expression is obtained by imposing antisymmetry under $p_1 \to -p_1$ and $p_2 \to -p_2$, which gives

$$\mathsf{A}_{0,2}^{(b)}(p_1, p_2) = -\frac{1}{2\sqrt{p_1^2}}\delta(p_1 - p_2) + \frac{1}{2\sqrt{p_1^2}}\delta(p_1 + p_2) \ . \tag{3.28}$$

This still gives the correct measure on the line $p_1, p_2 \in \mathrm{e}^{-\frac{\pi i}{4}}\mathbb{R}_{\geqslant 0}$, where the physical spectrum is supported.

Curiously, one might think that one can derive (3.28) by applying the dilaton equation (3.16) to $\mathsf{A}_{0,3}^{(b)}$. This almost works, but would predict a lattice of delta functions supported at $p_1 \pm p_2 = mb + nb^{-1}$ for $m, n \in \mathbb{Z}$ and predict an additional factor of $\frac{1}{2}$. We are in particular unsure about the factor of $\frac{1}{2}$, which could be related to a similar numerical mismatch found in [62] when applying the dilaton equation to compute the bosonic string sphere partition function in AdS$_3$. This suggests also that the expression below for $\mathsf{A}_{0,1}^{(b)}(p)$ should be taken with a grain of salt.

**$\mathsf{A}_{0,1}^{(b)}$.** It is now relatively simple to apply the dilaton equation again to (3.28) to infer lower-point amplitudes. Their worldsheet definition becomes more and more subtle, but we essentially take the dilaton equation to define them, a perspective that was repeatedly taken in the literature [13, 62]. It is simplest to use the dilaton equation in the form

$$\mathsf{A}_{g,n+1}^{(b)}(\boldsymbol{p}, p = \tfrac{1}{2}Q) + \mathsf{A}_{g,n+1}^{(b)}(\boldsymbol{p}, p = \tfrac{1}{2}\hat{Q}) = -b\,(2g - 2 + n)\mathsf{A}_{g,n}^{(b)}(\boldsymbol{p}) \ , \tag{3.29}$$

which is obtained by combining (3.16) with its image under swap symmetry. We recall that $\hat{Q} = b^{-1} - b$. Thus plugging in $n = 1$ and $g = 0$ gives

$$\begin{aligned}
\mathsf{A}_{0,1}^{(b)}(p) &= b^{-1}\mathsf{A}_{0,2}^{(b)}(p, \tfrac{1}{2}Q) + b^{-1}\mathsf{A}_{0,2}^{(b)}(p, \tfrac{1}{2}\hat{Q}) \\
&= \frac{1}{2(1 + b^2)}\left(\delta(p + \tfrac{1}{2}Q) - \delta(p - \tfrac{1}{2}Q)\right) \\
&\quad + \frac{1}{2(1 - b^2)}\left(\delta(p + \tfrac{1}{2}\hat{Q}) - \delta(p - \tfrac{1}{2}\hat{Q})\right) \ .
\end{aligned} \tag{3.30}$$

We chose $b \in e^{\frac{\pi i}{4}} \mathbb{R}_{>0}$ and this choice is important to get the relative signs right. Consequently, this answer breaks the $b \to b^{-1}$ duality symmetry and the $b \to -b$ symmetry since they don't preserve the phase of $b$.

This equation tells us in particular that the one-point function does not have any support on the physical spectrum and the one-point functions in string theory of generic operators vanish.

**Sphere partition function.** Finally we obtain the sphere partition function directly from the dilaton equation starting from (3.30). We get

$$\mathsf{A}_{0,0}^{(b)} = \frac{1}{2b}\big(\mathsf{A}_{0,1}^{(b)}(\tfrac{1}{2}Q) + \mathsf{A}_{0,1}^{(b)}(\tfrac{1}{2}\hat{Q})\big) = \infty \ . \tag{3.31}$$

**Torus partition function.** For the torus partition function, we can similarly apply the dilaton equation to $\mathsf{A}_{1,1}^{(b)}(p_1)$. Recalling that $\mathsf{A}_{1,1}^{(b)}(p_1)$ has a pole at $p_1 = \frac{1}{2}\hat{Q}$ and that the Euler characteristic on the right-hand side vanishes gives

$$\infty = 0 \cdot \mathsf{A}_{1,0}^{(b)} \ , \tag{3.32}$$

which means that $\mathsf{A}_{1,0}^{(b)} = \infty$ is also divergent. This can be also directly seen from the worldsheet theory because the Liouville torus paritition functions are divergent.

# 4 Ground ring and higher equations of motion

In the context of the minimal string, there is a known strategy to evaluate string diagrams for low values of $(g, n)$ directly [39]. The method relies on writing the corresponding correlators as total derivatives of a logarithmic correlator on moduli space which localizes the integral to the boundary of moduli space. The logarithmic fields are logarithmic counterparts of the ground ring operators – physical operators at zero ghost number [21, 63, 64]. For the minimal string, this is quite subtle since physical vertex operators have null vectors and these null vectors do not generically decouple. This is the reason why a certain analytic continuation to a 'generalized minimal model' has to be performed.

We will now show that this method works much more straightforwardly for the complex Liouville string. Since physical states are parametrized by a continuum, the subtle phenomena mentioned above do not appear. We will first review the ground ring construction. It is essentially independent of the matter theory under consideration. In the present case, the higher equations of motion do not give the full result for the string amplitudes, but only their values when one external vertex operator is specified to a degenerate value.[13]

---

[13]In contrast, the Virasoro minimal string [13] does not have any ground ring operators, since they rely on null vectors in the representation module. The matter theory for the Virasoro minimal string is timelike Liouville theory which does not admit any degenerate modules in its set of correlation functions, even after analytically continuing them away from the physical spectrum.

## 4.1 Ground ring

**BRST operator.** It is most convenient to describe the ground ring in the BRST formalism of the worldsheet theory. Recall that the BRST operator takes the form

$$\mathcal{Q} = \frac{1}{2\pi i} \oint \mathfrak{c}\big(T^+ + T^- + \tfrac{1}{2}T^{\text{gh}}\big) \, , \tag{4.1}$$

with $T^+$ and $T^-$ the energy momentum tensors of the two Liouville theories and $T^{\text{gh}} = -2(\mathfrak{b}\partial\mathfrak{c}) - (\partial\mathfrak{b}\mathfrak{c})$ the energy momentum tensor of the ghost theory. Similarly, we can construct the right-moving BRST charge $\tilde{\mathcal{Q}}$.

The physical vertex operators that we considered above had ghost number 1 and were of the form

$$\mathcal{V}_p(z) = \mathcal{N}_b(p) \, \mathfrak{c}\tilde{\mathfrak{c}}V_p^+ V_{ip}^- \, , \tag{4.2}$$

and the combined Liouville vertex operator was discussed around eq. (2.14). We also included the leg factor $\mathcal{N}_b(p)$ in this definition. However, it is clear that this list is incomplete. For example the identity vertex operator is clearly BRST closed. It forms the first of an infinite series of BRST closed vertex operators at ghost number zero – the ground ring.

**Ground ring operators.** Ground ring operators will be denoted by $\mathcal{O}_{m,n}$ with $m \in \mathbb{Z}_{\geqslant 1}$ and $n \in \mathbb{Z}_{\geqslant 1}$. They take the form

$$\mathcal{O}_{m,n} = -\frac{\pi\,\mathcal{N}_b\big(\frac{mb}{2} - \frac{n}{2b}\big)}{B_{m,n}}\,\mathcal{H}_{m,n}\tilde{\mathcal{H}}_{m,n}V^+_{p=\frac{mb}{2}+\frac{n}{2b}}V^-_{p=i(\frac{mb}{2}-\frac{n}{2b})} \, . \tag{4.3}$$

Here $\mathcal{H}_{m,n}$ is an expression involving the modes of $T^+$, $T^-$, $\mathfrak{b}$ and $\mathfrak{c}$ of ghost number 0, so that we get a level $mn - 1$ descendant. To fix the normalization of the vertex operators, we stipulate that $\mathcal{H}_{m,n}$ starts with the unit-normalized term

$$\mathcal{H}_{m,n} = (L^+_{-1})^{mn-1} + \dots \tag{4.4}$$

$B_{m,n}$ is a normalization constant that will be introduced below, see eq. (4.14).

The appearing Liouville momenta are the degenerate Liouville momenta. In particular, both $V^+$ and $V^-$ with these choices of Liouville momenta possess a level $mn$ null vector. One may verify that the total conformal weight of these vertex operators is 0.

**Examples.** To make the present discussion more concrete, let us explicitly discuss the first operators in the series. Obviously, $\mathcal{O}_{1,1}$ is the identity vertex operator which gives the first ground ring operator. Thus let us concentrate on the next example $\mathcal{O}_{2,1}$. The operator $\mathcal{H}_{2,1}$ takes the form [38]

$$\mathcal{H}_{2,1} = L^+_{-1} - L^-_{-1} + b^2(\mathfrak{b}\mathfrak{c})_{-1} \, , \tag{4.5}$$

and similarly with $\tilde{\mathcal{H}}_{2,1}$. It is simple to check that $\mathcal{O}_{2,1}$ as defined in (4.3) is closed. In practice, we did these computations with the help of the Mathematica package OPEdefs [65]. We similarly explicitly constructed $\mathcal{O}_{m,n}$ explicitly for $mn \leqslant 6$ in Mathematica. The operator $\mathcal{H}_{3,1}$ takes for example the form

$$\mathcal{H}_{3,1} = (L_{-1}^+)^2 - L_{-1}^+ L_{-1}^- + (L_{-1}^-)^2 + 2b^2(2b^2+1)L_{-2}^+ + 2b^2(2b^2-1)L_{-2}^-$$
$$- 2b^2(2b^2-1)(\mathfrak{b}\mathfrak{c})_{-1}L_{-1}^+ - 2b^2(2b^2+1)(\mathfrak{b}\mathfrak{c})_{-1}L_{-1}^- . \quad (4.6)$$

In this case, there is also a BRST exact term $\mathcal{Q}\mathfrak{b}_{-2}$ one could add and we made an arbitrary choice. The higher operators become very quickly quite complicated.

**Properties.** Since the operators $\mathcal{O}_{m,n}$ are of ghost number 0, we can insert them in the full worldsheet correlators (including the ghosts) without spoiling ghost number conservation. Notice that derivatives $\partial \mathcal{O}_{m,n}$ are BRST exact. This follows from the simple manipulation

$$\partial \mathcal{O}_{m,n} = L_{-1}^{\text{tot}} \mathcal{O}_{m,n} = \{\mathcal{Q}, \mathfrak{b}\} \mathcal{O}_{m,n} = \mathcal{Q}(\mathfrak{b}\mathcal{O}_{m,n}) , \quad (4.7)$$

where we used that the total stress tensor is BRST exact with primitive $\mathfrak{b}$ and $\mathcal{O}_{m,n}$ is closed. This means that up to BRST exact terms, correlation functions involving ground ring operators $\mathcal{O}_{m,n}$ are independent of the location of the ground ring operator. This in particular implies that we can consider the OPE, $\mathcal{O}_{m,n}(z)\mathcal{O}_{m',n'}(0)$, which up to BRST exact terms cannot depend on the separation $z$. By ghost number conservation, the appearing terms must again be of ghost number zero and thus the only non-trivial operators in BRST cohomology are again ground ring operators. In other words, ground ring operators form a ring, which explains their name.

It is simple to work out the ring structure. Clearly, the identity operator $\mathcal{O}_{1,1}$ is the identity of the ring. The remaining ring relations are severely limited by associativity and the fusion rules of the degenerate vertex operators. Up to the choice of $B_{m,n}$ in (4.3) that we have not specified so far, we must have

$$\mathcal{O}_{m,n}(z)\mathcal{O}_{m',n'}(0) = \sum_{m''\overset{2}{=}|m-m'|+1}^{m+m'-1} \sum_{n''\overset{2}{=}|n-n'|+1}^{n+n'-1} \mathcal{O}_{m'',n''}(0) + \text{BRST exact} . \quad (4.8)$$

Here the notation $\overset{2}{=}$ means that the summation index increases in steps of two. We checked directly in Mathematica that this equation holds with unit coefficients with the definition of $B_{m,n}$ as in (4.14), at least for $mn \leqslant 6$. There does not seem to be a proof available that demonstrates this analytically.

More importantly than the ring structure itself will be the fact that the ordinary physical vertex operators as given in (4.2) must form modules of the ground ring operators. Up to the normalization on which we comment below, their form is

again severely limited by the knowledge of degenerate Virasoro fusion rules and associativity of the action. We have

$$\mathcal{O}_{m,n}(z)\mathcal{V}_p(0) = \sum_{r \overset{2}{=} 1-m}^{m-1} \sum_{s \overset{2}{=} 1-n}^{n-1} \mathcal{V}_{p+\frac{rb}{2}+\frac{s}{2b}}(0) + \text{BRST exact} . \tag{4.9}$$

This assumes that we chose $p$ generically since if $p$ corresponds to a degenerate value, some of the terms appearing on the right hand side may be forbidden. This is not a severe restriction in our case, since $p$ can take continuous values, but is at the origin of the subtleties that appear in the ordinary minimal string [66]. For degenerate values of $p$ as they appear in the minimal string, one has to modify the higher equations of motion that we will explain below by hand to account for this [67]. We again checked in `Mathematica` that (4.9) holds with unit coefficients, at least up to levels $mn \leqslant 6$.[14]

## 4.2   Logarithmic operators and higher equations of motion

We now describe logarithmic analogues of $\mathcal{O}_{m,n}$ and their appearance in the higher equations of motion.

**Liouville theory.**   Let us start by considering ordinary Liouville theory, say the first factor in the worldsheet theory. This was first discussed in [38]. We can consider the operators

$$V_p'^+ \equiv \frac{1}{2}\frac{\partial}{\partial p}V_p^+ . \tag{4.10}$$

These are not primary operators. In terms of states, we have $L_0|p\rangle = (\frac{Q^2}{4} - p^2)|p\rangle$ and hence for the state $|p\rangle'$ corresponding to $V_p'$,

$$L_0^+|p\rangle' = \frac{1}{2}\frac{\partial}{\partial p}\Big((\tfrac{Q^2}{4} - p^2)|p\rangle\Big) = (\tfrac{Q^2}{4} - p^2)|p\rangle' - p|p\rangle . \tag{4.11}$$

Hence $L_0$ does not act diagonally which is the hallmark of a logarithmic field. We can straightforwardly define correlation functions of $V_p'$ by taking derivatives of correlation functions in Liouville theory with ordinary primary vertex operators.

In the following we will be interested in $V_p'$ where $p$ takes a degenerate value. Recall that in this case $V_p$ has a level $mn$ null vector and hence there is an operator $D_{m,n}$ made out of Virasoro modes that annihilates the primary vertex operator

$$D_{m,n}V_{p=\frac{mb}{2}+\frac{n}{2b}}^+ = 0 , \tag{4.12}$$

---

[14]The appearance of unit coefficients is the main motivation from the worldsheet for the choice of leg factors (2.20a).

and similarly for the right-moving modes. We normalize $D_{m,n}$ so that it starts with the term $L_{-1}^{mn}$. The remarkable observation of [38] was that[15]

$$V^+_{p=\frac{mb}{2}-\frac{n}{2b}} = B^{-1}_{m,n} D_{m,n} \tilde{D}_{m,n} V'^+_{p=\frac{mb}{2}+\frac{n}{2b}} \ . \tag{4.13}$$

Here $B_{m,n}$ is a constant that we determine below (and is somewhat magically the same constant that we included in the definition of the ground ring operators in (4.3)).

It is simple to verify that the right-hand-side transforms like a primary field. The main reason for this is that the logarithmic terms do not matter for this since they are shared between the left- and right-movers, but we applied both $D_{m,n}$ and $\tilde{D}_{m,n}$. Thus the logarithmic terms produced by acting with left-moving Virasoro modes will be proportional to $\tilde{D}_{m,n} V^+_{p=\frac{mb}{2}+\frac{n}{2b}}$, which vanishes from the null-vector constraint. One can then compute $B_{m,n}$ explicitly by inserting this into a three-point function. As explained in [38], the action of the operators $D_{m,n}$ and $\tilde{D}_{m,n}$ on the three-point function can be computed from the knowledge of the fusion rules. The result is

$$B_{m,n} = \frac{\prod_{r=1-m}^{m-1} \prod_{s=1-n}^{n-1} (p_1 \pm p_2 - \frac{rb}{2} - \frac{s}{2b})^2}{C_b(\frac{mb}{2} - \frac{n}{2b}, p_1, p_2)} \frac{1}{2} \frac{\partial}{\partial p}\bigg|_{p=\frac{mb}{2}+\frac{n}{2b}} C_b(p, p_1, p_2) \ . \tag{4.14}$$

This can be evaluated more explicitly, but the expression is not particularly illuminating [38]. Importantly, one can however check that the right-hand-side is independent of $p_1$ and $p_2$. The relation (4.13) is the quantum analogue of the infinite number of conservation equations of Liouville theory, reflecting its integrable structure. In particular, $V'^+_{p=\frac{b}{2}+\frac{1}{2b}}$ is just the Liouville field $\varphi$ and thus the case $m = n = 1$ corresponds to the ordinary equation of motion.

**Logarithmic ground ring operators.** We will now define logarithmic ground ring operators $\mathcal{O}'^+_{m,n}$ and $\mathcal{O}'^-_{m,n}$. They are *not* BRST closed and are thus *not* physical operators. Nonetheless, they will be the corresponding analogues of the logarithmic operators $V'^+_p$ in the full worldsheet theory. We define them in the obvious way analogous to (4.3) as

$$\mathcal{O}'^+_{m,n} = -\frac{\pi \mathcal{N}_b\left(\frac{mb}{2} - \frac{n}{2b}\right)}{B_{m,n}} \mathcal{H}_{m,n} \tilde{\mathcal{H}}_{m,n} V'^+_{p=\frac{mb}{2}+\frac{n}{2b}} V^-_{p=i\left(\frac{mb}{2} - \frac{n}{2b}\right)} \ , \tag{4.15}$$

and similarly the vertex operator of the second Liouville theory becomes logarithmic for $\mathcal{O}'^-_{m,n}$.

---

[15]This holds in flat space. In curved space, there is a correction involving the Riemann scalar. It can be fully fixed by imposing that the right-hand side of (4.13) transforms correctly under Weyl rescalings.

**Higher equations of motion.** If we act with $\mathcal{Q}\tilde{\mathcal{Q}}$ on the logarithmic ground ring operators, we do not get zero. However, we are guaranteed that $\mathcal{Q}\tilde{\mathcal{Q}}\mathcal{O}'^{+}_{m,n}$ is a BRST-closed operator since $\mathcal{Q}^2 = \tilde{\mathcal{Q}}^2 = 0$. This operator can only be $\mathcal{V}_{p=\frac{mb}{2}-\frac{n}{2b}}$. We can fix the normalization by looking at the leading term of $\mathcal{H}_{m,n}$, which we fixed in (4.4). The action of the BRST operator (4.1) in particular gives the term $\mathfrak{c}_1 L^+_{-1}$. Comparing with the normalization of $D_{m,n}$ gives

$$
\begin{aligned}
\mathcal{Q}\tilde{\mathcal{Q}}\mathcal{O}'^{+}_{m,n} &= -\frac{\pi \mathcal{N}_b\left(\frac{mb}{2}-\frac{n}{2b}\right)}{B_{m,n}} \mathfrak{c}_1\tilde{\mathfrak{c}}_1 D^+_{m,n}\tilde{D}^+_{m,n} V'^{+}_{p=\frac{mb}{2}+\frac{n}{2b}} V^{-}_{p=i\left(\frac{mb}{2}-\frac{n}{2b}\right)} + \text{BRST exact} \\
&= -\pi \mathcal{N}_b\left(\tfrac{mb}{2}-\tfrac{n}{2b}\right) \mathfrak{c}_1\tilde{\mathfrak{c}}_1 V^+_{p=\frac{mb}{2}-\frac{n}{2b}} V^{-}_{p=i\left(\frac{mb}{2}-\frac{n}{2b}\right)} + \text{BRST exact} \\
&= -\pi \mathcal{V}_{p=\frac{mb}{2}-\frac{n}{2b}} + \text{BRST exact} .
\end{aligned}
\tag{4.16}
$$

Here we used the Liouville higher equation of motion (4.13). This also shows why it was a good idea to include the normalization factors in (4.3). Similarly, we have

$$
\begin{aligned}
\mathcal{Q}\tilde{\mathcal{Q}}\mathcal{O}'^{-}_{m,n} &= -\frac{\pi \mathcal{N}_b\left(\frac{mb}{2}-\frac{n}{2b}\right)}{B_{m,n}} \mathfrak{c}_1\tilde{\mathfrak{c}}_1 V^+_{p=\frac{mb}{2}+\frac{n}{2b}} D^-_{m,n}\tilde{D}^-_{m,n} V'^{-}_{p=i\left(\frac{mb}{2}-\frac{n}{2b}\right)} + \text{BRST exact} \\
&= -\frac{\pi B^-_{m,n} \mathcal{N}_b\left(\frac{mb}{2}-\frac{n}{2b}\right)}{B_{m,n}\mathcal{N}_b\left(\frac{mb}{2}+\frac{n}{2b}\right)} \mathcal{V}_{p=\frac{mb}{2}+\frac{n}{2b}} + \text{BRST exact} \\
&= -\pi i\, \mathcal{V}_{p=\frac{mb}{2}+\frac{n}{2b}} + \text{BRST exact}
\end{aligned}
\tag{4.17}
$$

where $B^-_{m,n}$ is $B_{m,n}$ with $b \to -ib$ replaced since it comes from the second Liouville factor. The last equality is simple to check with the explicit formulas given in [38]. The extra $i$ comes essentially from the fact that we defined the logarithmic operator in the second theory by a derivative in $p^-$, which differs from $p^+$ derivatives by a factor of $i$.

**OPE of the logarithmic ground ring operators.** The last ingredient that we need for the application of the higher equations of motion to the string diagrams is the OPE of $\mathcal{O}'^{\pm}_{m,n}$ with a standard physical vertex operator (4.2). $\mathcal{O}'^{\pm}_{m,n}$ is not BRST closed, and thus this OPE is much more complicated. Since $\mathcal{O}'^{\pm}_{m,n}$ is a logarithmic field, there will be in particular a logarithmic term in the OPE and this is the only term that is relevant for our purposes. Even though $\mathcal{O}'^{\pm}_{m,n}$ is not BRST closed, the field appearing in the logarithmic term has to be BRST closed. This follows from the higher equations of motion. We have

$$
\mathcal{Q}\tilde{\mathcal{Q}}\big(\mathcal{O}'^{+}_{m,n}(z)\mathcal{V}_p(0)\big) = -\pi \mathcal{V}_{p=\frac{mb}{2}+\frac{n}{2b}}(z)\mathcal{V}_p(0) ,
\tag{4.18}
$$

which does not contain a logarithmic term and hence the logarithmic term appearing in the $\mathcal{O}'^{\pm}_{m,n}(z)\mathcal{V}_p(0)$ OPE has to be BRST closed. Since the ghost number of the appearing operators is 1, and in view of the degenerate Virasoro fusion rules the only

possibility is that

$$\mathcal{O}'^+_{m,n}(z)\mathcal{V}_p(0) = \sum_{r\overset{2}{=}1-m}^{m-1}\sum_{s\overset{2}{=}1-n}^{n-1} C^{r,s}_{m,n}(p)\mathcal{V}_{p+\frac{rb}{2}+\frac{s}{2b}}(0)\log|z|^2 + \dots \,, \qquad (4.19)$$

where ... contains both BRST-exact terms and non-logarithmic terms and $C^{r,s}_{m,n}(p)$ are the structure constants. They can be computed as follows. The logarithmic term can only appear when the $\frac{1}{2}\frac{\partial}{\partial p'}$ derivative in the definition of the logarithmic ground ring operators (4.15) hits the exponent of the position dependence $|z|$ of the non-logarithmic OPE. The remaining structure constant is then exactly the same as in the OPE of $\mathcal{O}_{m,n}$ with $\mathcal{V}_p$ and thus $C^{r,s}_{m,n}(p)$ is given by the derivative of the $z$-exponent. Thus we pick out a discrete term in the OPE of the primary fields, which gives rise to a $z$-dependence

$$|z|^{-2(\frac{Q^2}{4}-(p')^2)-2(\frac{Q^2}{4}-p^2)+2(\frac{Q^2}{4}-(p-p'+\frac{1}{2}(r+m)b+\frac{1}{2}(s+n)b^{-1})^2)}\,, \qquad (4.20)$$

where $p'$ is the momentum of the Liouville vertex operator before specifying to $p' = \frac{mb}{2}+\frac{n}{2b}$. With these choices, the momentum $p-p'+\frac{1}{2}(r+m)b+\frac{1}{2}(s+n)b^{-1}$ appearing in the OPE will simplify to $p+\frac{rb}{2}+\frac{s}{2b}$ after specializing $p'$. The operator $\mathcal{H}_{m,n}\tilde{\mathcal{H}}_{m,n}$ leads to further $z$-dependence, but it is independent of $p'$. We thus simply have to take a $p'$ derivative and put $p' = \frac{mb}{2}+\frac{n}{2b}$, which leads to

$$C^{r,s}_{m,n}(p) = p + \frac{rb}{2} + \frac{s}{2b} + \frac{mb}{2} + \frac{n}{2b}\,. \qquad (4.21)$$

This cannot be the full answer since it would break reflection symmetry in $p$. In fact there is a second discrete term with momentum $p+p'+\frac{1}{2}(r-m)b+\frac{1}{2}(s-n)b^{-1}$ that could also have contributed to the logarithm and it would give rise to

$$C^{r,s}_{m,n}(p) = -\left(p + \frac{rb}{2} + \frac{s}{2b}\right) + \frac{mb}{2} + \frac{n}{2b}\,. \qquad (4.22)$$

To decide which one is the correct answer, we should note that the computation of the logarithmic term is in general ill-defined. Indeed, the OPE can contain a continuum of operators and it is not clear how we would separate the logarithmic term. This only works in special circumstances when the continuum of operators appears with positive exponents. In other words, to get an unambiguous answer, we must require that the discrete term that we considered appears indeed in the OPE (i.e. the corresponding pole crossed the OPE contour) and is more singular than the continuum. For the discrete term $p - p' + \frac{1}{2}(r + m)b + \frac{1}{2}(s + n)b^{-1}$, the contour is crossed (for $p'$ close to $\frac{mb}{2}+\frac{n}{2b}$) when

$$\text{Re}\left(p - p' + \tfrac{1}{2}(r + m)b + \tfrac{1}{2}(s + n)b^{-1}\right) = \text{Re}\left(p + \tfrac{rb}{2} + \tfrac{s}{2b}\right) < 0\,, \qquad (4.23)$$

while the inequality is reversed for the other discrete contribution. For both contributions, they are separated from the continuum when

$$\mathrm{Re}\left(p + \tfrac{rb}{2} + \tfrac{s}{2b}\right)^2 > 0 \ . \tag{4.24}$$

Thus we finally obtain

$$C_{m,n}^{r,s}(p) = -\sqrt{\left(p + \frac{rb}{2} + \frac{s}{2b}\right)^2} + \frac{mb}{2} + \frac{n}{2b} \tag{4.25}$$

provided that $\mathrm{Re}\left(p + \tfrac{rb}{2} + \tfrac{s}{2b}\right)^2 > 0$. The reasoning of the square root of the square was explained below eq. (3.16). When $\mathrm{Re}\left(p + \tfrac{rb}{2} + \tfrac{s}{2b}\right)^2 < 0$, the answer is in principle undefined. One can of course analytically continue (4.25), but the branch of the square root is then ambiguous.

## 4.3 Higher equations of motion for $\mathsf{A}_{0,4}^{(b)}$

We now explain the technology towards the analytic computation of $\mathsf{A}_{0,4}^{(b)}$. We consider the case when $p_4 = \tfrac{mb}{2} - \tfrac{n}{2b}$. Then we can write

$$
\begin{aligned}
\mathsf{A}_{0,4}^{(b)}&(p_1, p_2, p_3, p_4 = \tfrac{mb}{2} - \tfrac{n}{2b}) \\
&= -C_{\mathrm{S}^2}^{(b)} \int \mathrm{d}^2 z \ \langle \mathcal{V}_{p_1}(z_1) \mathcal{V}_{p_2}(z_2) \mathcal{V}_{p_3}(z_3) \mathfrak{b}_{-1}\tilde{\mathfrak{b}}_{-1} \mathcal{V}_{p_4=\frac{mb}{2}-\frac{n}{2b}}(z) \rangle \\
&= \frac{1}{\pi} C_{\mathrm{S}^2}^{(b)} \int \mathrm{d}^2 z \ \langle \mathcal{V}_{p_1}(z_1) \mathcal{V}_{p_2}(z_2) \mathcal{V}_{p_3}(z_3) \mathfrak{b}_{-1}\tilde{\mathfrak{b}}_{-1} \mathcal{Q}\tilde{\mathcal{Q}} \mathcal{O}_{m,n}'^{+}(z) \rangle \\
&= -\frac{1}{\pi} C_{\mathrm{S}^2}^{(b)} \int \mathrm{d}^2 z \ \langle \mathcal{V}_{p_1}(z_1) \mathcal{V}_{p_2}(z_2) \mathcal{V}_{p_3}(z_3) \{\mathcal{Q}, \mathfrak{b}_{-1}\}\{\tilde{\mathcal{Q}}, \tilde{\mathfrak{b}}_{-1}\} \mathcal{O}_{m,n}'^{+}(z) \rangle \\
&= -\frac{1}{\pi} C_{\mathrm{S}^2}^{(b)} \int \mathrm{d}^2 z \ \langle \mathcal{V}_{p_1}(z_1) \mathcal{V}_{p_2}(z_2) \mathcal{V}_{p_3}(z_3) \partial\bar{\partial} \mathcal{O}_{m,n}'^{+}(z) \rangle \ . 
\end{aligned}
\tag{4.26}
$$

The minus sign is necessary since we have to anticommute $\tilde{\mathfrak{b}}_{-1}$ with $\mathfrak{c}_1$ to turn the last vertex operator back to the form of (2.19). Thus, the integrand is at this point a total derivative and reduces to boundary contributions from $z_j$ and $\infty$ (where we have a curvature singularity).

**Contribution from $z_j$.** The contribution from $z_1$, $z_2$ and $z_3$ is dictated by the logarithmic term in the OPE between $\mathcal{O}_{m,n}'^{+}$ and the primary $\mathcal{V}_{p_j}$. We go to polar coordinates around $z_j$. Then

$$\int \mathrm{d}^2 z \ \partial\bar{\partial} f = \frac{1}{4} \times 2\pi \times \int_\varepsilon \mathrm{d}r \ r \frac{1}{r} \partial_r(r\partial_r f) = -\frac{\pi}{2} \lim_{r \to 0} r\partial_r f \tag{4.27}$$

Thus, for a logarithmic dependence $f \sim \log r^2$, we get $-\pi$ times the prefactor of the logarithm. Thus the boundary contribution from $z_1$ is equal to

$$C_{\text{S}^2}^{(b)} \sum_{r\overset{2}{=}1-m} \sum_{s\overset{2}{=}1-n}^{m-1} \sum^{n-1} C_{m,n}^{r,s}(p_1) \langle \mathcal{V}_{p_1+\frac{rb}{2}+\frac{s}{2b}}(z_1) \mathcal{V}_{p_2}(z_2) \mathcal{V}_{p_3}(z_3) \rangle$$

$$= \mathsf{A}_{0,3}^{(b)}(p_1 + \tfrac{(m-1)b}{2} + \tfrac{n-1}{2b}, p_2, p_3) \sum_{r\overset{2}{=}1-m}^{m-1} \sum_{s\overset{2}{=}1-n}^{n-1} \left( \frac{mb}{2} + \frac{n}{2b} - \sqrt{\left(p_1 + \frac{rb}{2} + \frac{s}{2b}\right)^2} \right) .$$

$$(4.28)$$

We used that the three-point function happens to be double-periodic and thus the shifts in $p_1$ all lead to identical results and only depend on the parity of $m$ and $n \bmod \mathbb{Z}$. In principle, this only holds when the logarithmic contribution is well-separated from the continuum, see (4.24), however we will suppress this condition, which leads to ambiguities of the sign.

**Contribution from $\infty$.** The higher equations of motion only hold with a flat background metric, see footnote 15. Thus we also get a boundary contribution from $z = \infty$. It can be read off from the anomalous transformation behaviour under conformal transformation behaviour. We have

$$\mathcal{O}_{m,n}'^+(z) \sim (mb + nb^{-1}) \mathcal{O}_{m,n}(z) \log |z|^2 \qquad (4.29)$$

as $z \to \infty$. We can evaluate the contribution similarly as above by going to polar coordinates. Since the insertion of $\mathcal{O}_{m,n}(z)$ is position-independent, we get for the contribution at infinity,

$$- C_{\text{S}^2}^{(b)}(mb + nb^{-1}) \langle \mathcal{V}_1(z_1) \mathcal{V}_2(z_2) \mathcal{V}_3(z_3) \mathcal{O}_{m,n}(\infty) \rangle$$

$$= -(mb + nb^{-1}) mn \, \mathsf{A}_{0,3}^{(b)}(p_1 + \tfrac{(m-1)b}{2} + \tfrac{n-1}{2b}, p_2, p_3) . \quad (4.30)$$

Here we used again double-periodicity of the three-point function, which gives $mn$ identical terms in the OPE. Finally, we get

$$\mathsf{A}_{0,4}^{(b)}(p_1, p_2, p_3, p_4 = \tfrac{mb}{2} - \tfrac{n}{2b}) = \mathsf{A}_{0,3}^{(b)}(p_1 + \tfrac{(m-1)b}{2} + \tfrac{n-1}{2b}, p_2, p_3)$$

$$\times \left[ \frac{mn(mb + nb^{-1})}{2} - \sum_{i=1}^{3} \sum_{r\overset{2}{=}1-m}^{m-1} \sum_{s\overset{2}{=}1-n}^{n-1} \sqrt{\left(p_i + \frac{rb}{2} + \frac{s}{2b}\right)^2} \right] . \quad (4.31)$$

Notice that the case $m = n = 1$ reproduces the dilaton equation (3.16). Notice also that combining the higher equations of motion with triality symmetry (3.25) maps these special values to the residues of the poles of $\mathsf{A}_{0,4}^{(b)}$.

## 4.4 Higher equations of motion for $\mathbf{A}_{1,1}^{(b)}$

Let us similarly derive the higher equations of the once-punctured torus. This is actually redundant, since they can be obtained directly from the higher equations of

motion of the four-punctured sphere, together with the relation (3.26). We anyway explain the direct derivation from the worldsheet for completeness. Partial results were already obtained in [68] for the minimal string, which we initially follow closely. We have

$$
\begin{aligned}
\mathsf{A}^{(b)}_{1,1}\left(p_1 = \tfrac{mb}{2} - \tfrac{n}{2b}\right) &= -\frac{1}{2(2\pi)^2} \int_{\mathcal{F}} \mathrm{d}^2\tau \; \langle \mathfrak{b}\tilde{\mathfrak{b}}(z) \mathcal{V}_{p_1 = \frac{mb}{2} - \frac{n}{2b}}(0) \rangle_\tau \\
&= \frac{1}{(2\pi)^3} \int_{\mathcal{F}} \mathrm{d}^2\tau \; \langle \mathfrak{b}\tilde{\mathfrak{b}}(z) \mathcal{Q}\tilde{\mathcal{Q}}\mathcal{O}'^{+}_{m,n}(0) \rangle_\tau \\
&= -\frac{1}{(2\pi)^3} \int_{\mathcal{F}} \mathrm{d}^2\tau \; \langle \mathcal{Q}\mathfrak{b}(z)\tilde{\mathcal{Q}}\tilde{\mathfrak{b}}(z)\mathcal{O}'^{+}_{m,n}(0) \rangle_\tau \\
&= -\frac{1}{(2\pi)^3} \int_{\mathcal{F}} \mathrm{d}^2\tau \; \langle T(z)\tilde{T}(z)\,\mathcal{O}'^{+}_{m,n}(0) \rangle_\tau \; .
\end{aligned}
\tag{4.32}
$$

Here, we used that we need to insert one pair of $\mathfrak{b}$-ghosts in the correlation function to saturate the ghost zero modes. To connect with (2.21), we use that

$$
\langle \mathfrak{b}(z)\mathfrak{c}(w)\tilde{\mathfrak{b}}(z)\tilde{\mathfrak{c}}(w) \rangle_\tau = (2\pi)^4 |\eta(\tau)|^4
\tag{4.33}
$$

in the ghost CFT. We then used the higher equations of motion and the fact that $\mathcal{Q}\mathfrak{b} = T$.

**Virasoro Ward identities.** We next apply the Virasoro Ward identities. Let us rederive them since they are potentially subtle in the presence of a logarithmic field.

To warm up, let us first show that the one-point function $\langle \mathcal{O}_{m,n}(0) \rangle$ is independent of $\tau$. We have

$$
\begin{aligned}
2\pi i \, \frac{\partial}{\partial \tau} \langle \mathcal{O}_{m,n}(0) \rangle_\tau &= \int_0^1 \mathrm{d}z \; \langle T(z)\mathcal{O}_{m,n} \rangle_\tau \\
&= \int_{\partial F} \frac{\mathrm{d}z}{2\pi i} \; \partial_z \log \vartheta_1(z|\tau) \; \langle T(z)\mathcal{O}_{m,n}(0) \rangle_\tau \\
&= \operatorname*{Res}_{z=0} \partial_z \log \vartheta_1(z|\tau) \; \langle T(z)\mathcal{O}_{m,n}(0) \rangle_\tau \\
&= \langle L_{-2}\mathcal{O}_{m,n}(0) \rangle_\tau \; .
\end{aligned}
\tag{4.34}
$$

Here, $F$ is the boundary of the fundamental domain of the torus (which can be taken to be the parallelogram spanned by the four points 0, 1, $\tau$ and $\tau + 1$). We used the quasiperiodicity properties of $\partial_z \log \vartheta_1(z|\tau)$ and then deformed the contour. We finally use that the OPE of $T(z)$ with $\mathcal{O}_{m,n}(0)$ does not contain a second order pole (since $\mathcal{O}_{m,n}$ has vanishing conformal weight) and the first order pole is proportional to a total derivative which vanishes on the torus due to translation invariance. However, we have

$$
\langle L_{-2}\mathcal{O}_{m,n}(0) \rangle_\tau = \langle \mathcal{Q}(\mathfrak{b}_{-2}\mathcal{O}_{m,n}(0)) \rangle_\tau = 0
\tag{4.35}
$$

since it is BRST exact.

We now repeat the same manipulations for the logarithmic partner. It proceeds the same way and we have

$$(2\pi)^2 \frac{\partial^2}{\partial\tau\partial\bar\tau} \langle \mathcal{O}'^+_{m,n}(0)\rangle_\tau = \langle L_{-2}\tilde{L}_{-2}\mathcal{O}'^+_{m,n}(0)\rangle = \langle T(z)\tilde{T}(z)\mathcal{O}'^+_{m,n}(0)\rangle_\tau .$$ (4.36)

In the process, we encounter again the OPE of $T(z)$ with $\mathcal{O}'^+_{m,n}$, which now also contains $\mathcal{O}_{m,n}$ in the second order pole due to the logarithmic structure. But because we have both a $\tau$ and a $\bar\tau$-derivative, this term eventually does not contribute.

This is precisely the correlator appearing the integral (4.32) and we hence get a total derivative over moduli space:

$$\mathsf{A}^{(b)}_{1,1}(p_1 = \tfrac{mb}{2} - \tfrac{n}{2b}) = -\frac{1}{2\pi} \int_{\mathcal{F}} \mathrm{d}^2\tau \frac{\partial^2}{\partial\tau\partial\bar\tau} \langle \mathcal{O}'^+_{m,n}(0)\rangle_\tau .$$ (4.37)

We now collect the different contributions from the boundary of moduli space. We use the standard fundamental domain $\mathcal{F}$. The contributions from the vertical lines $\mathrm{Re}\,\tau = \frac{1}{2}$ and $\mathrm{Re}\,\tau = -\frac{1}{2}$ cancel, since the integrand is periodic in $\tau \to \tau + 1$.

**Contribution from the unit arc.** The fundamental domain has an arc of the unit circle as a boundary. Since we can map the left part of the arc to the right part, they partially cancel out. In fact, we have the transformation property

$$\langle \mathcal{O}'^+_{m,n}(0)\rangle_{-\frac{1}{\tau}} = \langle \mathcal{O}'^+_{m,n}(0)\rangle_\tau - \frac{1}{2}(mb + nb^{-1}) \log|\tau|^2 \langle \mathcal{O}_{m,n}(0)\rangle_\tau ,$$ (4.38)

which follows from taking a derivative of the corresponding relation

$$\langle \Phi(0)\rangle_{-\frac{1}{\tau}} = |\tau|^{2h}\langle \Phi(0)\rangle_\tau$$ (4.39)

of a one-point function of a primary field. We go to polar coordinates and compute[16]

$$
\begin{aligned}
\frac{1}{4} & \int_1 \mathrm{d}r \int_0^{\frac{\pi}{6}} \mathrm{d}\phi \, \frac{\partial}{\partial r} r \frac{\partial}{\partial r} \big( \langle \mathcal{O}'_{m,n}(0)\rangle_{ire^{i\phi}} + \langle \mathcal{O}'_{m,n}(0)\rangle_{ire^{-i\phi}} \big) \\
& = -\frac{1}{4} \int_0^{\frac{\pi}{6}} \mathrm{d}\phi \, \frac{\partial}{\partial r} \big( \langle \mathcal{O}'^+_{m,n}(0)\rangle_{ire^{i\phi}} + \langle \mathcal{O}'^+_{m,n}(0)\rangle_{ire^{-i\phi}} \big)\big|_{r=1} \\
& = -\frac{1}{4} \int_0^{\frac{\pi}{6}} \mathrm{d}\phi \, \frac{\partial}{\partial r} \big( \langle \mathcal{O}'^+_{m,n}(0)\rangle_{ire^{i\phi}} + \langle \mathcal{O}'^+_{m,n}(0)\rangle_{ir^{-1}e^{i\phi}} \\
& \qquad - (mb + nb^{-1}) \log r^{-1}\langle \mathcal{O}_{m,n}(0)\rangle_\tau \big)\big|_{r=1} \\
& = -\frac{mb + nb^{-1}}{4} \int_0^{\frac{\pi}{6}} \mathrm{d}\phi \, \langle \mathcal{O}_{m,n}(0)\rangle_\tau \\
& = -\frac{\pi\,(mb + nb^{-1})}{24}\langle \mathcal{O}_{m,n}(0)\rangle_\tau .
\end{aligned}
$$ (4.40)

---

[16]The factor of $\frac{1}{4}$ comes from $\frac{\partial^2}{\partial\tau\partial\bar\tau} = \frac{1}{4}\Delta_\tau$.

Thus, it remains to compute the one-point function $\langle \mathcal{O}_{m,n} \rangle$. We can do it by inserting a complete set of states, just as we would do in the computation of the one-point function on the torus when we expand in terms of conformal blocks. In the limit $\mathrm{Im}\,\tau \to \infty$, we only have to keep primary states (and the conformal blocks trivialize). Since $\mathcal{O}_{m,n}$ is not part of the physical spectrum of Liouville theory, we have to deform the contour. We hence pick up some number of discrete terms. Because of the OPE (4.8), we have[17]

$$\langle \mathcal{V}_p \, | \mathcal{O}_{m,n}(1) | \mathcal{V}_p \rangle = \sum_{r \overset{2}{=} 1-m}^{m-1} \sum_{s \overset{2}{=} 1-n}^{n-1} \langle \mathcal{V}_p \, | \, \mathcal{V}_{p + \frac{rb}{2} + \frac{s}{2b}} \rangle \, . \tag{4.41}$$

If $m$ and $n$ are both odd, the right-hand-side contains the norm $\langle \mathcal{V}_p | \mathcal{V}_p \rangle$. In this case $\langle \mathcal{O}_{m,n} \rangle$ diverges. Indeed, these values correspond to the poles of $\mathsf{A}_{1,1}^{(b)}$. So we will assume that either $m$ or $n$ is even. Then the right-hand-side of (4.41) vanishes. The only way to get around this conclusion is if

$$p = -\left( p + \frac{rb}{2} + \frac{s}{2b} \right) \tag{4.42}$$

for some $r$ or $s$. This is impossible for purely imaginary $p$ and thus only the discrete terms in the complete sum over states can contribute. The appearance of discrete contributions is analyzed as in section 3.1 for the four-point function. The result is that we get the additional discrete contributions for

$$p = -\frac{kb}{4} - \frac{\ell}{4b} \, , \tag{4.43}$$

where $k$ runs from $1-m$ to $m-1$ in steps of 2 and $\ell$ runs from $1-n$ to $n-1$ in steps of 2. They precisely satisfy (4.42) for $r = k$ and $s = \ell$. Each of them contributes $-1$ to $\langle \mathcal{O}_{m,n} \rangle$ (since the leg factor (2.20a) is odd under reflection). We get an additional factor of $\frac{1}{2}$ since $(k, \ell) \to (-k, -\ell)$ corresponds to the identical state (which by our convention we only count once). Thus we have simply[18]

$$\langle \mathcal{O}_{m,n}(0) \rangle_\tau = -\frac{1}{2} mn \tag{4.44}$$

and thus the unit arc contributes

$$-\frac{1}{32} \frac{mn(mb + nb^{-1})}{3} \tag{4.45}$$

to $\mathsf{A}_{1,1}^{(b)}$.

---

[17]We use this notation since $\langle \mathcal{V}_p |$ is has ghost number 2 and is the field $\mathfrak{c}\bar{\mathfrak{c}}\mathcal{V}_p$ placed at infinity. This is necessary to saturate the ghost number.

[18]The overall normalization of this expression may be easily checked directly for $\mathcal{O}_{2,1}$, where we gave the explicit form of the operator $\mathcal{H}_{2,1}$ in (4.5). Since we have to saturate the ghost numbers, only the last term in (4.5) contributes.

**Contribution from $\tau = i\infty$.** We get another contribution from the cusp. Going to Cartesian coordinates, it reads

$$-\frac{1}{8\pi} \times \lim_{\tau_2 \to \infty} \frac{\partial}{\partial \tau_2} \langle \mathcal{O}_{m,n}^{\prime+}(0) \rangle \ , \tag{4.46}$$

where $\tau_2 = \mathrm{Im}\,\tau$. This selects the logarithmic part of the correlator that grows linearly in $\tau_2$. We can compute it by using essentially the same logic as in the derivative of the logarithmic ground ring OPE (4.25). Let $p'$ denote again the momentum of the vertex operator in the first Liouville theory before we take the derivative $\frac{1}{2}\frac{\partial}{\partial p'}$. The role of $z$ in (4.19) is played by the local coordinate $q = \mathrm{e}^{2\pi i\tau}$. The leading behaviour of the one-point block is given by $|q|^{2(\frac{Q^2}{4} - p_{\mathrm{int}}^2)}$, where $p_{\mathrm{int}}$ is the internal Liouville momentum of the first Liouville theory, which takes the form

$$p_{\mathrm{int}} = -\frac{p'}{2} + \frac{(k+m)b}{4} + \frac{\ell + n}{4b} \quad \text{or} \quad \frac{p'}{2} + \frac{(k-m)b}{4} + \frac{\ell - n}{4b} \ . \tag{4.47}$$

. Thus a discrete term contributes to the to the logarithmic part of $\langle \mathcal{O}_{m,n}'(0) \rangle$ as

$$\frac{\pi \tau_2}{2}(kb + \ell b^{-1}) \quad \text{or} \quad -\frac{\pi \tau_2}{2}(kb + \ell b^{-1}) \ , \tag{4.48}$$

depending on which discrete term we pick up.

Similarly to the discussion around (4.25), the first term is to be taken when

$$\mathrm{Re}(p_{\mathrm{int}}) = \mathrm{Re}\left(-\frac{p'}{2} + \frac{(k+m)b}{4} + \frac{\ell + n}{4b}\right) \sim \mathrm{Re}\left(\frac{kb}{4} + \frac{\ell}{4b}\right) < 0 \ , \tag{4.49}$$

while the second term is the correct choice for the reverse inequality. The result is only unambiguous when additionally $\mathrm{Re}\left(\frac{kb}{4} + \frac{\ell}{4b}\right)^2 > 0$ since otherwise the logarithmic term cannot be separated from the continuum and the limit (4.46). We will omit this condition in the following. Finally, we get a factor of $\frac{1}{2}$ since the contributions corresponding to $(r,s)$ and $(-r,-s)$ are equivalent. Plugging this behaviour into (4.46) gives the following contribution from the cusp

$$\frac{1}{32} \sum_{r \overset{2}{=} 1-m}^{m-1} \sum_{s \overset{2}{=} 1-n}^{n-1} \sqrt{\left(rb + sb^{-1}\right)^2} \tag{4.50}$$

**Summary.** Combining (4.45) and (4.50), we hence finally derived

$$\mathsf{A}_{1,1}^{(b)}(p_1 = \tfrac{mb}{2} - \tfrac{n}{2b}) = \frac{1}{32}\left[ \sum_{r \overset{2}{=} 1-m}^{m-1} \sum_{s \overset{2}{=} 1-n}^{n-1} \sqrt{\left(rb + sb^{-1}\right)^2} - \frac{mn(mb + nb^{-1})}{3}\right] \ . \tag{4.51}$$

Recall that we had to assume that either $m$ or $n$ is even, since otherwise the result is divergent. One can rederive this result by combining (3.26) and (4.31).

# 5 Explicit checks of the proposal

In this section, we will check that our conjectured answer (1.6c) satisfies all the properties that we derived. In order to avoid confusions, we will denote the string amplitudes as defined through the worldsheet integral (2.19), (2.21) by $\mathsf{A}_{g,n}^{(b)}(p_1, \ldots, p_n)$ and our conjectured formulas (1.6b), (1.6c) by $\mathsf{a}_{g,n}^{(b)}(p_1, \ldots, p_n)$. Thus we want to demonstrate that

$$\mathsf{a}_{g,n}^{(b)}(\boldsymbol{p}) = \mathsf{A}_{g,n}^{(b)}(\boldsymbol{p}) , \qquad \text{for} \quad (g, n) = (0, 4), (1, 1) . \tag{5.1}$$

We do this by checking that $\mathsf{a}_{0,4}^{(b)}(\boldsymbol{p})$ and $\mathsf{a}_{1,1}^{(b)}(\boldsymbol{p})$ satisfies all the properties we derived for $\mathsf{A}_{g,n}^{(b)}(\boldsymbol{p})$. Let us also remind the reader that we already demonstrated that $\mathsf{a}_{0,3}^{(b)}(\boldsymbol{p}) = \mathsf{A}_{0,3}^{(b)}(\boldsymbol{p})$, see section 2.4.

## 5.1 Analytic continuation

The formula eq. (1.6c) only converges in a neighborhood of the physical kinematics $bp \in \mathbb{R}$. To explore the analytic structure of (1.6c), we hence first have to discuss the analytic continuation. We will thus first derive a formula that converges for any complex $p_j$ and is the analytic continuation of (1.6c). For completeness we present it here again

$$
\begin{aligned}
\mathsf{a}_{0,4}^{(b)}(\boldsymbol{p}) = &\sum_{m=1}^{\infty} \frac{2b^2 \mathsf{V}_{0,4}^{(b)}(ip_1, ip_2, ip_3, ip_4) \prod_{j=1}^{4} \sin(2\pi m b p_j)}{\sin(\pi m b^2)^2} \\
&- \sum_{m_1, m_2=1}^{\infty} \frac{(-1)^{m_1+m_2} \sin(2\pi m_1 b p_1) \sin(2\pi m_1 b p_2) \sin(2\pi m_2 b p_3) \sin(2\pi m_2 b p_4)}{\pi^2 \sin(\pi m_1 b^2) \sin(\pi m_2 b^2)} \\
&\times \left( \frac{1}{(m_1 + m_2)^2} - \frac{\delta_{m_1 \neq m_2}}{(m_1 - m_2)^2} \right) + 2 \text{ perms} .
\end{aligned}
\tag{5.2}
$$

**Single sum.** Let us first give the analytic continuation of the second term in (5.2). We have

$$
\begin{aligned}
\sum_{m=1}^{\infty} \frac{\prod_{j=1}^{4} \sin(2\pi m b p_j)}{\sin(\pi m b^2)^2} &= -\frac{1}{4} \sum_{m=1}^{\infty} \sum_{\sigma_1, \ldots, \sigma_4 = \pm} \frac{\sigma_1 \sigma_2 \sigma_3 \sigma_4 \, e^{2\pi i m b \sum_{j=1}^{4} \sigma_j p_j}}{(e^{\pi i m b^2} - e^{-\pi i m b^2})^2} \\
&= -\frac{1}{4} \sum_{m=1}^{\infty} \sum_{k=0}^{\infty} \sum_{\sigma_1, \ldots, \sigma_4 = \pm} k \sigma_1 \sigma_2 \sigma_3 \sigma_4 \, e^{2\pi i m b (\sum_{j=1}^{4} \sigma_j p_j + kb)} \\
&= -\frac{1}{4} \sum_{k=0}^{\infty} \sum_{\sigma_1, \ldots, \sigma_4 = \pm} \frac{k \sigma_1 \sigma_2 \sigma_3 \sigma_4}{e^{-2\pi i b (\sum_{j=1}^{4} \sigma_j p_j + kb)} - 1} .
\end{aligned}
\tag{5.3}
$$

Here we assumed that $\mathrm{Im}(b^2) > 0$ to expand the sines in the denominator and perform the geometric sum from the second to the third line. The resulting expression is now everywhere convergent and has various poles that we will further analyze below.

**Double sum.** Let us repeat a similar rewriting for the other terms in (5.2) involving double sums. We can focus on the first permutation. We obtain

$$\sum_{m_1,m_2=1}^{\infty} \frac{\prod_{j=1}^{2}\sin(2\pi m_1 b p_j)\prod_{j=3}^{4}\sin(2\pi m_2 b p_j)}{(-1)^{m_1+m_2}\sin(\pi m_1 b^2)\sin(\pi m_2 b^2)}\left(\frac{1}{(m_1+m_2)^2}-\frac{1-\delta_{m_1,m_2}}{(m_1-m_2)^2}\right)$$

$$= -\frac{1}{4}\sum_{m_1,m_2=1}^{\infty}\sum_{\sigma_1,\dots,\sigma_4=\pm}\frac{\sigma_1\sigma_2\sigma_3\sigma_4\,(-1)^{m_1+m_2}\,\mathrm{e}^{2\pi i b(m_1\sum_{j=1}^{2}\sigma_j p_j+m_2\sum_{j=3}^{4}\sigma_j p_j)}}{(\mathrm{e}^{\pi i m_1 b^2}-\mathrm{e}^{-\pi i m_1 b^2})(\mathrm{e}^{\pi i m_2 b^2}-\mathrm{e}^{-\pi i m_2 b^2})}$$

$$\times\left(\frac{1}{(m_1+m_2)^2}-\frac{1-\delta_{m_1,m_2}}{(m_1-m_2)^2}\right)$$

$$= -\frac{1}{4}\sum_{m_1,m_2=1}^{\infty}\sum_{k_1,k_2=0}^{\infty}\sum_{\sigma_1,\dots,\sigma_4=\pm}\sigma_1\sigma_2\sigma_3\sigma_4\,(-1)^{m_1+m_2}\,\mathrm{e}^{2\pi i b m_1(\sum_{j=1}^{2}\sigma_j p_j+(k_1+\frac{1}{2})b)}$$

$$\times\,\mathrm{e}^{2\pi i b m_2(\sum_{j=3}^{4}\sigma_j p_j+(k_2+\frac{1}{2})b)}\left(\frac{1}{(m_1+m_2)^2}-\frac{1-\delta_{m_1,m_2}}{(m_1-m_2)^2}\right)$$

$$= -\frac{1}{4}\sum_{k_1,k_2=0}^{\infty}\sum_{\sigma_1,\dots,\sigma_4=\pm}\sigma_1\sigma_2\sigma_3\sigma_4\left[\frac{\mathrm{Li}_2\!\left(-\mathrm{e}^{2\pi i b(\sum_{j=1}^{2}\sigma_j p_j+(k_1+\frac{1}{2})b)}\right)}{\mathrm{e}^{2\pi i b(\sum_{j=1}^{2}\sigma_j p_j-\sum_{j=3}^{4}\sigma_j p_j+(k_1-k_2)b)}-1}\right.$$

$$\left.-\frac{\mathrm{Li}_2\!\left(-\mathrm{e}^{2\pi i b(\sum_{j=1}^{2}\sigma_j p_j+(k_1+\frac{1}{2})b)}\right)}{\mathrm{e}^{-2\pi i b(\sum_{j=1}^{4}\sigma_j p_j+(k_1+k_2+1)b)}-1}\right]+(\{1,2\}\leftrightarrow\{3,4\})$$

$$= \frac{1}{4}\sum_{k_1,k_2=0}^{\infty}\sum_{\sigma,\sigma_1,\dots,\sigma_4=\pm}\frac{\sigma\sigma_1\sigma_2\sigma_3\sigma_4\,\mathrm{Li}_2\!\left(-\mathrm{e}^{2\pi i b(\sigma\sum_{j=1}^{2}\sigma_j p_j+(k_1+\frac{1}{2})b)}\right)}{\mathrm{e}^{-2\pi i b(\sum_{j=1}^{4}\sigma_j p_j+\sigma(k_1+\frac{1}{2})+k_2+\frac{1}{2})b)}-1}$$

$$+(\{1,2\}\leftrightarrow\{3,4\})\,.\tag{5.4}$$

We performed the sum over $m_1$ and $m_2$ by using the definition of the dilogarithm. To go to the last line, we renamed $\sigma_1\to-\sigma_1$ and $\sigma_2\to-\sigma_2$ in the first term. This sum is again absolutely convergent for arbitrary complex momenta and hence provides the desired analytic continuation. As expected, the expression has branch cuts due to the presence of the dilogarithm.

**Summary.** To summarize, we can write the conjectured four-point function (5.2) equivalently in the form (assuming $\mathrm{Im}(b^2)>0$)

$$\mathsf{a}_{0,4}^{(b)}(p_1,p_2,p_3,p_4)=-\frac{b^2}{2}\sum_{k=0}^{\infty}\sum_{\sigma_1,\dots,\sigma_4=\pm}\frac{k\sigma_1\sigma_2\sigma_3\sigma_4\,\mathsf{V}_{0,4}^{(b)}(ip_1,ip_2,ip_3,ip_4)}{\mathrm{e}^{-2\pi i b(\sum_{j=1}^{4}\sigma_j p_j+kb)}-1}$$

$$-\frac{1}{4\pi^2}\sum_{1\leqslant j<\ell\leqslant 4}\sum_{k_1,k_2=0}^{\infty}\sum_{\sigma,\sigma_1,\dots,\sigma_4=\pm}\frac{\sigma\sigma_1\sigma_2\sigma_3\sigma_4\,\mathrm{Li}_2\!\left(-\mathrm{e}^{2\pi i b(\sigma(\sigma_j p_j+\sigma_\ell p_\ell)+(k_1+\frac{1}{2})b)}\right)}{\mathrm{e}^{-2\pi i b(\sum_{j=1}^{4}\sigma_j p_j+(\sigma(k_1+\frac{1}{2})+k_2+\frac{1}{2})b)}-1}\,,\tag{5.5}$$

which in particular defines its analytic continuation to arbitrary complex momenta.

**Poles.** From the representation, the location of the poles and branch points is manifest. The single-sum term has poles when

$$\sum_{j=1}^{4} \sigma_j p_j = -kb + nb^{-1} \tag{5.6}$$

with $k \in \mathbb{Z}_{\geqslant 1}$ and $n \in \mathbb{Z}$. The double sum term has the same pole, except for the term with $\sigma = -1$ and $k_1 = k_2$ that seemingly also has a pole at $\sum_{j=1}^{4} \sigma_j p_j = nb^{-1}$. However, it is simple to see that the numerator cancels out and the residue vanishes. This was obvious from the representation (5.2) that we started with since the infinite sums are absolutely convergent for $\sum_{j=1}^{4} \sigma_j p_j = nb^{-1}$ and $p_j$ close enough to the origin. Finally, also the pole with $n = 0$ in (5.6) cancels out, which will follow directly once we establish the duality symmetry that exchanges $b \to b^{-1}$. Thus we see that $\mathsf{a}_{0,4}^{(b)}(\boldsymbol{p})$ leads to the correct poles as in (3.8).

**Branch cuts and discontinuities.** The location of the branch cuts is also easy to read off from (5.5). They clearly originate from the dilogarithm. We recognize that we can perform the sum over $k_2$ in the second term of (5.5) by recognizing the form (2.37) of the three-point function. This leads to

$$\mathsf{a}_{0,4}^{(b)}(p_1, p_2, p_3, p_4)^{(2)} = -\frac{1}{2\pi^2 b} \sum_{1 \leqslant j < \ell \leqslant 4} \sum_{k=0}^{\infty} \sum_{\sigma_j, \sigma_\ell = \pm} \sigma_j \sigma_\ell \operatorname{Li}_2\left(-\mathrm{e}^{2\pi i b(\sigma_j p_j + \sigma_\ell p_\ell + (k + \frac{1}{2})b)}\right)$$
$$\times \mathsf{a}_{0,3}^{(b)}\left(\tfrac{b+b^{-1}}{2} + \sigma_j p_j + \sigma_\ell p_\ell, p_m, p_n\right), \tag{5.7}$$

for the second term in (5.5) that captures the discontinuity. Here $\{j, \ell, m, n\} = \{1, 2, 3, 4\}$, so $m$ and $n$ are the other two indices not equal to $j$ and $\ell$.

Recalling that the dilogarithm has a branch point at $z = 1$, we see that the expression has branch points at

$$p_* = \sigma_j p_j + \sigma_\ell p_\ell + (r + \tfrac{1}{2})b + (s + \tfrac{1}{2})b^{-1} = 0 , \tag{5.8}$$

as required from our general discussion in section 3.1. The discontinuity follows directly from the discontinuity of the dilogarithm, $\operatorname{Disc}_{z=1} \operatorname{Li}_2(z) = -2\pi i \log(z)$. The discontinuity at $p_*$ is then

$$\operatorname{Disc}_{p_*=0} \mathsf{a}_{0,4}^{(b)}(\boldsymbol{p}) = 8\pi i p_* \operatorname{Res}_{p=p_*} \left(\mathsf{a}_{0,3}(p_1, p_2, p) \mathsf{a}_{0,3}(p, p_3, p_4) + 2 \text{ perms}\right) . \tag{5.9}$$

We also added the discontinuities corresponding to the other two ways of pairing up $(p_1, p_2, p_3, p_4)$. This confirms (3.12).

$\mathsf{a}_{1,1}^{(b)}$. One can of course repeat the same exercise for the one-point function on the torus, which leads to the result

$$
\mathsf{a}_{1,1}^{(b)}(p_1) = b \sum_{k=0}^{\infty} \sum_{\sigma_1 = \pm} \frac{\sigma \, \mathsf{V}_{1,1}^{(b)}(ip_1)}{e^{-2\pi i b(\sigma_1 p_1 + (k+\frac{1}{2})b)} + 1}
$$

$$
+ \frac{1}{16\pi^2 b} \sum_{k=0}^{\infty} \sum_{\sigma_1 = \pm} \sigma_1 \, \mathrm{Li}_2\left( - e^{2\pi i b(\sigma_1 p_1 + (k+\frac{1}{2})b)} \right) . \quad (5.10)
$$

One can repeat the same analysis as above and confirm that

$$
\mathop{\mathrm{Disc}}_{p_* = 0} \mathsf{a}_{1,1}^{(b)}(p_1) = 2\pi i p_* \mathop{\mathrm{Res}}_{p = \frac{1}{2} p_*} \mathsf{a}_{0,3}^{(b)}(p, p, p_1) , \quad (5.11)
$$

where $p_*$ runs over all possible poles of $\mathsf{a}_{0,3}^{(b)}(p, p, p_1)$. This matches with the prediction of (3.14).

## 5.2   Duality and swap symmetries

We next establish the three symmetries (2.24a), (2.25) and (2.24b) of the claimed solution (1.6c) as well as (1.6b).

**Simple symmetries.**   The $b \to -b$ symmetry and swap symmetry as in (2.24a) and (2.25) are essentially manifest in the proposed expressions (1.6), since all the terms transform in a simple way. Thus we will focus on the duality symmetry. We use essentially the same argument as for the three-point function that we explained in section 2.4.

$\mathsf{a}_{1,1}^{(b)}$.   Let us warm up with the one-point function on the torus. We can write it as

$$
\mathsf{a}_{1,1}^{(b)}(p_1) = \frac{i}{64\pi^2} \int_{\gamma} \mathrm{d}p \, \frac{\sin(4\pi p p_1)}{\sin(2\pi b p) \sin(2\pi b^{-1} p)} \left( \frac{1}{p^2} + \frac{8\pi^2}{3} \left( p_1^2 - \frac{b^2 + b^{-2}}{4} \right) \right) . \quad (5.12)
$$

The contour is the same as in figure 1. The integrand has the property that the residue at $p = 0$ vanishes. As in the corresponding formula for the three-point function (2.36), the integrand is antisymmetric in $p$ and we may hence freely change the integration contour to go around any of the four quadrants in figure 1. This integral representation hence makes the symmetry under exchange $b \leftrightarrow b^{-1}$ essentially manifest.

$\mathsf{a}_{0,4}^{(b)}$.   Let us next repeat the same logic for $\mathsf{a}_{0,4}^{(b)}$. This requires a double contour integral. Let us note that we can write

$$
\mathsf{a}_{0,4}^{(b)}(\boldsymbol{p}) = \sum_{m=-\infty}^{\infty} \mathop{\mathrm{Res}}_{p=\frac{mb}{2}} f_b(p) + \sum_{m_2=-\infty}^{\infty} \sum_{\substack{m_1=-\infty, \\ m_1 \neq m_2}}^{\infty} \mathop{\mathrm{Res}}_{p=\frac{m_1 b}{2}} \mathop{\mathrm{Res}}_{p'=\frac{m_2 b}{2}} g_b(p, p') , \quad (5.13)
$$

where

$$
f_b(p) = \frac{2\pi b \, \mathsf{V}_{0,4}^{(b)}(i\boldsymbol{p}) \cos(2\pi b^{-1} p) \prod_{j=1}^{4} \sin(4\pi p p_j)}{\sin(2\pi b p)^2 \sin(2\pi b^{-1} p)} , \quad (5.14a)
$$

$$g_b(p, p') = -\frac{\sin(4\pi p p_1)\sin(4\pi p p_2)\sin(4\pi p' p_3)\sin(4\pi p' p_4)}{2\sin(2\pi bp)\sin(2\pi b^{-1}p)\sin(2\pi bp')\sin(2\pi bp')(p-p')^2} + 2 \text{ perms .}$$

$$(5.14b)$$

We now use that the sum over all residues vanishes to rewrite this infinite sum as a sum over all the other residues. We obtain

$$\mathsf{a}_{0,4}^{(b)}(\boldsymbol{p}) = \sum_{m=-\infty}^{\infty} \underset{p=\frac{mb}{2}}{\text{Res}}\, f_b(p) + \sum_{m_2=-\infty}^{\infty}\sum_{\substack{m_1=-\infty,\\m_1\neq m_2}}^{\infty} \underset{p=\frac{m_1}{2b}}{\text{Res}}\,\underset{p'=\frac{m_2}{2b}}{\text{Res}}\, g_b(p, p')$$

$$- \sum_{m=-\infty}^{\infty} \underset{p=\frac{mb}{2}}{\text{Res}}\,\underset{p'=\frac{mb}{2}}{\text{Res}}\, g_b(p, p') + \sum_{m=-\infty}^{\infty} \underset{p'=\frac{m}{2b}}{\text{Res}}\,\underset{p=\frac{m}{2b}}{\text{Res}}\, g_b(p, p') \qquad (5.15)$$

$$= \mathsf{a}_{0,4}^{(b^{-1})}(\boldsymbol{p}) + \sum_{m=-\infty}^{\infty}\left[ \underset{p'=\frac{m}{2b}}{\text{Res}}\,\underset{p=\frac{m}{2b}}{\text{Res}}\, g_b(p, p') - \underset{p=\frac{mb}{2}}{\text{Res}}\,\underset{p'=\frac{mb}{2}}{\text{Res}}\, g_b(p, p') \right.$$

$$\left. - \underset{p=\frac{m}{2b}}{\text{Res}}\, f_{b^{-1}}(p) + \underset{p=\frac{mb}{2}}{\text{Res}}\, f_b(p) \right] \qquad (5.16)$$

After symmetrizing the residue in $p$ and $p'$, one can check that

$$\underset{p=\frac{mb}{2}}{\text{Res}}\, f_b(p) - \frac{1}{2}\left( \underset{p=\frac{mb}{2}}{\text{Res}}\,\underset{p'=\frac{mb}{2}}{\text{Res}} + \underset{p'=\frac{mb}{2}}{\text{Res}}\,\underset{p=\frac{mb}{2}}{\text{Res}} \right) g_b(p, p')$$

$$= \underset{p=\frac{mb}{2}}{\text{Res}}\, \frac{\frac{\pi}{2}\left(b\cot(2\pi bp) - b^{-1}\cot(2\pi b^{-1}p)\right)\prod_{j=1}^{4}\sin(4\pi p p_j)}{\sin(2\pi bp)^2 \sin(2\pi b^{-1}p)^2} , \quad (5.17)$$

which shows that the remaining terms cancel since the sum over the residues of this function vanishes.

## 5.3 Triality symmetry

Let us next establish triality symmetry as discussed in section 3.3. We give the idea how to show this. If we pretend again for a moment that we have a gauge theory at our hand, we could write

$$\mathsf{a}_{0,4}^{(b)}(p_1, p_2, p_3, p_4) \sim \int (-2p\,\mathrm{d}p)\, \mathsf{a}_{0,3}^{(b)}(p_1, p_2, p)\mathsf{a}_{0,3}^{(b)}(p_3, p_4, p) , \qquad (5.18)$$

but as remarked before, this integral is divergent. Let us proceed anyway. Using the theta-function representation of the three-point function (2.31), we see that this integral is of the form

$$\mathsf{A}_{0,4}^{(b)\prime}(p_1, p_2, p_3, p_4) \sim \int (-2p\,\mathrm{d}p)\, \frac{\mathrm{e}^{2\pi i \sum_{j=1}^{4} p_j^2}\left(ib\,\eta(b^2)^3\vartheta_1(2bp|b^2)\right)^2}{4\vartheta_3(bp \pm bp_1 \pm bp_2|b^2)\vartheta_3(bp \pm bp_3 \pm bp_4|b^2)} , \qquad (5.19)$$

where we stripped of the same factor as in (3.23). The numerator is not relevant here, but we notice that the form of the denominator makes the desired triality symmetry

manifest. While the actual four-point function (1.6c) is not literally obtained by integrating a product of three-point functions, it is close enough and this argument can be made rigorous. We spell out the details in appendix C.2.

## 5.4   Higher equations of motion

Let us next demonstrate that $\mathsf{a}_{0,4}^{(b)}$ and $\mathsf{a}_{1,1}^{(b)}$ as given by (1.6c) and (1.6b) satisfy the higher equations of motion (4.31) and (4.51). For $\mathsf{a}_{0,4}^{(b)}$, this implies in particular that the dilaton equation (3.16) is satisfied.

$\mathbf{\mathsf{a}_{0,4}^{(b)}}$.   We use the form (5.5) since we will need $\mathsf{a}_{0,4}^{(b)}$ outside the region of convergence of (1.6c). Let us recall that the higher equations of motion are somewhat ambiguous since we arbitrarily chose a branch of the function $\sqrt{p^2}$ for most of the terms. Thus we will allow ourselves to be somewhat liberal in the choice of branches in our verification.

The non-analytic terms come from the dilogarithm in the second term of (5.5), for which we use the form (5.7). Since the three-point function with a degenerate argument vanishes, only terms with $\ell = 4$ can be non-zero. Thus

$$\mathsf{a}_{0,4}^{(b)}(p_1,p_2,p_3,\tfrac{mb}{2}-\tfrac{n}{2b})^{(2)} = -\sum_{j=1}^{3}\sum_{k=0}^{\infty}\sum_{\sigma_j,\sigma_4=\pm}\frac{\sigma_j\sigma_4}{2\pi^2 b}\,\mathrm{Li}_2\big(-\mathrm{e}^{2\pi ib(\sigma_j p_j+\sigma_4(\frac{mb}{2}-\frac{n}{2b})+(k+\frac{1}{2})b)}\big)$$
$$\times\,\mathsf{a}_{0,3}^{(b)}\big(\tfrac{b+b^{-1}}{2}+\sigma_j p_j+\sigma_4\big(\tfrac{mb}{2}-\tfrac{n}{2b}\big),p_{j'},p_{j''}\big)\,,\quad (5.20)$$

where $\{j,j',j''\} = \{1,2,3\}$. We can use ellipticity of the three-point function to simplify

$$\mathsf{a}_{0,3}^{(b)}\big(\tfrac{b+b^{-1}}{2}+\sigma_j p_j+\sigma_4\big(\tfrac{mb}{2}-\tfrac{n}{2b}\big),p_{j'},p_{j''}\big) = \sigma_j\mathsf{a}_{0,3}^{(b)}\big(p_1+\tfrac{(m-1)b}{2}+\tfrac{n-1}{2b},p_2,p_3\big)\quad (5.21)$$

regardless of the value of $j$. We see that the sum over $k$ is effectively shifted by $\frac{m}{2}$ up or down, depending on the sign of $\sigma_4$. Because of the overall $\sigma_4$ this means that most terms cancel out identically and we obtain

$$\mathsf{a}_{0,4}^{(b)}(p_1,p_2,p_3,\tfrac{mb}{2}-\tfrac{n}{2b})^{(2)} = \frac{1}{2\pi^2 b}\sum_{j=1}^{3}\sum_{r=1-m}^{m-1}\sum_{\sigma_j=\pm}\mathrm{Li}_2\big(-\mathrm{e}^{2\pi ib\sigma_j(p_j+\frac{rb}{2})+\pi in}\big)$$
$$\times\,\mathsf{a}_{0,3}^{(b)}\big(p_1+\tfrac{(m-1)b}{2}+\tfrac{n-1}{2b},p_2,p_3\big)\,,\quad (5.22)$$

To understand how to continue, let us first consider the case of the dilaton equation where $m = n = 1$. We have to compute the following sum over dilogarithms:

$$\mathrm{Li}_2(\mathrm{e}^{2\pi ibp}) + \mathrm{Li}_2(\mathrm{e}^{-2\pi ibp}) = -\frac{\pi^2}{6} - \frac{1}{2}\big(\log(-\mathrm{e}^{2\pi ibp})\big)^2\,.\quad (5.23)$$

Here we used a standard identity of the dilogarithm. This can clearly be simplified, but we have to be careful about the choice of branch of the logarithm. To motivate

the correct choice for the dilaton equation, imagine that $p$ is small, so that $-\mathrm{e}^{2\pi ibp}$ is close to $-1$. For $\mathrm{Re}(p^2) > 0$, the principal branch choice of the logarithm gives

$$-\frac{\pi^2}{6} - \frac{1}{2}\big(\log(-\mathrm{e}^{2\pi ibp})\big)^2 = \frac{\pi^2}{3} + 2\pi^2 b^2 p^2 - 2\pi^2 b\sqrt{p^2} , \tag{5.24}$$

while for other values of $p$ we assume that this result holds by analytic continuation.

Similarly, (5.22) can be simplified further for general $m$ and $n$. For higher values of $n$, the correct branch choice for the logarithm is instead

$$\mathrm{Li}_2((-1)^{n-1}\mathrm{e}^{2\pi ibp}) + \mathrm{Li}_2((-1)^{n-1}\mathrm{e}^{-2\pi ibp})$$
$$= \frac{3n^2\pi^2 - \pi^2}{6} + 2\pi^2 b^2 p^2 - 2\pi^2 b \sum_{s\overset{2}{=}1-n}^{n-1} \sqrt{(p + \tfrac{s}{2b})^2} . \tag{5.25}$$

The reason for this choice is that the right hand side equals (5.23) as long as $bp$ is at most of order $\frac{n}{2}$. It however is sensitive to the existence of all the branch cuts at $\frac{s}{2b}$. Using this in eq. (5.22) yields

$$\mathsf{a}_{0,4}^{(b)}(p_1, p_2, p_3, \tfrac{mb}{2} - \tfrac{n}{2b})^{(2)} = \frac{1}{2\pi^2 b} \sum_{j=1}^{3} \sum_{r\overset{2}{=}1-m}^{m-1} \left(\frac{3n^2\pi^2 - \pi^2}{6} + 2\pi^2 b^2\big(p_j + \tfrac{rb}{2}\big)^2\right.$$
$$\left. - 2\pi^2 b \sum_{s\overset{2}{=}1-n}^{n-1} \sqrt{\big(p_j + \frac{rb}{2} + \frac{s}{2b}\big)^2}\right) \mathsf{a}_{0,3}^{(b)}\big(p_1 + \tfrac{(m-1)b}{2} + \tfrac{n-1}{2b}, p_2, p_3\big) . \tag{5.26}$$

The contribution of the first term in (5.5) when $p_4 = \frac{mb}{2} - \frac{n}{2b}$ is simpler to work out since it does not have branch cuts. We obtain

$$\mathsf{a}_{0,4}^{(b)}(p_1, p_2, p_3, \tfrac{mb}{2} - \tfrac{n}{2b})^{(1)} = bm\mathsf{V}_{0,4}^{(b)}(ip_1, ip_2, ip_3, i(\tfrac{mb}{2} - \tfrac{n}{2b}))\mathsf{a}_{0,3}^{(b)}\big(p_1 + \tfrac{(m-1)b}{2} + \tfrac{n-1}{2b}, p_2, p_3\big) . \tag{5.27}$$

Taking (5.26) and (5.27) together gives

$$\mathsf{A}_{0,4}^{(b)}(p_1, p_2, p_3, \tfrac{mb}{2} - \tfrac{n}{2b}) = \mathsf{A}_{0,3}^{(b)}(p_1 + \tfrac{(m-1)b}{2} + \tfrac{n-1}{2b}, p_2, p_3)$$
$$\times \left[\frac{mn(mb + nb^{-1})}{2} - \sum_{i=1}^{3} \sum_{r\overset{2}{=}1-m}^{m-1} \sum_{s\overset{2}{=}1-n}^{n-1} \sqrt{\big(p_i + \frac{rb}{2} + \frac{s}{2b}\big)^2}\right] . \tag{5.28}$$

This matches indeed with (4.31).

$\mathsf{a}_{1,1}^{(b)}$. The computation is essentially identical. Starting from (5.10) and applying the same logic gives

$$\mathsf{a}_{1,1}^{(b)}(p_1 = \tfrac{mb}{2} - \tfrac{n}{2b}) = \frac{1}{32}\left[\sum_{r\overset{2}{=}1-m}^{m-1} \sum_{s\overset{2}{=}1-n}^{n-1} \sqrt{(rb + sb^{-1})^2} - \frac{mn(mb + nb^{-1})}{3}\right] . \tag{5.29}$$

## 5.5 Relation between $\mathsf{a}_{0,4}^{(b)}$ and $\mathsf{a}_{1,1}^{(b)}$

Finally, we verify that (3.26) is satisfied by the explicit functions (1.6b) and (1.6c). For this it is convenient to use the expressions (5.5) and (5.10). The two terms satisfy the relation (3.26) separately. We show this in three steps. We first show that the discontinuous parts of the left- and right-hand side of (3.26) agree (which is easy). The hard step is to show that the residues of the poles agree. This shows that the left-hand side and the right-hand side differ by at most a polynomial, which in turn has to vanish because of the trivial zeros of the amplitude. We spell out the details in appendix C.3.

## 5.6 Uniqueness

One may ask whether all the constraints that we derived admit (1.6c) and (1.6b) as unique solutions. We will discuss this for $\mathsf{A}_{0,4}^{(b)}$. We can look at the difference

$$\mathsf{B}_{0,4}^{(b)}(\boldsymbol{p}) \equiv \mathsf{A}_{0,4}^{(b)}(\boldsymbol{p}) - \mathsf{a}_{0,4}^{(b)}(\boldsymbol{p}) \ . \tag{5.30}$$

$\mathsf{B}_{0,4}^{(b)}(\boldsymbol{p})$ has the following properties:

1. $\mathsf{B}_{0,4}^{(b)}$ is an entire function in the momenta $\boldsymbol{p}$. Indeed, we have shown that both functions have identical discontinuities and poles and thus the difference is meromorphic. The poles also cancel since as remarked after eq. (4.31), the residues are fixed by triality symmetry and the higher equations of motion, which are satisfied by both $\mathsf{A}_{0,4}^{(b)}$ and $\mathsf{a}_{0,4}^{(b)}$. Thus they cancel in the difference.

2. $\mathsf{B}_{0,4}^{(b)}$ vanishes when $p_j = \frac{rb}{2} + \frac{s}{2b}$ for any $r$, $s \in \mathbb{Z}$ and any $j = 1, \ldots, 4$. This follows from the trivial zeros, together with the higher equations of motion.

3. $\mathsf{B}_{0,4}^{(b)}$ is triality symmetric in the sense of (3.25).

4. $\mathsf{B}_{0,4}^{(b)}$ is permutation symmetric and odd in all momenta and satisfies the various dualities (2.24a), (2.24b) and (2.25).

Functions satisfying these constraints exist. To construct them, it is more useful to look at the triality symmetric normalization (3.23)

$$\mathsf{B}_{0,4}^{(b)\prime}(p_1, p_2, p_3, p_4) = \prod_{j=1}^{4} \frac{\mathrm{e}^{-2\pi i p_j^2}}{\vartheta_1(2bp_j, b^2)} \, \mathsf{B}_{0,4}^{(b)}(p_1, p_2, p_3, p_4) \ . \tag{5.31}$$

The theta-function cancels all the zeros of $\mathsf{B}_{0,4}^{(b)}(\boldsymbol{p})$, while the Gaussian is necessary to preserve the $b$-duality symmetry. Thus $\mathsf{B}_{0,4}^{(b)\prime}(p_1, p_2, p_3, p_4)$ could be *any* even entire function in the momenta that is invariant under the discrete symmetries. The simplest choice is clearly 1, but we could choose e.g. also $\mathsf{V}_{0,4}^{(b)}(ip_1, ip_2, ip_3, ip_4)$.

**Growth at infinity.**   However, there is a good reason to suspect that such solutions are not allowed. Such functions grow *much* faster at infinity than the solution that we found. Indeed, due to the quasi-periodicity of the theta-function, we have (assuming $b \in e^{\frac{\pi i}{4}} \mathbb{R}_{\geqslant 0}$)

$$\mathsf{B}_{0,4}^{(b)}(p_1, p_2, p_3, p_4) \sim \mathcal{O}(1) \, e^{2\pi \sum_{j=1}^{4} |p_j|^2} \, , \tag{5.32}$$

where $\mathcal{O}(1)$ is at best constant when we choose $\mathsf{B}_{0,4}^{(b)\prime}(p_1, p_2, p_3, p_4) = 1$. This is hence very rapidly growing near infinity.

In contrast, our proposed solution $\hat{\mathsf{A}}_{0,4}^{(b)}(\boldsymbol{p})$ given by (1.6c) grows only polynomially fast. We can see this by noticing that, for $i \neq j$

$$\Delta_b^i \Delta_b^j \hat{\mathsf{A}}_{0,4}^{(b)}(\boldsymbol{p}) = (\Delta_b^i)^3 \hat{\mathsf{A}}_{0,4}^{(b)}(\boldsymbol{p}) = 0 \, . \tag{5.33}$$

Here $\Delta_b^i$ is the discrete difference operator in the $i$-th momentum, for example

$$\Delta_b^1 f(p_1, \dots) = f(p_1 + \tfrac{b}{2}, \dots) - f(p_1 - \tfrac{b}{2}, \dots) \, . \tag{5.34}$$

(5.33) holds of course also with $b$ replaced by $\frac{1}{b}$. This implies recursively that $\hat{\mathsf{A}}_{0,4}^{(b)}$ grows like a quadratic polynomial near infinity (as long as we avoid the poles).

**Generalized ellipticity.**   It seems very natural to us to assume (5.33). Indeed, this is the natural analogue of the ellipticity of the three-point function that we observed. In particular, the three-point function does not grow at infinity. Thus it seems very unnatural to us that the four-point function should grow as fast as (5.32). Under this assumption, our bootstrap problem has then a unique solution given by (1.6c). It seems possible to us that one can fill this gap and turn our arguments into a complete proof, but we have not managed to do so.[19]

# 6   Direct numerical evaluation of $\mathsf{A}_{0,4}^{(b)}$

In this section, we present results for the direct numerical evaluation of the sphere four-point amplitude $\mathsf{A}_{0,4}^{(b)}(p_1, p_2, p_3, p_4)$ providing further strong evidence for the analytic expression (1.6c). We will follow a similar strategy of numerical integration as described in [5, 13, 69]. We restrict our attention to this case, since $\mathsf{A}_{1,1}^{(b)}(p_1)$ follows from this computation via the relation (3.26).

## 6.1   Method

As explained in [70, 71], we may use crossing symmetry of the Liouville CFT four-point functions in order to restrict the moduli integration from $z \in \mathbb{C}$ to the compact

---

[19]We notice also that the property (5.33) also holds true in the Virasoro minimal string [13], where the analogues of the amplitudes $\mathsf{A}_{g,n}^{(b)}$ are just polynomials of order $2(3g - 3 + n)$ in $p_j$.

region $R = \{z \in \mathbb{C} \mid |1 - z| \leqslant 1, \text{Re}(z) \leqslant \frac{1}{2}\}$. Thus, the sphere four-point takes the form,

$$\mathsf{A}_{0,4}^{(b)}(p_1, p_2, p_3, p_4) \equiv C_{\mathrm{S}^2}^{(b)} \prod_{j=1}^{4} \mathcal{N}_b(p_j) \int_R \mathrm{d}^2 z \, \left| \langle V_{p_1}^+(0) V_{p_2}^+(z) V_{p_3}^+(1) V_{p_4}^+(\infty) \rangle \right|^2$$
$$+ \big(5 \text{ perms. of } \{1,2,3\}\big). \quad (6.1)$$

where the correlator is given in (2.18) and the leg factors and the normalization are given in (2.20a) and (2.20b). The five permutations correspond to the integral over the complement of $R$.

It is more convenient [13] to perform a change of variables from the cross-ratio $z$ to upper half plane, via

$$t = i \frac{K(1 - z)}{K(z)}, \quad \text{where} \quad K(z) = {}_2F_1(\tfrac{1}{2}, \tfrac{1}{2}, 1|z), \quad z = \left( \frac{\vartheta_2(t)}{\vartheta_3(t)} \right)^4. \quad (6.2)$$

With this change of variables the region $R \subset \mathbb{C}$ is mapped to the fundamental domain of the complex $t$-plane, $F_0 = \{t \in \mathbb{C} \mid |t| \geqslant 1, |\text{Re}(t)| \leqslant \frac{1}{2}\}$. Furthermore, in terms of elliptic Virasoro conformal blocks $\mathcal{H}_{0,4}^{(b)}(p_j; p|q)$ defined through the elliptic nome $q = \mathrm{e}^{\pi i t}$ by

$$\mathcal{F}_{0,4}^{(b)}(p_j; p|q) = (16q)^{-p^2} \vartheta_2(t)^{-Q^2 + 4p_1^2 + 4p_2^2} \vartheta_4(t)^{-Q^2 + 4p_2^2 + 4p_3^2} \vartheta_3(t)^{Q^2 - 4p_2^2 + 4p_4^2} \mathcal{H}_{0,4}^{(b)}(p_j; p|q),$$
$$(6.3)$$

the sphere four point amplitude takes the slightly simpler form,

$$\mathsf{A}_{0,4}^{(b)}(p_1, p_2, p_3, p_4) = \pi^2 C_{\mathrm{S}^2}^{(b)} \prod_{j=1}^{4} \mathcal{N}_b(p_j) \int_{F_0} \mathrm{d}^2 t \times \int_{i\mathbb{R}_{\geqslant 0}} \mathrm{d}p^+ \int_{i\mathbb{R}_{\geqslant 0}} \mathrm{d}p^- \prod_{\sigma = \pm} \rho_{b^\sigma}(p^\sigma)$$
$$\times C_{b^\sigma}(p_1^\sigma, p_2^\sigma, p^\sigma) C_{b^\sigma}(p_3^\sigma, p_4^\sigma, p^\sigma) |16q|^{-2(p^\sigma)^2} \mathcal{H}_{0,4}^{(b^\sigma)}(p_j^\sigma; p^\sigma|q) \mathcal{H}_{0,4}^{(b^\sigma)}(p_j^\sigma; p^\sigma|\bar{q})$$
$$+ \big(5 \text{ perms. of } \{1,2,3\}\big), \quad (6.4)$$

where we recall that we write $p_j^+ = p_j$ and $b^+ = b$, with $p_j^- = ip_j$ and $b^- = -ib$.

The elliptic conformal blocks $\mathcal{H}_{0,4}^{(b)}(p_j; p|q)$ admit a series expansion in powers of $q = \mathrm{e}^{i\pi t}$, which starts at one, and whose coefficients may be computed with Zamolodchikov's elliptic recursion relation [72, 73] (see for example, [13, appendix C.2] whose conventions we follow in this paper).

With the string four-point amplitude written in the form (6.4), we can proceed with the following strategy for numerical integration. First, we divide the fundamental domain $F_0$ into two regions: (I) $t \in F_0$ with $t_2 \leqslant t_2^{\max}$, where we write $t = t_1 + it_2$. In this region I, we numerically integrate (6.4) directly. (II) $t \in F_0$ with $t_2 \geqslant t_2^{\max}$. In this region II, for sufficiently large $t_2^{\max}$ we can approximate the elliptic conformal blocks by its leading term $\mathcal{H}_{0,4}^{(b)}(p_j; p|q) \simeq 1$. We then switch the order of integrations, first performing the integral over region II of the complex $t$-plane analytically, and

then performing the remaining integrals over the intermediate Liouville momenta $p^+$ and $p^-$ numerically[20]. For the numerical results presented below, we used a cutoff of $t_2^{\max} = 5$.

## 6.2 Results

For the direct numerical evaluation of the four-point string diagram (6.4) we will make the following choices for the external momenta of the asymptotic closed string states and for the Liouville parameter $b$,

$$(a) \quad b = \frac{e}{\pi} e^{\frac{i\pi}{4}} \ , \ \{p_1, p_2, p_3, p_4\} = \left\{\tfrac{1}{3}, |p_2|, \tfrac{1}{7}, \tfrac{1}{4}\right\} \times e^{-\frac{i\pi}{4}} \ , \ |p_2| \in \left(0, \tfrac{1}{2}\right) , \tag{6.5a}$$

$$(b) \quad b = \frac{e}{\pi} e^{\frac{i3\pi}{13}} \ , \ \{p_1, p_2, p_3, p_4\} = \left\{\tfrac{1}{3}e^{-\frac{i\pi}{7}}, |p_2|e^{-\frac{i\pi}{6}}, \tfrac{1}{7}e^{-\frac{i\pi}{5}}, \tfrac{1}{4}e^{-\frac{i\pi}{4}}\right\} \ , \ |p_2| \in \left(0, \tfrac{1}{2}\right) . \tag{6.5b}$$

Note that choice (6.5a) corresponds to the case in which the moduli integrand of the sphere four-point amplitude is simply the absolute value squared of a single copy of Liouville CFT with parameter $b$. On the other hand, choice (6.5b) assigns generic phases for the Liouville parameter $b$ and for the external Liouville momenta $p_j$, and therefore the amplitude is complex-valued.

Figure 6 shows the numerical results for the four-point sphere string diagram (6.4) for the choice of external closed string momenta (6.5), computed with the strategy outlined above. We find that the numerical results demonstrate a remarkable agreement with the exact form for the string four-point diagram (1.6c). The largest discrepancy between the numerical results in the data sets (6.5) and the exact result (1.6c) is of order $10^{-5}\,\%$.

## Acknowledgements

We would like to thank Aleksandr Artemev, Mattia Biancotto, Matthias R. Gaberdiel, Davide Gaiotto, Juan Maldacena, Jörg Teschner, Herman Verlinde, Edward Witten and Xi Yin for discussions. We would also like to thank Aleksandr Artemev for thoroughly reading a first version of this draft and many helpful commments. SC, LE and BM thank l'Institut Pascal at Université Paris-Saclay, with the support of the program "Investissements d'avenir" ANR-11-IDEX-0003-01, and SC and VAR thank the Kavli Institute for Theoretical Physics (KITP), which is supported in part by grant NSF PHY-2309135, for hospitality during the course of this work. VAR is supported in part by the Simons Foundation Grant No. 488653, by the Future Faculty in the Physical Sciences Fellowship at Princeton University, and a

---

[20]Note that this strategy of integration in region II yields more accurate results compared to those of [13], whose strategy in this region II utilized a saddle point approximation that resulted in an asymptotic expansion in $(t_2^{\max})^{-1}$.

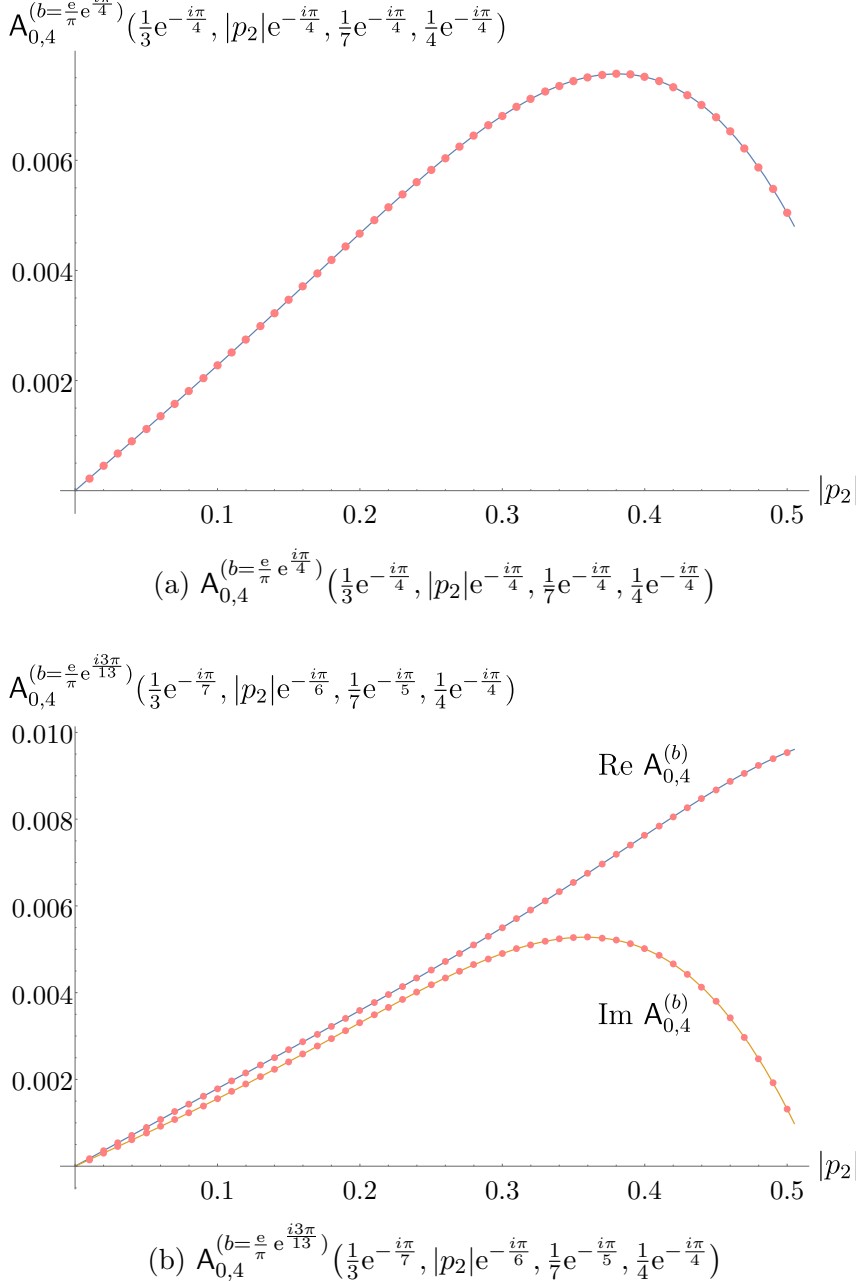

(a) $A_{0,4}^{(b=\frac{e}{\pi}e^{\frac{i\pi}{4}})}\left(\frac{1}{3}e^{-\frac{i\pi}{4}},|p_2|e^{-\frac{i\pi}{4}},\frac{1}{7}e^{-\frac{i\pi}{4}},\frac{1}{4}e^{-\frac{i\pi}{4}}\right)$

(b) $A_{0,4}^{(b=\frac{e}{\pi}e^{\frac{i3\pi}{13}})}\left(\frac{1}{3}e^{-\frac{i\pi}{7}},|p_2|e^{-\frac{i\pi}{6}},\frac{1}{7}e^{-\frac{i\pi}{5}},\frac{1}{4}e^{-\frac{i\pi}{4}}\right)$

**Figure 6**: Shown in dots are the numerical results for the four-point string diagram (6.4) in the complex Liouville string theory with the choices (6.5) for the external momenta of the asymptotic closed string states. The exact result (1.6c) is shown in the solid curve.

DeBenedictis Postdoctoral Fellowship and funds from UCSB. SC is supported by the U.S. Department of Energy, Office of Science, Office of High Energy Physics of U.S. Department of Energy under grant Contract Number DE-SC0012567 (High Energy Theory research), DOE Early Career Award DE-SC0021886 and the Packard Foun-

dation Award in Quantum Black Holes and Quantum Computation. BM gratefully acknowledges funding provided by the Sivian Fund at the Institute for Advanced Study and the National Science Foundation.

# A    More details on the worldsheet theory

The worldsheet theory under consideration is an interesting theory that pushes the boundaries of the usual CFT axioms in interesting ways. We will now explain in more detail the implication of the perhaps unusual reality conditions.

## A.1    The global generators

Let us begin by considering the left-moving global generators

$$L^+_{\pm 1,0} , \qquad L^-_{\pm 1,0} . \tag{A.1}$$

They satisfy the reality condition $(L^+_m)^\dagger = L^-_{-m}$, which defines an $\mathfrak{sl}(2,\mathbb{C})$ algebra. The global conformal group is $\text{PSL}(2,\mathbb{C})$ – its generators are given by the diagonal combination of the Virasoro generators, $L_m = L^+_m + L^-_m$ and their reality conditions are the standard ones as in any CFT. Together with the right-movers, the theory hence has a global $\text{PSL}(2,\mathbb{C}) \times \text{PSL}(2,\mathbb{C})$ symmetry in *Lorentzian* signature which extends the $\text{PSL}(2,\mathbb{R}) \times \text{PSL}(2,\mathbb{R})$ global conformal symmetry.

**Some $\text{PSL}(2,\mathbb{C})$ representation theory.** To continue, we have to recall some $\text{PSL}(2,\mathbb{C})$ representation theory. The group $\text{PSL}(2,\mathbb{C})$ admits three classes of representations: principal series, complementary series and finite-dimensional representations. $\text{PSL}(2,\mathbb{C})$ acts on the complex plane by fractional linear transformations. We can hence define the action

$$f(z) \longmapsto (cz+d)^{-2j}\overline{(cz+d)}^{-2\tilde{j}} f\left(\frac{az+b}{cz+d}\right) \tag{A.2}$$

on a function $f$ on the complex plane. Here $j$ and $\tilde{j}$ are the two spins characterizing the representations. Well-definedness of the representation imposes $j - \tilde{j} \in \mathbb{Z}$. For $j + \tilde{j} \in 1 + i\mathbb{R}$, the representation is unitary with respect to the standard $\text{L}^2$-inner product, which is the principal series. For $\frac{1}{2} < j = \tilde{j} < 1$, the representation is also unitary with respect to a modified inner product, which is the complementary series. For $j = -\frac{n}{2} \in \frac{1}{2}\mathbb{Z}_{\leqslant 0}$ and $\tilde{j} = -\frac{\tilde{n}}{2} \in \frac{1}{2}\mathbb{Z}_{\leqslant 0}$, the representation can be truncated to polynomials of degree $n$ in $z$ and degree $\tilde{n}$ in $\bar{z}$, which gives the finite-dimensional representations.

Notably, contrary to $\text{PSL}(2,\mathbb{R})$, $\text{PSL}(2,\mathbb{C})$ does not possess discrete series representation. This is obvious from the fact that the Weyl reflection is an inner automorphism of $\text{PSL}(2,\mathbb{C})$ but the discrete series is not invariant under Weyl reflection.

Thus if we take the theory at face value in Lorentzian signature, we run into immediate trouble. Since $\text{PSL}(2,\mathbb{C})$ does not possess highest weight representations, there cannot be primary states in the spectrum.

**Wick rotation.** We think of the worldsheet theory as a 2d theory of gravity of which we want to compute Euclidean gravity partition functions. We are thus interested in the theory in worldsheet Euclidean signature. As in any CFT, Wick rotating the theory does not change the reality conditions on the conformal generators since they act on the Hilbert space which is unaffected by the Wick rotation. However, it *does* change how they act on the fields. In an ordinary Wick rotated 2d CFT, $L_{-1}$ acts as $\partial_z$, while $\tilde{L}_{-1}$ acts as $\partial_{\bar{z}}$. Even though $L_{-1}$ and $\tilde{L}_{-1}$ are not related by the hermitian adjoint, $\partial_z$ and $\partial_{\bar{z}}$ are complex conjugates.[21] Such a Wick rotation cannot work in the standard way in this CFT, since we already have an $\mathrm{PSL}(2,\mathbb{C}) \times \mathrm{PSL}(2,\mathbb{C})$ Moebius symmetry in Lorentzian signature. The only way to do this is to perform a *double* Wick rotation, which redefines

$$\mathrm{PSL}(2,\mathbb{C}) \times \mathrm{PSL}(2,\mathbb{C}) \to \mathrm{PSL}(2,\mathbb{R})^4 \to \mathrm{PSL}(2,\mathbb{C}) \times \mathrm{PSL}(2,\mathbb{C}) \ . \tag{A.3}$$

We can think of the first operation as a Wick rotation in target space. The two copies of $\mathrm{PSL}(2,\mathbb{C})$ that we end up with are different than the ones above and are generated by $L_n^+$, $\tilde{L}_n^+$, and $L_n^-$, $\tilde{L}_n^-$, respectively.

This means that we can get away by defining the action of $L_n^+$ and $L_n^-$ as usual in product of two 2d CFTs. For a factorized vertex operator of the form (2.14), we can hence let $L_{-1}^+$ as a $z$-derivative on the first factor and $L_{-1}^-$ as a $z$-derivative on the second factor. This ensures in particular that $L_{-1}$ acts as $\partial_z$ on the total expression. This means also that we *can* define primary states and primary vertex operators as usual since these concepts do not make use of the hermitian adjoint of the Virasoro algebra.

## A.2 Unitarity and no-ghost theorem

An obviously interesting question is whether this theory is unitary with respect to this modified inner product. Unsurprising it is only unitary after passing to the BRST cohomology of the worldsheet theory.

**Unitarity.** Consider a highest weight state $|h^+, h^-\rangle$. Since $(L_0^+)^\dagger = L_0^-$, we have $(h^+)^* = h^-$ and thus we label the highest weight state by $|h\rangle \equiv |h^+ = h, h^- = h^*\rangle$. The level-1 Kac-matrix reads

$$M_1 = \begin{pmatrix} \langle h|L_1^- L_{-1}^+|h\rangle & \langle h|L_1^- L_{-1}^-|h\rangle \\ \langle h|L_1^+ L_{-1}^+|h\rangle & \langle h|L_1^+ L_{-1}^-|h\rangle \end{pmatrix} = \begin{pmatrix} 0 & 2h^* \\ 2h & 0 \end{pmatrix} \ . \tag{A.4}$$

The eigenvalues are $2|h|$ and $-2|h|$ and thus there is one positive norm state and one negative norm state. This is as we would like it to be in string theory, since it

---

[21]The reality conditions that would truly come from Euclidean signature would relate the hermitian adjoint of a left-moving mode to a right-moving mode and read indeed $L_n^\dagger = -\tilde{L}_n$. This is only compatible with the conformal algebra provided that $c^* = -\tilde{c}$, where $c$ is the central charge of the left-moving algebra and $\tilde{c}$ the central charge of the right-moving algebra. Such a reality condition is relevant e.g. for the dual CFT description of dS$_3$.

means that in a target space interpretation, there is one spatial dimension and one temporal dimension. Similarly, the level-2 Kac-Matrix is obtained by computing the outer product of the five states

$$L_{-2}^+|h\rangle \ , \quad L_{-1}^+ L_{-1}^+|h\rangle \ , \quad L_{-2}^-|h\rangle \ , \quad L_{-1}^- L_{-1}^-|h\rangle \ , \quad L_{-1}^+ L_{-1}^-|h\rangle \ . \tag{A.5}$$

It reads

$$M_2 = \begin{pmatrix} 0 & 0 & \frac{c^*}{2}+4h^* & 6h^* & 0 \\ 0 & 0 & 6h^* & 4h^*(2h^*+1) & 0 \\ \frac{c}{2}+4h & 6h & 0 & 0 & 0 \\ 6h & 4h(2h+1) & 0 & 0 & 0 \\ 0 & 0 & 0 & 0 & 4|h|^2 \end{pmatrix} \tag{A.6}$$

It is a hermitian matrix and thus has as required only real eigenvalues, but it is not positive definite and hence there are again negative norm states. This structure persists at higher levels: the Kac-matrices are hermitian but cannot be positive definite due to their block diagonal structure.

**Null states.** One may ask whether representations with this norm have null states. The answer is the same as for the ordinary null-state condition. The determinant of the Kac-matrices discussed above factorizes into a product of ordinary Virasoro Kac-matrices and in particular the condition for null-vectors is equivalent. This can also be seen by noting that null-states form themselves modules under the algebra. The module of null-descendants has a primary state known as the singular vector which is annihilated by positive modes of $L_n^+$ and $L_n^-$. The existence of the singular vector is independent of the norm we use and hence leads to the same conditions. Thus the representation does not have null states unless $h = \frac{c-1}{24} - p^2$ with $p = \frac{rb}{2} + \frac{s}{2b}$ with $r, s \in \mathbb{Z}_{\geqslant 1}$, which leads to *two* linearly independent singular vectors at level $rs$.

**No-ghost theorem.** In string theory using old covariant quantization, only states $|\psi\rangle$ with

$$(L_n - \delta_{n,0})|\psi\rangle = (\tilde{L}_n - \delta_{n,0})|\psi\rangle = 0 \ , \qquad n \geqslant 0 \tag{A.7}$$

are physical, where $L_n = L_n^+ + L_n^-$. We also assume that $b^2 \in i\mathbb{R}$ as is necessary for the definition of the worldsheet theory. The mass-shell condition implies

$$2\operatorname{Re}(h) + N = 1 \ , \tag{A.8}$$

where $N \in \mathbb{Z}_{\geqslant 0}$ is the descendant level of the state. For $h \neq h_{\langle r,s\rangle}$, any physical state is BRST-equivalent to a primary state. This can for example be seen by computing the torus partition function of the combined worldsheet + ghost system. Roughly speaking the $\mathfrak{bc}$-ghosts remove two oscillator degrees worth of freedom and thus reduce everything to primary states. For $h = h_{\langle r,s\rangle}$, we have

$$2\operatorname{Re}(h_{\langle r,s\rangle}) = 1 - rs \ , \tag{A.9}$$

and thus we can potentially have an extra physical state at level $N = rs$. This extra state is however precisely the null-state that we discussed above and thus decouples from the spectrum.

We conclude that up to BRST-equivalence, the only normalizable physical states are primary states with $2\operatorname{Re}(h) = 1$ as claimed in section 2.1.[22]

## A.3  Conformal blocks

Correlation functions of this theory may essentially be defined as usual. The definition of conformal blocks is independent of the norm that one chooses on the Hilbert space, since they may for example be defined as a holomorphic sections of a certain line bundle over Teichmüller space [74].[23] This notion makes no mention of the norm and thus we can define conformal blocks in both theories as usual. Similarly we can define three-point functions as normal which tells us that the correlation functions can be computed as in eq. (2.17).

# B  Numerical implementation of $\Gamma_b(x)$

In the numerical evaluation of $\mathsf{A}_{0,4}^{(b)}(p_1, p_2, p_3, p_4)$ in section 6, we will make use of an efficient implementation of a single-product formula for the Barnes double gamma function $\Gamma_b(z)$, derived in [76], and that we briefly review here.

Following [76], we define the coefficients

$$A(\tau) \equiv \frac{\tau}{2} \log(2\pi\tau) + \frac{1}{2}\log(\tau) - \tau C(\tau) \,, \tag{B.1}$$

$$B(\tau) \equiv -\tau \log(\tau) - \tau^2 D(\tau) \,, \tag{B.2}$$

where the so-called gamma modular forms can be expanded as

$$C(\tau) \equiv \sum_{k=1}^{m-1} \psi(k\tau) + \frac{1}{\tau}\log(\sqrt{2\pi}) - \sum_{\ell=0}^{m} \frac{B_\ell \tau^{\ell-1}}{\ell!} \psi^{(\ell-1)}(m\tau) \,, \tag{B.3}$$

$$D(\tau) \equiv \sum_{k=1}^{m-1} \psi^{(1)}(k\tau) - \sum_{\ell=0}^{m} \frac{B_\ell \tau^{\ell-1}}{\ell!} \psi^{(\ell)}(m\tau) \,, \tag{B.4}$$

where $\psi^{(m)}(z) \equiv \frac{\mathrm{d}^{m+1}}{\mathrm{d}z^{m+1}} \log \Gamma(z)$ denotes the polygamma function (and $\psi(z) \equiv \psi^{(0)}(z)$, $\psi^{(-1)}(z) = \log\Gamma(z)$), and $B_\ell$ are the Bernoulli numbers. Here $m$ is a cutoff. Using

---

[22]The ground ring operators discussed in section 4.1 define further non-normalizable physical states at ghost number 0. Since they crucially involve the ghosts, they cannot easily be described in old coveriant quantization.

[23]This is especially well-known for conformal blocks with a degenerate external field which satisfy the BPZ differential equation [75] on the four-punctured sphere. The derivation of the BPZ differential equation does not need a reality condition.

$m = 10$ is more than enough to get good numerical results. Further, define the polynomials $P_n(z; \tau)$ via the recursion

$$P_n(z; \tau) = z^{n-1} - \frac{1}{\tau} \sum_{k=1}^{n-1} \binom{n+1}{k+2} \frac{(1+\tau)^{k+2} - 1 - \tau^{k+2}}{(n-k+1)(n-k+2)} P_{n-k}(z; \tau) . \qquad (B.5)$$

With these definitions, we can implement the product formula of [76, Theorem 1] and write the three-point coefficients as[24]

$$\log C_b(p_1, p_2, p_3) = \sum_{\substack{\sigma_j = \pm \\ j=1,2,3}} \log\Gamma_b^{\text{aux}} \left( \tfrac{1}{2}(b + b^{-1}) + \sigma_1 p_1 + \sigma_2 p_2 + \sigma_3 p_3 \right)$$

$$- \sum_{j=1}^{3} \sum_{\sigma=\pm} \log\Gamma_b^{\text{aux}} \left( b + b^{-1} + 2\sigma p_j \right)$$

$$+ \log\Gamma_b^{\text{aux}} \left( 2(b + b^{-1}) \right) - 3\log\Gamma_b^{\text{aux}} \left( b + b^{-1} \right) - \frac{1}{2}\log(2) , \quad (B.6)$$

where the function $\log\Gamma_b^{\text{aux}}(z)$ is the logarithm of $\Gamma_b$ up to an additive $z$-independent constant that cancels between the numerator and denominator in the definition of $C_b(p_1, p_2, p_3)$, and is computed as

$$\log\Gamma_b^{\text{aux}}(z) = \frac{bz}{2}\log(2\pi) + \frac{z(Q-z)}{2}\log(b) + \log\Gamma(zb) - zb^{-1}A(b^2) - \frac{1}{2}(zb^{-1})^2 B(b^2)$$

$$- \sum_{m=1}^{N} \left( \log\Gamma(mb^2) - \log\Gamma(zb + mb^2) + zb\,\psi(mb^2) + \frac{1}{2}(zb)^2\psi^{(1)}(mb^2) \right)$$

$$- (zb)^3 \sum_{k=1}^{M} \frac{1}{N^k} \frac{(-b^2)^{-k-1} P_k(zb, b^2)}{k(k+1)(k+2)} , \qquad (B.7)$$

where the positive integers $N$ and $M$ are cutoffs for the product formula. It is also convenient numerically to use the Mathematica function LogGamma for the numerical evaluation of the logarithm of the Gamma function. For the numerical results presented in section 6.2, we find it sufficient to take the cutoff value $m = 10$ in (B.3) and (B.4), and $M = 10$, and $N = 30$ in (B.7).

## C   Some computations

In this appendix we perform some of the computations that were referenced in the main part of this paper.

### C.1   Expressing $\mathbf{A}_{1,1}^{(b)}$ through $\mathbf{A}_{0,4}^{(b)}$

We derive eq. (3.26).

---

[24]It is numerically more stable to compute the logarithm of the three-point functions since this avoids large cancellations.

**A relation between the OPE densities.** One can first relate the OPE densities appearing in the Liouville four-point function on the sphere and the Liouville one-point function on the torus as follows,

$$\rho_{\sqrt{2}b}(\sqrt{2}p)\, C_{\sqrt{2}b}\left(\frac{\sqrt{2}b}{4}, \frac{\sqrt{2}b}{4}, \sqrt{2}p\right) C_{\sqrt{2}b}\left(\frac{\sqrt{2}b}{4}, \sqrt{2}p, \frac{p_1}{\sqrt{2}}\right)$$

$$= \mathcal{R}_b \times 2^{16p^2 - 2p_1^2} \frac{\Gamma_b\left(\frac{Q}{2} \pm p_1 + \frac{1}{2b}\right)}{\Gamma_b\left(\frac{Q}{2} \pm p_1\right)} \rho_b(p)\, C_b(p, p, p_1)\,, \quad \text{(C.1)}$$

where $\mathcal{R}_b$ is independent of the momenta and given by

$$\mathcal{R}_b \equiv \frac{1}{\pi^{5/2}} 2^{-\frac{1}{2}\left(\frac{2}{b^2} + 1 - b^2\right)} b^{\frac{1}{b^2} + \frac{7}{2}} \frac{\Gamma(\frac{1}{b^2})\Gamma(b^2)^3\Gamma(2 + b^2)}{\Gamma(2 + 2b^2)^2} \sin\left(\frac{\pi}{2b^2}\right)^2 \frac{\Gamma_b(-\frac{1}{2b})^2}{\Gamma_b(\frac{1}{b})^2}\,. \quad \text{(C.2)}$$

A similar expression appeared in [40]. This can be demonstrated straightforwardly by the taking the ratio of the left- and right-hand side, using the doubling formula for the double Gamma-function (see e.g. [13, eq. (C.6)]) and using the shift equation repeatedly.

**Worldsheet integrand.** In the sphere four point function of the Liouville string we combine (C.1) with the expression for the $b \to b^-$ theory. It follows from (2.33) that

$$\frac{\Gamma_b\left(\frac{Q}{2} \pm p_1 + \frac{1}{2b}\right) \Gamma_{b^-}\left(\frac{Q^-}{2} \pm p_1^- + \frac{1}{2b^-}\right)}{\Gamma_b\left(\frac{Q}{2} \pm p_1\right) \Gamma_{b^-}\left(\frac{Q^-}{2} \pm p_1^-\right)} = 2e^{-\frac{i\pi}{8b^2}} \frac{\vartheta_3(bp_1|b^2)}{\vartheta_4(bp_1|b^2)} \cos\left(\frac{\pi p_1}{b}\right)\,, \quad \text{(C.3)}$$

where we used (2.11).

In particular we obtain for $p_1 = Q/2$ and consequently $p_1^- = iQ/2$,

$$\frac{\Gamma_b\left(-\frac{1}{2b}\right)^2}{\Gamma_b\left(\frac{1}{b}\right)} \frac{\Gamma_{-ib}\left(-\frac{i}{2b}\right)^2}{\Gamma_{-ib}\left(\frac{i}{b}\right)\Gamma_{-ib}(-ib)}$$

$$= -e^{-\frac{i\pi}{8b^2}} \sqrt{\frac{\pi}{2}} \frac{b^{5/2} e^{-\frac{i\pi}{4}(1 + \frac{1}{b^2})}}{\sin^2\left(\frac{\pi}{2b^2}\right) \vartheta_4\left(b\frac{Q}{2}|b^2\right)} \lim_{\epsilon \to 0} \Gamma_b(\epsilon)\vartheta_3\left(b\left(\frac{Q}{2} - \epsilon\right)\Big|b^2\right)\,. \quad \text{(C.4)}$$

The Jacobi-theta function vanishes for $z = Q/2 - \epsilon$, but expanding to order $\epsilon$ gives a factor of $\eta(b^2)^3$. We similarly evaluate the residue of $\Gamma_b(\epsilon)$ from the shift equation to obtain

$$\frac{\Gamma_b\left(-\frac{1}{2b}\right)^2}{\Gamma_b\left(\frac{1}{b}\right)^2} \frac{\Gamma_{-ib}\left(-\frac{i}{2b}\right)^2}{\Gamma_{-ib}\left(\frac{i}{b}\right)^2} = \pi e^{-\frac{i\pi}{8b^2}} \frac{b^5 e^{-\frac{i\pi}{4}(1 + \frac{1}{b^2} + b^2)}}{\sin^2\left(\frac{\pi}{2b^2}\right)} \frac{\eta(b^2)^3}{\vartheta_4\left(b\frac{Q}{2}|b^2\right)}\,, \quad \text{(C.5)}$$

where $\eta(\tau)$ denotes the Dedekind $\eta$-function which satisfies the relation $2\eta(\tau)^3 = \vartheta_2(0|\tau)\vartheta_3(0|\tau)\vartheta_4(0|\tau)$ and we used $\Gamma_b(b) = b\Gamma_b(b^{-1})$. To evaluate the moduli space

integral we additionally need the relation between the Virasoro conformal blocks, which is simply [40, 77]

$$\mathcal{H}_{1,1}^{(b)}(q^2) = \mathcal{H}_{0,4}^{\sqrt{2}b}\left(\frac{\sqrt{2}b}{4}, \frac{\sqrt{2}b}{4}, \frac{\sqrt{2}b}{4}, \frac{p_1}{\sqrt{2}}; \sqrt{2}p\,\middle|\,q\right) . \tag{C.6}$$

**Assembling the result.** We can now combine all the integredients. Accounting for the leg factors (2.20a) and the normalization $C_{\mathrm{S}^2}^{(b)}$ for the sphere and $C_{\mathrm{T}^2}^{(b)} = 1$ for the torus leads to the relation

$$\mathsf{A}_{0,4}^{(\sqrt{2}b)}\left(\frac{b}{2\sqrt{2}}, \frac{b}{2\sqrt{2}}, \frac{b}{2\sqrt{2}}, \frac{p_1}{\sqrt{2}}\right) = \kappa(b, p_1)\,\mathsf{A}_{1,1}^{(b)}(p_1) , \tag{C.7}$$

where

$$\kappa(b, p_1) \equiv 2 \times 6\pi^2 \mathcal{R}_b\,\mathcal{R}_{b^-} \times \frac{\mathcal{N}_{\sqrt{2}b}(\frac{b}{2\sqrt{2}})^3 \mathcal{N}_{\sqrt{2}b}(\frac{p_1}{\sqrt{2}})}{\mathcal{N}_b(p_1)} C_{\mathrm{S}^2}^{(\sqrt{2}b)} \frac{2}{(2\pi)^2} \frac{\vartheta_3(bp_1|b^2)}{\vartheta_4(bp_1|b^2)} \cos\left(\frac{\pi p_1}{b}\right)$$

$$= 12b\vartheta_3(0|b^2)\vartheta_4(0|b^2) \times \frac{\vartheta_3(bp_1|b^2)}{\vartheta_4(bp_1|b^2)} . \tag{C.8}$$

The $6\pi^2$ is due to the fact that we map the fundamental domain in the cross ratio $z$-plane, $= \{z \in \mathbb{C} \,|\, \mathrm{Re}\, z \leqslant \frac{1}{2}, |1 - z| \leqslant 1\}$, of the sphere four-point diagram, to the fundamental domain $F_0 = \{t \in \mathbb{C} \,|\, -\frac{1}{2} \leqslant \mathrm{Re}\, t \leqslant \frac{1}{2}, |t| \geqslant 1\}$ in the complex $t$-plane via the change of variables

$$t = i\frac{K(1 - z)}{K(z)} , \tag{C.9}$$

see the discussion around (6.4). The $2 = (\sqrt{2})^2$ is the Jacobian when we go from an integration over $\sqrt{2}p$ on the sphere to $p$ on the torus in (C.1) and similarly for the second Liouville theory. We also used some standard theta-function identities to simplify the result.

## C.2 Verifying triality symmetry of the four-point function

We demonstrate the triality symmetry of (1.6c). The two contributions are separately triality symmetric.

**First term.** Let us start with the first one. Obviously $\mathsf{V}_{0,4}^{(b)}(i\boldsymbol{p})$ is triality symmetric. Thus it suffices to consider

$$\sum_{m=1}^{\infty} \frac{2b^2 \prod_{j=1}^{4} \sin(2\pi mb p_j)}{\sin(\pi mb^2)^2}$$

$$= \sum_{m_1,m_2=1}^{\infty} \int_0^1 \mathrm{d}(bp)\, \frac{2b(-1)^{m_1} \sin(2\pi m_1 bp_1) \sin(2\pi m_1 bp_2) \sin(2\pi m_1 bp)}{\sin(\pi m_1 b^2)}$$

$$\times \frac{2b(-1)^{m_2} \sin(2\pi m_2 bp_3) \sin(2\pi m_2 bp_4) \sin(2\pi m_2 bp)}{\sin(\pi m_2 b^2)}$$

$$= \int_0^1 \mathrm{d}(bp)\, \mathsf{a}_{0,3}^{(b)}(p_1, p_2, p)\, \mathsf{a}_{0,3}^{(b)}(p_3, p_4, p)\,, \tag{C.10}$$

where we used that the integral over $p$ projects to $m_1 = m_2$. Since the precise form of the measure did not enter the argument in section 3.3, we can continue like there to see that this term has the correct triality symmetry.

**Second term.**  For the second term, we can use a similar trick. Observe that we can write it as

$$\int_0^{e^{-\frac{\pi i}{4}}\infty} (-2p\mathrm{d}p)\left[ \sum_{m_1,m_2=1}^{\infty} \frac{2b(-1)^{m_1}\sin(2\pi m_1 bp_1)\sin(2\pi m_1 bp_2)\sin(2\pi m_1 bp)}{\sin(\pi m_1 b^2)} \right.$$

$$\times \frac{2b(-1)^{m_2}\sin(2\pi m_2 bp_3)\sin(2\pi m_2 bp_4)\sin(2\pi m_2 bp)}{\sin(\pi m_2 b^2)}$$

$$\left. - \frac{2b^2 \prod_{j=1}^4 \sin(2\pi m bp_j)}{\sin(\pi m b^2)^2} \right]$$

$$= \int_0^{e^{-\frac{\pi i}{4}}\infty} (-2p\mathrm{d}p)\left[ \mathsf{a}_{0,3}^{(b)}(p_1, p_2, p)\, \mathsf{a}_{0,3}^{(b)}(p_3, p_4, p) - \frac{2b^2 \prod_{j=1}^4 \sin(2\pi m bp_j)}{\sin(\pi m b^2)^2} \right]. \tag{C.11}$$

The integral is not convergent, but can be made sense of by decomposing all sine factors into exponentials and rotating the contour for each term slightly into the complex direction to get something convergent.[25] This only fails when $m_1 = m_2$ and we subtract the problematic terms. The subtracted term is identical to the first term that we considered above and hence triality symmetric. We can then continue with the argument as in section 3.3 to conclude that this contribution is also triality symmetric.

## C.3  Verifying the relation between $\mathsf{a}_{0,4}^{(b)}$ and $\mathsf{a}_{1,1}^{(b)}$

In this appendix, we spell out the strategy outlined in section 5.5.

**Discontinuous piece.**  Let us first check the discontinuous piece in the form (5.7). Some of the appearing three-point functions vanish and the expression simplifies to

$$\mathsf{a}_{0,4}^{(\sqrt{2}b)}\left(\tfrac{p_1}{\sqrt{2}}, \tfrac{b}{2\sqrt{2}}, \tfrac{b}{2\sqrt{2}}, \tfrac{b}{2\sqrt{2}}\right)^{(2)} = -\frac{3}{2\sqrt{2}\pi^2 b}\sum_{k=0}^{\infty}\sum_{\sigma_1,\sigma_2=\pm}\sigma_1\sigma_2\mathrm{Li}_2\left(-e^{\pi ib(2\sigma_1 p_1 + (2+4k+\sigma_2)b)}\right)$$

$$\times \mathsf{a}_{0,3}^{(\sqrt{2}b)}\left(\tfrac{b}{2\sqrt{2}}, \tfrac{b}{2\sqrt{2}}, \tfrac{b}{\sqrt{2}} + \tfrac{1}{2\sqrt{2}b} + \tfrac{\sigma_1 p_1}{\sqrt{2}} + \tfrac{b\sigma_2}{2\sqrt{2}}\right). \tag{C.12}$$

By matching poles and zeros, one can verify that

$$\mathsf{a}_{0,3}^{(\sqrt{2}b)}\left(\tfrac{b}{2\sqrt{2}}, \tfrac{b}{2\sqrt{2}}, \tfrac{b}{\sqrt{2}} + \tfrac{1}{2\sqrt{2}b} + \tfrac{\sigma_1 p_1}{\sqrt{2}} + \tfrac{b\sigma_2}{2\sqrt{2}}\right) = -\sigma_2 \frac{b\vartheta_3(0, b^2)\vartheta_4(0, b^2)\vartheta_3(bp_1, b^2)}{\sqrt{2}\,\vartheta_4(bp_1, b^2)}. \tag{C.13}$$

---

[25]Alternatively, we can include a regulating term $e^{-\varepsilon bp}$, which for $\varepsilon > 0$ renders the integral convergent. The limit $\varepsilon \to 0$ is then well-defined.

One can then combine the sum over $k$ and $\sigma_2$ into a single sum by setting $2k + \frac{1+\sigma_2}{2} = k' \in \mathbb{Z}_{\geqslant 0}$. Thus we have

$$\mathsf{a}_{0,4}^{(\sqrt{2}b)}\left(\tfrac{p_1}{\sqrt{2}}, \tfrac{b}{2\sqrt{2}}, \tfrac{b}{2\sqrt{2}}, \tfrac{b}{2\sqrt{2}}\right)^{(2)} = \frac{3}{4\pi^2} \frac{\vartheta_3(0, b^2)\vartheta_4(0, b^2)\vartheta_3(bp_1, b^2)}{\vartheta_4(bp_1, b^2)}$$

$$\times \sum_{k'=0}^{\infty} \sum_{\sigma_1=\pm} \sigma_1 \mathrm{Li}_2\left(-e^{2\pi i b(\sigma_1 p_1 + (k'+\frac{1}{2})b)}\right)$$

$$= \frac{12b\vartheta_3(0, b^2)\vartheta_4(0, b^2)\vartheta_3(bp_1, b^2)}{\vartheta_4(bp_1, b^2)} \mathsf{a}_{1,1}^{(b)}(p_1)^{(2)}, \qquad \text{(C.14)}$$

where we matched to the second term of (5.10).

**Meromorphic piece.** We can similarly compute from the first piece of (5.5)

$$\mathsf{a}_{0,4}^{(\sqrt{2}b)}\left(\tfrac{p_1}{\sqrt{2}}, \tfrac{b}{2\sqrt{2}}, \tfrac{b}{2\sqrt{2}}, \tfrac{b}{2\sqrt{2}}\right)^{(1)} = 12b^2 \mathsf{V}_{1,1}^{(b)}(ip_1) \sum_{k=0}^{\infty} \sum_{\sigma_1,\sigma_2=\pm} \left[\frac{3k\sigma_1\sigma_2}{e^{-2\pi i b(\sigma_1 p_1 + b(2k+\frac{\sigma_2}{2}))} - 1}\right.$$

$$\left. - \frac{k\sigma_1\sigma_2}{e^{-2\pi i b(\sigma_1 p_1 + b(2k+\frac{3\sigma_2}{2}))} - 1}\right]$$

$$= 12b^2 \mathsf{V}_{1,1}^{(b)}(ip_1) \sum_{k=0}^{\infty} \sum_{\sigma_1=\pm} \frac{(2k+1)(-1)^k\sigma_1}{e^{-2\pi i b(\sigma_1 p_1 + b(k+\frac{1}{2}))} - 1}, \qquad \text{(C.15)}$$

where we combined equal terms appropriately in the last line. Notice that the poles of this infinite sum are precisely compensated by the zeros of $\vartheta_4(bp_1, b^2)$. Thus all poles in the right-hand side of (3.26) are generated by the zeros of $\vartheta_3(bp_1, b^2)$. The residue is for $r, s \in \mathbb{Z} + \frac{1}{2}$

$$\operatorname*{Res}_{p_1=rb+sb^{-1}} \frac{\vartheta_4(bp_1|b^2)}{12b\vartheta_3(0|b^2)\vartheta_4(0|b^2)\vartheta_3(bp_1|b^2)} \mathsf{a}_{0,4}^{(\sqrt{2}b)}\left(\tfrac{p_1}{\sqrt{2}}, \tfrac{b}{2\sqrt{2}}, \tfrac{b}{2\sqrt{2}}, \tfrac{b}{2\sqrt{2}}\right)^{(2)}$$

$$= \frac{b\vartheta_4(rb^2 + s|b^2)\mathsf{V}_{1,1}^{(b)}(i(rb + sb^{-1}))}{\vartheta_3(0|b^2)\vartheta_4(0|b^2)}$$

$$\times \operatorname*{Res}_{p_1=rb+sb^{-1}} \frac{1}{\vartheta_3(bp_1|b^2)} \sum_{k=0}^{\infty} \sum_{\sigma_1=\pm} \frac{(2k+1)(-1)^{k+1}\sigma_1}{e^{-2\pi i b^2(k+\sigma_1 r + \frac{1}{2})} + 1}$$

$$= \frac{\mathsf{V}_{1,1}^{(b)}(i(rb + sb^{-1}))}{2\pi i} \frac{\vartheta_2(0|b^2)}{\vartheta_3(0|b^2)\vartheta_4(0|b^2)\eta(b^2)^3} \sum_{k=0}^{\infty} \sum_{\sigma_1=\pm} \frac{(2k+1)(-1)^{k+1}\sigma_1}{e^{-2\pi i b^2(k+\frac{1+\sigma_1}{2})} + 1}, \qquad \text{(C.16)}$$

where we used the periodicity properties of the theta-functions and that the infinite sum only depends on $r$ via $(-1)^r$. Let us finally simplify the infinite sum. We have

$$\sum_{k=0}^{\infty} \sum_{\sigma_1=\pm} \frac{(2k+1)(-1)^{k+1}\sigma_1}{e^{-2\pi i b^2(k+\frac{1+\sigma_1}{2})} + 1} = \frac{1}{2} + \sum_{k=1}^{\infty} \frac{4k(-1)^k q^k}{1 + q^k}$$

$$= \frac{1}{2} - \sum_{k=1}^{\infty} \sum_{r=1}^{\infty} 4k(-1)^r e^{2\pi i k(r\tau + \frac{1}{2})}$$

$$= \frac{1}{2} + \frac{1}{\pi^2} \sum_{r=1}^{\infty} \sum_{m \in \mathbb{Z} + \frac{1}{2}} \frac{(-1)^r}{(r\tau + m)^2}$$

$$= \frac{1}{2\pi^2} \sum_{r \in \mathbb{Z},\, m \in \mathbb{Z} + \frac{1}{2}} \frac{(-1)^r}{(r\tau + m)^2} \,, \tag{C.17}$$

where $\tau = b^2$. This is a twisted version of the Eisenstein series $E_2(\tau)$. Because of the $(-1)^r$, the sum is actually convergent and defines a modular form of weight 2 under the congruence subgroup

$$\Gamma_0(4) = \left\{ \begin{pmatrix} a & b \\ c & d \end{pmatrix} \in \mathrm{PSL}(2, \mathbb{Z}) \,\middle|\, c \equiv 0 \bmod 4 \right\}, \tag{C.18}$$

which has index 6 inside $\mathrm{PSL}(2, \mathbb{Z})$. Let us recall the Sturm bound [78], which is the integer

$$\left\lfloor \frac{km}{12} \right\rfloor = 1 \,, \tag{C.19}$$

where $m = [\mathrm{PSL}(2, \mathbb{Z}) : \Gamma_0(4)] = 6$ and $k = 2$ is the weight. This integer tells us that we only need to check the agreement of two modular forms in $\Gamma_0(4)$ of weight 2 to first order in $q$. We are then guaranteed that all other orders agree as a consequence of the finite-dimensionality of the space of modular forms.[26] We thus find that

$$\sum_{k=0}^{\infty} \sum_{\sigma_1 = \pm} \frac{(2k+1)(-1)^{k+1}\sigma_1}{e^{-2\pi i b^2 \left(k + \frac{1 + \sigma_1}{2}\right)} + 1} = \frac{1}{2} \vartheta_3(\tau)^2 \vartheta_4(\tau)^2 \,. \tag{C.20}$$

Indeed, the right-hand side can also be verified to be a modular form under $\Gamma_0(4)$ of weight 2 and has the same first two terms of the $q$-expansion.

Going back to (C.16) and using also that $\eta(\tau)^3 = \frac{1}{2}\vartheta_2(0|\tau)\vartheta_3(0|\tau)\vartheta_4(0|\tau)$, we get

$$\operatorname*{Res}_{p_1 = rb + sb^{-1}} \frac{\vartheta_4(bp_1|b^2)}{12b\vartheta_3(0|b^2)\vartheta_4(0|b^2)\vartheta_3(bp_1|b^2)} \mathsf{a}_{0,4}^{(\sqrt{2}b)} \left( \frac{p_1}{\sqrt{2}}, \frac{b}{2\sqrt{2}}, \frac{b}{2\sqrt{2}}, \frac{b}{2\sqrt{2}} \right)^{(2)}$$

$$= \frac{1}{2\pi i} \mathsf{V}_{1,1}^{(b)}(i(rb + sb^{-1}))$$

$$= \operatorname*{Res}_{p_1 = rb + sb^{-1}} \mathsf{a}_{1,1}^{(b)}(p_1)^{(2)} \,. \tag{C.21}$$

This shows that all the residues agree.

**Finishing the argument.** It then also quickly follows that (C.7) is satisfied by $\mathsf{a}_{0,4}^{(b)}$ and $\mathsf{a}_{1,1}^{(b)}$. Indeed, both the left- and the right-hand side grow only polynomially for large momenta as discussed around eq. (5.33). Thus the left-hand side minus the right-hand side could at most be a polynomial in $p_1$. However, both the left-hand side and the right-hand side of (3.26) vanish for $p_1 = \frac{m}{2b}$, $m \in \mathbb{Z}$, which proves their equality.

---

[26] In the present case, the space of modular forms is 2-dimensional.

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
