# Peer review of "The complex Liouville string: the worldsheet"

_SciPost Physics_

## Round 2 · Referee Report · Sylvain Ribault (Referee 1) · 2025-3-14

Report

Warnings issued while processing user-supplied markup:

  • Inconsistency: plain/Markdown and reStructuredText syntaxes are mixed. Markdown will be used.
    Add "#coerce:reST" or "#coerce:plain" as the first line of your text to force reStructuredText or no markup.
    You may also contact the helpdesk if the formatting is incorrect and you are unable to edit your text.

Two-dimensional theories of quantum gravity can be built by coupling Liouville theory with a matter CFT, such that their central charges add up to 26. If the matter CFT is simple enough, one can hope to solve the resulting theory. This amounts to computing its correlation numbers, which are defined as integrals of the coupled CFT's correlation functions over their moduli, including the fields' positions.

This idea has first been implemented in the case of minimal Liouville gravity, whose matter CFT is a minimal model. More recently, the authors of the present preprint have introduced the Virasoro minimal string, whose matter CFT is Liouville theory with a central charge c<1. And in the present preprint, the matter CFT is Liouville theory with $c\in 13+i\mathbb{R}$ --- then the gravitational Liouville CFT also has a central charge of that type. This leads to a theory that the authors call the complex Liouville string.

Computing correlation numbers is a technical challenge. Depending on the case, different methods are used:

  1. In minimal Liouville gravity, since matter fields are degenerate, they give rise to a ground ring of operators whose derivatives are BRST exact. This allows moduli integrals to be performed, by localizing the integrals on singular points.

  2. In the Virasoro minimal string, a relation with intersection theory on moduli spaces shows that correlation numbers are polynomial functions of the fields' momentums. These functions can be computed using a recursion relation, deduced from the matrix integral representation of the theory.

  3. In the complex Liouville string, the correlation numbers are bootstrapped by solving analyticity constraints. These constraints are deduced from the analytic properties of the correlation functions in the coupled CFT, including properties that are obtained with the help of the ground ring. This approach is conceptually more straightforward than in the case of the Virasoro minimal string. It is nevertheless technically nontrivial to implement, and in fact the resulting correlation numbers are more complicated than polynomials. In order for the solution of their equations to be unique, the authors have to make assumptions on their asymptotic properties: these assumptions are simple and reasonable, but they are not proved.

One may wonder whether a similar bootstrap-type approach could work for the Virasoro minimal string. The authors argue in footnote 13 that there is no ground ring in that case, because of the well-known feature of c<1 Liouville theory that degenerate fields are not limit of generic fields. This argument is unconvincing, for two reasons: first, degenerate fields can be consistently added to c<1 Liouville theory, and generic fields can be recovered as limits of degenerate fields. Second, since correlation numbers are polynomial, degenerate correlation numbers could plausibly be limits of generic correlation numbers, even though this does not work for CFT correlation functions.

It would also be interesting to study limits that relate minimal Liouville gravity, the Virasoro minimal string, and the complex Liouville string. At the level of the underlying matter CFTs, it is possible to deduce generalized minimal models (= CFTs of diagonal degenerate fields) from Liouville gravity at generic central charge. From generalized minimal models, minimal models and c<1 Liouville theory follow. Do such relations hold at the level of correlation numbers?

These are just some of the technical questions that arise from this rich and interesting work. More conceptual questions arise from its applications, which are the subjects of the authors' subsequent articles.

This article is generally clear, well-written, and convincing. I have a few suggestions for small improvements:

  1. Appendix B does not seem to be referenced from elsewhere.

  2. There are too many footnotes! Footnotes 1 and 2 are even referenced contiguously. I will make a few suggestions about specific footnotes, but I believe most of them could be integrated in the main text.

  3. In (1.6b) and (1.6c), the quantum volumes are $m$-independent and could be pulled out of the sums, making the formulas more readable.

  4. The quantum volumes' expressions (1.4) and (1.5) are important formulas and do not belong to a footnote.

  5. On page 5, the outline of the paper is superfluous: the table of contents is clear enough.

  6. On page 6, it is not quite correct that Liouville theory with complex central charge has no inner product. It has a bilinear product called the Shapovalov form. What does not exist is a Hermitian sesquilinear form.

  7. On page 8, the statement "However ... standard way" is vague and confusing. Footnote 4 could be integrated in the main text, and Appendix A referenced only once (rather than 3 times in the same paragraph).

  8. After (2.15), what is meant exactly by conventions?

  9. In (2.20a), the formula could be written as $b$-dependent prefactors times $p$-dependent factors.

  10. How do we deduce the special cases (2.19) and (2.21) from the general formula (2.22)? That formula involves ghosts, it is not clear why they are absent from the special cases.

  11. After (2.26), the discussion about trivial zeros is clumsy and redundant, especially the "at least".

  12. Page 14, the derivation of the infinite sum representation is nicely improved compared to the first arXiv version. However, I do not understand why on page 15 the word "second" disappeared from the title of the paragraph about the second infinite sum representation.

  13. In Section 4 and also in (2.22) and appendix A.1, some right-moving quantities seem to come with a tilde rather than a bar. It is not clear to me why this unconventional notation is used. In any case it should be stated explicitly. Notice that the Liouville action (2.1) also involves tildes.

  14. On page 49, is the simplest choice 1 or 0? Could we choose any polynomial function of the quantum volume?

  15. On page 49, is the uppercase hatted A the same as the lowercase a?

  16. In (5.32), the notation O(1) does not seem appropriate: it means a function that is at most constant, whereas we want a function that is a least constant.

  17. Equation (5.33) looks like an important property of the correlation numbers. It could be emphasized more, and maybe proved from first principles.

As a matter of principle I do not make recommendations on whether to publish articles or not. Editorial decisions belong to editors. Since the computer system makes it necessary to have a recommendation, I picked the first one in the list.

Recommendation

Publish (surpasses expectations and criteria for this Journal; among top 10%)

---

## Round 2 · Referee Report · Anonymous (Referee 2) · 2025-6-1

Strengths

clear, well-written and original.

Report

This paper is first in a list of interesting papers on a new two-dimensional string model obtained by coupling two Liouville theories. Two other papers in the list have been already published in SciPost. In this paper the author show how to evaluate string amplitutes using their analytical structure and some reasonable assumptions. This is an interesting and original line of research, and the results are sound and convincing. I think that this is an excellent paper that deserves to be published.

Recommendation

Publish (surpasses expectations and criteria for this Journal; among top 10%)

---

## Editorial Decision

resubmitted